# Correlation Aware Sparsified Mean Estimation Using Random Projection

**Shuli Jiang**
Robotics Institute
Carnegie Mellon University
shulij@andrew.cmu.edu

**Pranay Sharma**
ECE
Carnegie Mellon University
pranaysh@andrew.cmu.edu

**Gauri Joshi**
ECE
Carnegie Mellon University
gaurij@andrew.cmu.edu

## Abstract

We study the problem of communication-efficient distributed vector mean estimation, a commonly used subroutine in distributed optimization and Federated Learning (FL). Rand-$k$ sparsification is a commonly used technique to reduce communication cost, where each client sends $k < d$ of its coordinates to the server. However, Rand-$k$ is agnostic to any correlations, that might exist between clients in practical scenarios. The recently proposed Rand-$k$-Spatial estimator leverages the cross-client correlation information at the server to improve Rand-$k$'s performance. Yet, the performance of Rand-$k$-Spatial is suboptimal. We propose the Rand-Proj-Spatial estimator with a more flexible encoding-decoding procedure, which generalizes the encoding of Rand-$k$ by projecting the client vectors to a random $k$-dimensional subspace. We utilize Subsampled Randomized Hadamard Transform (SRHT) as the projection matrix and show that Rand-Proj-Spatial with SRHT outperforms Rand-$k$-Spatial, using the correlation information more efficiently. Furthermore, we propose an approach to incorporate varying degrees of correlation and suggest a practical variant of Rand-Proj-Spatial when the correlation information is not available to the server. Experiments on real-world distributed optimization tasks showcase the superior performance of Rand-Proj-Spatial compared to Rand-$k$-Spatial and other more sophisticated sparsification techniques.

## 1 Introduction

In modern machine learning applications, data is naturally distributed across a large number of edge devices or clients. The underlying learning task in such settings is modeled by distributed optimization or the recent paradigm of Federated Learning (FL) [1, 2, 3, 4]. A crucial subtask in distributed learning is for the server to compute the mean of the vectors sent by the clients. In FL, for example, clients run training steps on their local data and once-in-a-while send their local models (or local gradients) to the server, which averages them to compute the new global model. However, with the ever-increasing size of machine learning models [5, 6], and the limited battery life of the edge clients, communication cost is often the major constraint for the clients. This motivates the problem of (empirical) *distributed mean estimation* (DME) under communication constraints, as illustrated in Figure 1. Each of the $n$ clients holds a vector $\mathbf{x}_i \in \mathbb{R}^d$, on which there are no distributional assumptions. Given a communication budget, each client sends a compressed version $\widehat{\mathbf{x}}_i$ of its vector to the server, which utilizes these to compute an estimate of the mean vector $\frac{1}{n} \sum_{i=1}^{n} \mathbf{x}_i$.

Quantization and sparsification are two major techniques for reducing the communication costs of DME. Quantization [7, 8, 9, 10] involves compressing each coordinate of the client vector to a given precision and aims to reduce the number of bits to represent each coordinate, achieving a constant reduction in the communication cost. However, the communication cost still remains $\Theta(d)$.

37th Conference on Neural Information Processing Systems (NeurIPS 2023).

Sparsification, on the other hand, aims to reduce the number of coordinates each clinet sends and compresses each client vector to only $k \ll d$ of its coordinates (e.g. Rand-$k$ [11]). As a result, sparsification reduces communication costs more aggressively compared to quantization, achieving better communication efficiency at a cost of only $O(k)$. While in practice, one can use a combination of quantization and sparsification techniques for communication cost reduction, in this work, we focus on the more aggressive sparsification techniques. We call $k$, the dimension of the vector each client sends to the server, the *per-client* communication budget.

Most existing works on sparsification ignore the potential correlation (or similarity) among the client vectors, which often exists in practice. For example, the data of a specific client in federated learning can be similar to that of multiple clients. Hence, it is reasonable to expect their models (or gradients) to be similar as well. To the best of our knowledge, [12] is the first work to account for *spatial* correlation across individual client vectors. They propose the Rand-$k$-Spatial family of unbiased estimators, which generalizes Rand-$k$ and achieves a better estimation error in the presence of cross-client correlation. However, their approach is focused only on the server-side decoding procedure, while the clients do simple Rand-$k$ encoding.

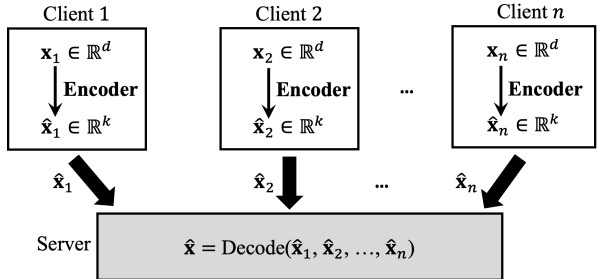

Figure 1: The problem of distributed mean estimation under limited communication. Each client $i \in [n]$ encodes its vector $\mathbf{x}_i$ as $\widehat{\mathbf{x}}_i$ and sends this compressed version to the server. The server decodes them to compute an estimate of the true mean $\frac{1}{n} \sum_{i=1}^{n} \mathbf{x}_i$.

In this work, we consider a more general encoding scheme that directly compresses a vector from $\mathbb{R}^d$ to $\mathbb{R}^k$ using a (random) linear map. The encoded vector consists of $k$ linear combinations of the original coordinates. Intuitively, this has a higher chance of capturing the large-magnitude coordinates ("heavy hitters") of the vector than randomly sampling $k$ out of the $d$ coordinates (Rand-$k$), which is crucial for the estimator to recover the true mean vector. For example, consider a vector where only a few coordinates are heavy hitters. For small $k$, Rand-$k$ has a decent chance of missing all the heavy hitters. But with a linear-maps-based general encoding procedure, the large coordinates are more likely to be encoded in the linear measurements, resulting in a more accurate estimator of the mean vector. Guided by this intuition, we ask:

> *Can we design an improved joint encoding-decoding scheme that utilizes the correlation information and achieves an improved estimation error?*

One naïve solution is to apply the same random rotation matrix $\boldsymbol{G} \in \mathbb{R}^{d \times d}$ to each client vector, before applying Rand-$k$ or Rand-$k$-Spatial encoding. Indeed, such preprocessing is applied to improve the estimator using quantization techniques on heterogeneous vectors [13, 10]. However, as we see in Appendix A.1, for sparsification, we can show that this leads to no improvement. But what happens if every client uses a different random matrix, or applies a random $k \times d$-dimensional linear map? How to design the corresponding decoding procedure to leverage cross-client correlation? As there is no way for one to directly apply the decoding procedure of Rand-$k$-Spatial in such cases. To answer these questions, we propose the Rand-Proj-Spatial family estimator. We propose a flexible encoding procedure in which each client applies its own random linear map to encode the vector. Further, our novel decoding procedure can better leverage cross-client correlation. The resulting mean estimator generalizes and improves over the Rand-$k$-Spatial family estimator.

Next, we discuss some reasonable restrictions we expect our mean estimator to obey. 1) *Unbiased.* An unbiased mean estimator is theoretically more convenient compared to a biased one [14]. 2) *Non-adaptive.* We focus on an encoding procedure that does not depend on the actual client data, as opposed to the *adaptive* ones, e.g. Rand-$k$ with vector-based sampling probability [11, 15]. Designing a data-adaptive encoding procedure is computationally expensive as this might require using an iterative procedure to find out the sampling probabilities [11]. In practice, however, clients often have limited computational power compared to the server. Further, as discussed earlier, mean estimation is often a subroutine in more complicated tasks. For applications with streaming data [16],

the additional computational overhead of adaptive schemes is challenging to maintain. Note that both Rand-$k$ and Rand-$k$-Spatial family estimator [12] are *unbiased* and *non-adaptive*.

In this paper, we focus on the severely communication-constrained case $nk \leq d$, when the server receives very limited information about any single client vector. If $nk \gg d$, we see in Appendix A.2 that the cross-client information has no additional advantage in terms of improving the mean estimate under both Rand-$k$-Spatial or Rand-Proj-Spatial, with different choices of random linear maps. Furthermore, when $nk \gg d$, the performance of both the estimators converges to that of Rand-$k$. Intuitively, this means when the server receives sufficient information regarding the client vectors, it does not need to leverage cross-client correlation to improve the mean estimator.

Our contributions can be summarized as follows:

1. We propose the Rand-Proj-Spatial family estimator with a more flexible encoding-decoding procedure, which can better leverage the cross-client correlation information to achieve a more general and improved mean estimator compared to existing ones.
2. We show the benefit of using Subsampled Randomized Hadamard Transform (SRHT) as the random linear maps in Rand-Proj-Spatial in terms of better mean estimation error (MSE). We theoretically analyze the case when the correlation information is known at the server (see Theorems 4.3, 4.4 and Section 4.3). Further, we propose a practical configuration called Rand-Proj-Spatial(Avg) when the correlation is unknown.
3. We conduct experiments on common distributed optimization tasks, and demonstrate the superior performance of Rand-Proj-Spatial compared to existing sparsification techniques.

## 2 Related Work

**Quantization and Sparsification.** Commonly used techniques to achieve communication efficiency are quantization, sparsification, or more generic compression schemes, which generalize the former two [17]. Quantization involves either representing each coordinate of the vector by a small number of bits [8, 9, 10, 18, 19, 20], or more involved vector quantization techniques [21, 22]. Sparsification [15, 23, 24, 25, 26], on the other hand, involves communicating a small number $k < d$ of coordinates, to the server. Common protocols include Rand-$k$ [11], sending $k$ uniformly randomly selected coordinates; Top-$k$ [27], sending the $k$ largest magnitude coordinates; and a combination of the two [28]. Some recent works, with a focus on distributed learning, further refine these communication-saving mechanisms [29] by incorporating temporal correlation or error feedback [14, 25].

**Distributed Mean Estimation (DME).** DME has wide applications in distributed optimization and FL. Most of the existing literature on DME either considers statistical mean estimation [30, 31], assuming that the data across clients is generated i.i.d. according to the same distribution, or empirical mean estimation [10, 32, 33, 12, 11, 34, 35], without making any distributional assumptions on the data. A recent line of work on empirical DME considers applying additional information available to the server, to further improve the mean estimate. This side information includes cross-client correlation [12, 13], or the memory of the past updates sent by the clients [36].

**Subsampled Randomized Hadamard Transformation (SRHT).** SRHT was introduced for random dimensionality reduction using sketching [37, 38, 39]. Common applications of SRHT include faster computation of matrix problems, such as low-rank approximation [40, 41], and machine learning tasks, such as ridge regression [42], and least square problems [43, 44, 45]. SRHT has also been applied to improve communication efficiency in distributed optimization [46] and FL [47, 48].

## 3 Preliminaries

**Notation.** We use bold lowercase (uppercase) letters, e.g. $\mathbf{x}$ ($\boldsymbol{G}$) to denote vectors (matrices). $\mathbf{e}_j \in \mathbb{R}^d$, for $j \in [d]$, denotes the $j$-th canonical basis vector. $\| \cdot \|_2$ denotes the Euclidean norm. For a vector $\mathbf{x}$, $\mathbf{x}(j)$ denotes its $j$-th coordinate. Given integer $m$, we denote by $[m]$ the set $\{1, 2, \ldots, m\}$.

**Problem Setup.** Consider $n$ geographically separated clients coordinated by a central server. Each client $i \in [n]$ holds a vector $\mathbf{x}_i \in \mathbb{R}^d$, while the server wants to estimate the mean vector $\bar{\mathbf{x}} \triangleq \frac{1}{n} \sum_{i=1}^{n} \mathbf{x}_i$. Given a per-client communication budget of $k \in [d]$, each client $i$ computes $\widehat{\mathbf{x}}_i$ and sends it to the central server. $\widehat{\mathbf{x}}_i$ is an approximation of $\mathbf{x}_i$ that belongs to a random $k$-dimensional

subspace. Each client also sends a random seed to the server, which conveys the subspace information, and can usually be communicated using a negligible amount of bits. Having received the encoded vectors $\{\widehat{\mathbf{x}}_i\}_{i=1}^n$, the server then computes $\widehat{\mathbf{x}} \in \mathbb{R}^d$, an estimator of $\bar{\mathbf{x}}$. We consider the severely communication-constrained setting where $nk \leq d$, when only a limited amount of information about the client vectors is seen by the server.

**Error Metric.** We measure the quality of the decoded vector $\widehat{\mathbf{x}}$ using the Mean Squared Error (MSE) $\mathbb{E}\left[\|\widehat{\mathbf{x}} - \bar{\mathbf{x}}\|_2^2\right]$, where the expectation is with respect to all the randomness in the encoding-decoding scheme. Our goal is to design an encoding-decoding algorithm to achieve an unbiased estimate $\widehat{\mathbf{x}}$ (i.e. $\mathbb{E}[\widehat{\mathbf{x}}] = \bar{\mathbf{x}}$) that minimizes the MSE, given the per-client communication budget $k$. To consider an example, in rand-$k$ sparsification, each client sends randomly selected $k$ out of its $d$ coordinates to the server. The server then computes the mean estimate as $\widehat{\mathbf{x}}^{(\text{Rand-}k)} = \frac{1}{n}\frac{d}{k}\sum_{i=1}^n \widehat{\mathbf{x}}_i$. By [12, Lemma 1], the MSE of Rand-$k$ sparsification is given by

$$\mathbb{E}\left[\|\widehat{\mathbf{x}}^{(\text{Rand-}k)} - \bar{\mathbf{x}}\|_2^2\right] = \frac{1}{n^2}\left(\frac{d}{k} - 1\right)\sum_{i=1}^n \|\mathbf{x}_i\|_2^2 \tag{1}$$

**The Rand-$k$-Spatial Family Estimator.** For large values of $\frac{d}{k}$, the Rand-$k$ MSE in Eq. 1 can be prohibitive. [12] proposed the Rand-$k$-Spatial family estimator, which achieves an improved MSE, by leveraging the knowledge of the correlation between client vectors at the server. The encoded vectors $\{\widehat{\mathbf{x}}_i\}$ are the same as in Rand-$k$. However, the $j$-th coordinate of the decoded vector is given as

$$\widehat{\mathbf{x}}^{(\text{Rand-}k\text{-Spatial})}(j) = \frac{1}{n}\frac{\bar{\beta}}{T(M_j)}\sum_{i=1}^n \widehat{\mathbf{x}}_i(j) \tag{2}$$

Here, $T : \mathbb{R} \to \mathbb{R}$ is a pre-defined transformation function of $M_j$, the number of clients which sent their $j$-th coordinate, and $\bar{\beta}$ is a normalization constant to ensure $\widehat{\mathbf{x}}$ is an unbiased estimator of $\mathbf{x}$. The resulting MSE is given by

$$\mathbb{E}\left[\|\widehat{\mathbf{x}}^{(\text{Rand-}k\text{-Spatial})} - \bar{\mathbf{x}}\|_2^2\right] = \frac{1}{n^2}\left(\frac{d}{k} - 1\right)\sum_{i=1}^n \|\mathbf{x}_i\|_2^2 + \left(c_1\sum_{i=1}^n \|\mathbf{x}_i\|_2^2 - c_2\sum_{i=1}^n\sum_{l\neq i}\langle\mathbf{x}_i, \mathbf{x}_l\rangle\right) \tag{3}$$

where $c_1, c_2$ are constants dependent on $n, d, k$ and $T$, but independent of client vectors $\{\mathbf{x}_i\}_{i=1}^n$. When the client vectors are orthogonal, i.e., $\langle\mathbf{x}_i, \mathbf{x}_l\rangle = 0$, for all $i \neq l$, [12] show that with appropriately chosen $T$, the MSE in Eq. 3 reduces to Eq. 1. However, if there exists a positive correlation between the vectors, the MSE in Eq. 3 is strictly smaller than that for Rand-$k$ Eq. 1.

## 4 The Rand-Proj-Spatial Family Estimator

While the Rand-$k$-Spatial family estimator proposed in [12] focuses only on improving the decoding at the server, we consider a more general encoding-decoding scheme. Rather than simply communicating $k$ out of the $d$ coordinates of its vector $\mathbf{x}_i$ to the server, client $i$ applies a (random) linear map $\boldsymbol{G}_i \in \mathbb{R}^{k \times d}$ to $\mathbf{x}_i$ and sends $\widehat{\mathbf{x}}_i = \boldsymbol{G}_i\mathbf{x}_i \in \mathbb{R}^k$ to the server. The decoding process on the server first projects the *encoded* vectors $\{\boldsymbol{G}_i\mathbf{x}_i\}_{i=1}^n$ back to the $d$-dimensional space and then forms an estimate $\widehat{\mathbf{x}}$. We motivate our new decoding procedure with the following regression problem:

$$\widehat{\mathbf{x}}^{(\text{Rand-Proj})} = \arg\min_{\mathbf{x}}\sum_{i=1}^n \|\boldsymbol{G}_i\mathbf{x} - \boldsymbol{G}_i\mathbf{x}_i\|_2^2 \tag{4}$$

To understand the motivation behind Eq. 4, first consider the special case where $\boldsymbol{G}_i = \boldsymbol{I}_d$ for all $i \in [n]$, that is, the clients communicate their vectors without compressing. The server can then exactly compute the mean $\bar{\mathbf{x}} = \frac{1}{n}\sum_{i=1}^n \mathbf{x}_i$. Equivalently, $\bar{\mathbf{x}}$ is the solution of $\arg\min_{\mathbf{x}}\sum_{i=1}^n \|\mathbf{x} - \mathbf{x}_i\|_2^2$. In the more general setting, we require that the mean estimate $\widehat{\mathbf{x}}$ when encoded using the map $\boldsymbol{G}_i$, should be "close" to the encoded vector $\boldsymbol{G}_i\mathbf{x}_i$ originally sent by client $i$, for all clients $i \in [n]$.

We note the above intuition can also be translated into different regression problems to motivate the design of the new decoding procedure. We discuss in Appendix B.2 intuitive alternatives which,

unfortunately, either do not enable the usage of cross-client correlation information, or do not use such information effectively. We choose the formulation in Eq. 4 due to its analytical tractability and its direct relevance to our target error metric MSE. We note that it is possible to consider the problem in Eq. 4 in the other norms, such as the sum of $\ell_2$ norms (without the squares) or the $\ell_\infty$ norm. We leave this as a future direction to explore.

The solution to Eq. 4 is given by $\widehat{\mathbf{x}}^{(\text{Rand-Proj})} = (\sum_{i=1}^n \boldsymbol{G}_i^T \boldsymbol{G}_i)^\dagger \sum_{i=1}^n \boldsymbol{G}_i^T \boldsymbol{G}_i \mathbf{x}_i$, where $\dagger$ denotes the Moore-Penrose pseudo inverse [49]. However, while $\widehat{\mathbf{x}}^{(\text{Rand-Proj})}$ minimizes the error of the regression problem, our goal is to design an *unbiased* estimator that also improves the MSE. Therefore, we make the following two modifications to $\widehat{\mathbf{x}}^{(\text{Rand-Proj})}$: First, to ensure that the mean estimate is unbiased, we scale the solution by a normalization factor $\bar{\beta}$[1]. Second, to incorporate varying degrees of correlation among the clients, we propose to apply a scalar transformation function $T : \mathbb{R} \to \mathbb{R}$ to each of the eigenvalues of $\sum_{i=1}^n \boldsymbol{G}_i^T \boldsymbol{G}_i$. The resulting Rand-Proj-Spatial family estimator is given by

$$\widehat{\mathbf{x}}^{(\text{Rand-Proj-Spatial})} = \bar{\beta} \Big( T(\sum_{i=1}^n \boldsymbol{G}_i^T \boldsymbol{G}_i) \Big)^\dagger \sum_{i=1}^n \boldsymbol{G}_i^T \boldsymbol{G}_i \mathbf{x}_i \tag{5}$$

Though applying the transformation function $T$ in Rand-Proj-Spatial requires computing the eigen-decomposition of $\sum_{i=1}^n \boldsymbol{G}_i^T \boldsymbol{G}_i$. However, this happens only at the server, which has more computational power than the clients. Next, we observe that for appropriate choice of $\{\boldsymbol{G}_i\}_{i=1}^n$, the Rand-Proj-Spatial family estimator reduces to the Rand-$k$-Spatial family estimator [12].

**Lemma 4.1** (Recovering Rand-$k$-Spatial). *Suppose client $i$ generates a subsampling matrix $\boldsymbol{E}_i = [\mathbf{e}_{i_1}, \quad \ldots, \quad \mathbf{e}_{i_k}]^\top$, where $\{\mathbf{e}_j\}_{j=1}^d$ are the canonical basis vectors, and $\{i_1, \ldots, i_k\}$ are sampled from $\{1, \ldots, d\}$ without replacement. The encoded vectors are given as $\widehat{\mathbf{x}}_i = \boldsymbol{E}_i \mathbf{x}_i$. Given a function $T$, $\widehat{\mathbf{x}}$ computed as in Eq. 5 recovers the Rand-$k$-Spatial estimator.*

The proof details are in Appendix C.5. We discuss the choice of $T$ and how it compares to Rand-$k$-Spatial in detail in Section 4.3.

**Remark 4.2.** *In the simple case when $\boldsymbol{G}_i$'s are subsampling matrices (as in Rand-$k$-Spatial [12]), the $j$-th diagonal entry of $\sum_{i=1}^n \boldsymbol{G}_i^T \boldsymbol{G}_i$, $M_j$ conveys the number of clients which sent the $j$-th coordinate. Rand-$k$-Spatial incorporates correlation among client vectors by applying a function $T$ to $M_j$. Intuitively, it means scaling different coordinates differently. This is in contrast to Rand-$k$, which scales all the coordinates by $d/k$. In our more general case, we apply a function $T$ to the eigenvalues of $\sum_{i=1}^n \boldsymbol{G}_i^T \boldsymbol{G}_i$ to similarly incorporate correlation in Rand-Proj-Spatial.*

To showcase the utility of the Rand-Proj-Spatial family estimator, we propose to set the random linear maps $\boldsymbol{G}_i$ to be scaled Subsampled Randomized Hadamard Transform (SRHT, e.g. [38]). Assuming $d$ to be a power of 2, the linear map $\boldsymbol{G}_i$ is given as

$$\boldsymbol{G}_i = \frac{1}{\sqrt{d}} \boldsymbol{E}_i \boldsymbol{H} \boldsymbol{D}_i \in \mathbb{R}^{k \times d} \tag{6}$$

where $\boldsymbol{E}_i \in \mathbb{R}^{k \times d}$ is the subsampling matrix, $\boldsymbol{H} \in \mathbb{R}^{d \times d}$ is the (deterministic) Hadamard matrix and $\boldsymbol{D}_i \in \mathbb{R}^{d \times d}$ is a diagonal matrix with independent Rademacher random variables as its diagonal entries. We choose SRHT due to its superior performance compared to other random matrices. Other possible choices of random matrices for Rand-Proj-Spatial estimator include sketching matrices commonly used for dimensionality reduction, such as Gaussian [50, 51], row-normalized Gaussian, and Count Sketch [52], as well as error-correction coding matrices, such as Low-Density Parity Check (LDPC) [53] and Fountain Codes [54]. However, in the absence of correlation between client vectors, all these matrices suffer a higher MSE.

In the following, we first compare the MSE of Rand-Proj-Spatial with SRHT against Rand-$k$ and Rand-$k$-Spatial in two extreme cases: when all the client vectors are identical, and when all the client vectors are orthogonal to each other. In both cases, we highlight the transformation function $T$ used in Rand-Proj-Spatial (Eq. 5) to incorporate the knowledge of cross-client correlation. We define

$$\mathcal{R} := \frac{\sum_{i=1}^n \sum_{l \neq i} \langle \mathbf{x}_i, \mathbf{x}_l \rangle}{\sum_{i=1}^n \|\mathbf{x}_i\|_2^2} \tag{7}$$

to measure the correlation between the client vectors. Note that $\mathcal{R} \in [-1, n-1]$. $\mathcal{R} = 0$ implies all client vectors are orthogonal, while $\mathcal{R} = n-1$ implies identical client vectors.

---

[1] We show that it suffices for $\bar{\beta}$ to be a scalar in Appendix B.1.

## 4.1 Case I: Identical Client Vectors ($\mathcal{R} = n - 1$)

When all the client vectors are identical ($\mathbf{x}_i \equiv \mathbf{x}$), [12] showed that setting the transformation $T$ to identity, i.e., $T(m) = m$, for all $m$, leads to the minimum MSE in the Rand-$k$-Spatial family of estimators. The resulting estimator is called Rand-$k$-Spatial (Max). Under the same setting, using the same transformation $T$ in Rand-Proj-Spatial with SRHT, the decoded vector in Eq. 5 simplifies to

$$\widehat{\mathbf{x}}^{(\text{Rand-Proj-Spatial})} = \bar{\beta}\Big(\sum_{i=1}^{n} \boldsymbol{G}_i^T \boldsymbol{G}_i\Big)^{\dagger} \sum_{i=1}^{n} \boldsymbol{G}_i^T \boldsymbol{G}_i \mathbf{x} = \bar{\beta}\boldsymbol{S}^{\dagger}\boldsymbol{S}\mathbf{x}, \tag{8}$$

where $\boldsymbol{S} := \sum_{i=1}^{n} \boldsymbol{G}_i^T \boldsymbol{G}_i$. By construction, $\text{rank}(\boldsymbol{S}) \leq nk$, and we focus on the case $nk \leq d$.

**Limitation of Subsampling matrices.** As mentioned above, with $\boldsymbol{G}_i = \boldsymbol{E}_i, \forall\, i \in [n]$, we recover the Rand-$k$-Spatial family of estimators. In this case, $\boldsymbol{S}$ is a diagonal matrix, where each diagonal entry $\boldsymbol{S}_{jj} = M_j, j \in [d]$. $M_j$ is the number of clients which sent their $j$-th coordinate to the server. To ensure $\text{rank}(\boldsymbol{S}) = nk$, we need $\boldsymbol{S}_{jj} \leq 1, \forall j$, i.e., each of the $d$ coordinates is sent by *at most* one client. If all the clients sample their matrices $\{\boldsymbol{E}_i\}_{i=1}^{n}$ independently, this happens with probability $\frac{\binom{d}{nk}}{\binom{d}{k}^n}$. As an example, for $k = 1$, $\text{Prob}(\text{rank}(\boldsymbol{S}) = n) = \frac{\binom{d}{n}}{d^n} \leq \frac{1}{n!}$ (because $\frac{d^n}{n^n} \leq \binom{d}{n} \leq \frac{d^n}{n!}$). Therefore, to guarantee that $\boldsymbol{S}$ is full-rank, each client would need the subsampling information of all the other clients. This not only requires additional communication but also has serious privacy implications. Essentially, the limitation with subsampling matrices $\boldsymbol{E}_i$ is that the eigenvectors of $\boldsymbol{S}$ are restricted to be canonical basis vectors $\{\mathbf{e}_j\}_{j=1}^{d}$. Generalizing $\boldsymbol{G}_i$'s to general rank $k$ matrices relaxes this constraint and hence we can ensure that $\boldsymbol{S}$ is full-rank with high probability. In the next result, we show the benefit of choosing $\boldsymbol{G}_i$ as SRHT matrices. We call the resulting estimator Rand-Proj-Spatial(Max).

**Theorem 4.3** (MSE under Full Correlation). *Consider $n$ clients, each holding the same vector $\mathbf{x} \in \mathbb{R}^d$. Suppose we set $T(\lambda) = \lambda$, $\bar{\beta} = \frac{d}{k}$ in Eq. 5, and the random linear map $\boldsymbol{G}_i$ at each client to be an SRHT matrix. Let $\delta$ be the probability that $\boldsymbol{S} = \sum_{i=1}^{n} \boldsymbol{G}_i^T \boldsymbol{G}_i$ does not have full rank. Then, for $nk \leq d$,*

$$\mathbb{E}\Big[\|\widehat{\mathbf{x}}^{(\text{Rand-Proj-Spatial(Max)})} - \bar{\mathbf{x}}\|_2^2\Big] \leq \Big[\frac{d}{(1-\delta)nk + \delta k} - 1\Big]\|\mathbf{x}\|_2^2 \tag{9}$$

The proof details are in Appendix C.1. To compare the performance of Rand-Proj-Spatial(Max) against Rand-$k$, we show in Appendix C.2 that for $n \geq 2$, as long as $\delta \leq \frac{2}{3}$, the MSE of Rand-Proj-Spatial(Max) is less than that of Rand-$k$. Furthermore, in Appendix C.3 we empirically demonstrate that with $d \in \{32, 64, 128, \ldots, 1024\}$ and different values of $nk \leq d$, the rank of $\boldsymbol{S}$ is full with high probability, i.e., $\delta \approx 0$. This implies $\mathbb{E}[\|\widehat{\mathbf{x}}^{(\text{Rand-Proj-Spatial(Max)})} - \bar{\mathbf{x}}\|_2^2] \approx (\frac{d}{nk} - 1)\|\mathbf{x}\|_2^2$.

Futhermore, since setting $\boldsymbol{G}_i$ as SRHT significantly increases the probability of recovering $nk$ coordinates of $\mathbf{x}$, the MSE of Rand-Proj-Spatial with SRHT (Eq. 4.3) is strictly less than that of Rand-$k$-Spatial (Eq. 3). We also compare the MSEs of the three estimators in Figure 2 in the following setting: $\|\mathbf{x}\|_2 = 1$, $d = 1024$, $n \in \{10, 20, 50, 100\}$ and small $k$ values such that $nk < d$.

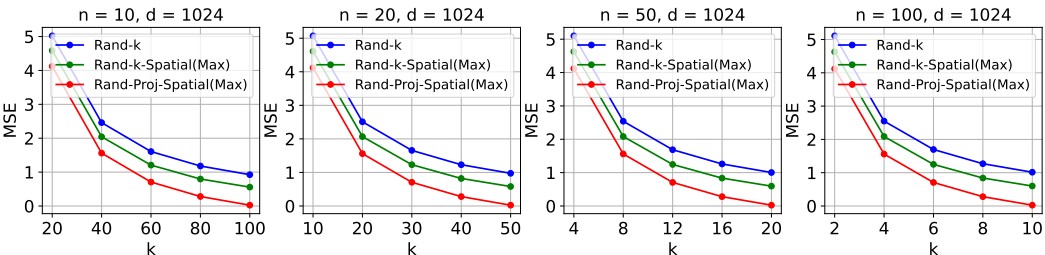

Figure 2: MSE comparison of Rand-$k$, Rand-$k$-Spatial(Max) and Rand-Proj-Spatial(Max) estimators, when all clients have identical vectors (maximum inter-client correlation).

## 4.2 Case II: Orthogonal Client Vectors ($\mathcal{R} = 0$)

When all the client vectors are orthogonal to each other, [12] showed that Rand-$k$ has the lowest MSE among the Rand-$k$-Spatial family of decoders. We show in the next result that if we set the

random linear maps $\boldsymbol{G}_i$ at client $i$ to be SRHT, and choose the fixed transformation $T \equiv 1$ as in [12], Rand-Proj-Spatial achieves the same MSE as that of Rand-$k$.

**Theorem 4.4** (MSE under No Correlation). *Consider $n$ clients, each holding a vector $\mathbf{x}_i \in \mathbb{R}^d$, $\forall i \in [n]$. Suppose we set $T \equiv 1$, $\bar{\beta} = \frac{d^2}{k}$ in Eq. 5, and the random linear map $\boldsymbol{G}_i$ at each client to be an SRHT matrix. Then, for $nk \leq d$,*

$$\mathbb{E}\Big[\|\widehat{\mathbf{x}}^{(\text{Rand-Proj-Spatial})} - \bar{\mathbf{x}}\|_2^2\Big] = \frac{1}{n^2}\Big(\frac{d}{k} - 1\Big)\sum_{i=1}^{n}\|\mathbf{x}_i\|_2^2. \tag{10}$$

The proof details are in Appendix C.4. Theorem 4.4 above shows that with zero correlation among client vectors, Rand-Proj-Spatial achieves the same MSE as that of Rand-$k$.

### 4.3 Incorporating Varying Degrees of Correlation

In practice, it unlikely that all the client vectors are either identical or orthogonal to each other. In general, there is some "imperfect" correlation among the client vectors, i.e., $\mathcal{R} \in (0, n-1)$. Given correlation level $\mathcal{R}$, [12] shows that the estimator from the Rand-$k$-Spatial family that minimizes the MSE is given by the following transformation.

$$T(m) = 1 + \frac{\mathcal{R}}{n-1}(m-1) \tag{11}$$

Recall from Section 4.1 (Section 4.2) that setting $T(m) = 1$ ($T(m) = m$) leads to the estimator among the Rand-$k$-Spatial family that minimizes MSE when there is zero (maximum) correlation among the client vectors. We observe the function $T$ defined in Eq. 11 essentially interpolates between the two extreme cases, using the normalized degree of correlation $\frac{\mathcal{R}}{n-1} \in [-\frac{1}{n-1}, 1]$ as the weight. This motivates us to apply the same function $T$ defined in Eq. 11 on the eigenvalues of $\boldsymbol{S} = \sum_{i=1}^{n} \boldsymbol{G}_i^T \boldsymbol{G}_i$ in Rand-Proj-Spatial. As we shall see in our results, the resulting Rand-Proj-Spatial family estimator improves over the MSE of both Rand-$k$ and Rand-$k$-Spatial family estimator.

We note that deriving a closed-form expression of MSE for Rand-Proj-Spatial with SRHT in the general case with the transformation function $T$ (Eq. 11) is hard (we elaborate on this in Appendix B.3), as this requires a closed form expression for the non-asymptotic distributions of eigenvalues and eigenvectors of the random matrix $\boldsymbol{S}$. To the best of our knowledge, previous analyses of SRHT, for example in [37, 38, 39, 45, 55], rely on the asymptotic properties of SRHT, such as the limiting eigen spectrum, or concentration bounds on the singular values, to derive asymptotic or approximate guarantees. However, to analyze the MSE of Rand-Proj-Spatial, we need an exact, non-asymptotic analysis of the eigenvalues and eigenvectors distribution of SRHT. Given the apparent intractability of the theoretical analysis, we compare the MSE of Rand-Proj-Spatial, Rand-$k$-Spatial, and Rand-$k$ via simulations.

**Simulations.** In each experiment, we first simulate $\bar{\beta}$ in Eq. 5, which ensures our estimator is unbiased, based on 1000 random runs. Given the degree of correlation $\mathcal{R}$, we then compute the squared error, i.e. $\|\widehat{\mathbf{x}}^{(\text{Rand-Proj-Spatial})} - \bar{\mathbf{x}}\|_2^2$, where Rand-Proj-Spatial has $\boldsymbol{G}_i$ as SRHT matrix (Eq. 6) and $T$ as in Eq. 11. We plot the average over 1000 random runs as an approximation to MSE. Each client holds a $d$-dimensional base vector $\mathbf{e}_j$ for some $j \in [d]$, and so so two clients either hold the same or orthogonal vectors. We control the degree of correlation $\mathcal{R}$ by changing the number of clients which hold the same vector. We consider $d = 1024$, $n \in \{21, 51\}$. We consider positive correlation values, where $\mathcal{R}$ is chosen to be linearly spaced within $[0, n-1]$. Hence, for $n = 21$, we use $\mathcal{R} \in \{4, 8, 12, 16\}$ and for $n = 51$, we use $\mathcal{R} \in \{10, 20, 30, 40\}$. All results are presented in Figure 3. As expected, given $\mathcal{R}$, Rand-Proj-Spatial consistently achieves a lower MSE than the lowest possible MSE from the Rand-$k$-Spatial family decoder. Additional results with different values of $n, d, k$, including the setting $nk \ll d$, can be found in Appendix B.4.

**A Practical Configuration.** In reality, it is hard to know the correlation information $\mathcal{R}$ among the client vectors. [12] uses the transformation function which interpolates to the middle point between the full correlation and no correlation cases, such that $T(m) = 1 + \frac{n}{2}\frac{m-1}{n-1}$. Rand-$k$-Spatial with such $T$ is called Rand-$k$-Spatial(Avg). Following this approach, we evaluate Rand-Proj-Spatial with SRHT using this $T$, and call it Rand-Proj-Spatial(Avg) in practical settings (see Figure 4).

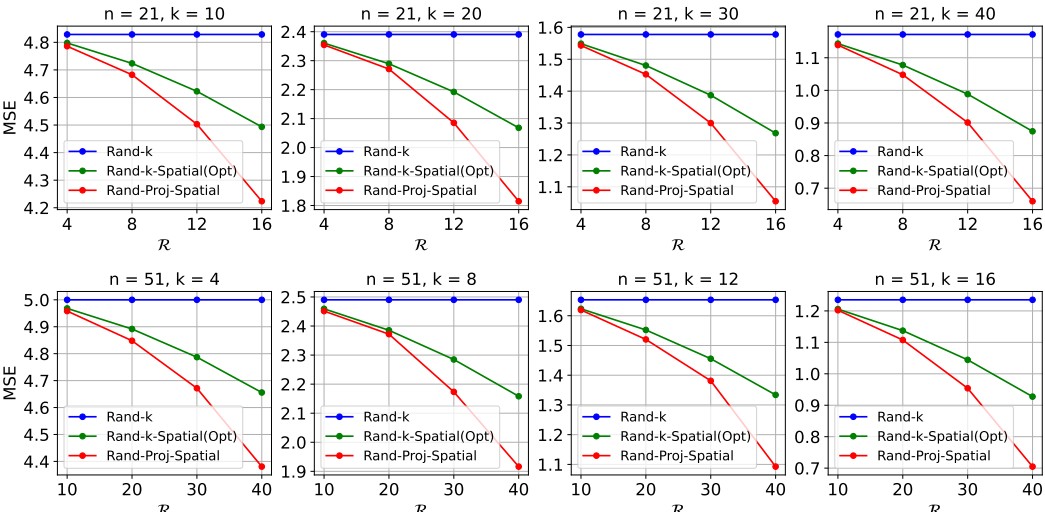

Figure 3: MSE comparison of estimators Rand-$k$, Rand-$k$-Spatial(Opt), Rand-Proj-Spatial, given the degree of correlation $\mathcal{R}$. Rand-$k$-Spatial(Opt) denotes the estimator that gives the lowest possible MSE from the Rand-$k$-Spatial family. We consider $d = 1024$, number of clients $n \in \{21, 51\}$, and $k$ values such that $nk < d$. In each plot, we fix $n, k, d$ and vary the degree of positive correlation $\mathcal{R}$. The y-axis represents MSE. Notice since each client has a fixed $\|\mathbf{x}_i\|_2 = 1$, and Rand-$k$ does not leverage cross-client correlation, the MSE of Rand-$k$ in each plot remains the same for different $\mathcal{R}$.

## 5   Experiments

We consider three practical distributed optimization tasks for evaluation: distributed power iteration, distributed $k$-means and distributed linear regression. We compare Rand-Proj-Spatial(Avg) against Rand-$k$, Rand-$k$-Spatial(Avg), and two more sophisticated but widely used sparsification schemes: non-uniform coordinate-wise gradient sparsification [15] (we call it Rand-$k$(Wangni)) and the Induced compressor with Rand-$k$ + Top-$k$ [14]. The results are presented in Figure 4.

**Dataset.**   For both distributed power iteration and distributed $k$-means, we use the test set of the `Fashion-MNIST` dataset [56] consisting of 10000 samples. The original images from `Fashion-MNIST` are $28 \times 28$ in size. We preprocess and resize each image to be $32 \times 32$. Resizing images to have their dimension as a power of 2 is a common technique used in computer vision to accelerate the convolution operation. We use the `UJIndoor` dataset [2] for distributed linear regression. We subsample 10000 data points, and use the first 512 out of the total 520 features on signals of phone calls. The task is to predict the longitude of the location of a phone call. In all the experiments in Figure 4, the datasets are split IID across the clients via random shuffling. In Appendix D.1, we have additional results for non-IID data split across the clients.

**Setup and Metric.**   Recall that $n$ denotes the number of clients, $k$ the per-client communication budget, and $d$ the vector dimension. For Rand-Proj-Spatial, we use the first 50 iterations to estimate $\bar{\beta}$ (see Eq. 5). Note that $\bar{\beta}$ only depends on $n, k, d$, and $T$ (the transformation function in Eq. 5), but is independent of the dataset. We repeat the experiments across 10 independent runs, and report the mean MSE (solid lines) and one standard deviation (shaded regions) for each estimator. For each task, we plot the squared error of the mean estimator $\widehat{\mathbf{x}}$, i.e., $\|\widehat{\mathbf{x}} - \bar{\mathbf{x}}\|_2^2$, and the values of the task-specific loss function, detailed below.

**Tasks and Settings:**

**1. Distributed power iteration.** We estimate the principle eigenvector of the covariance matrix, with the dataset (`Fashion-MNIST`) distributed across the $n$ clients. In each iteration, each client computes a local principle eigenvector estimate based on a single power iteration and sends an encoded version to the server. The server then computes a global estimate and sends it back to the clients. The task-specific loss here is $\|\mathbf{v}_t - \mathbf{v}_{top}\|_2$, where $\mathbf{v}_t$ is the global estimate of the principal eigenvector at iteration $t$, and $\mathbf{v}_{top}$ is the true principle eigenvector.

---

[2]`https://archive.ics.uci.edu/ml/datasets/ujiindoorloc`

**2. Distributed $k$-means.** We perform $k$-means clustering [57] with the data distributed across $n$ clients (`Fashion-MNIST`, 10 classes) using Lloyd's algorithm. At each iteration, each client performs a single iteration of $k$-means to find its local centroids and sends the encoded version to the server. The server then computes an estimate of the global centroids and sends them back to the clients. We report the average squared mean estimation error across 10 clusters, and the $k$-means loss, i.e., the sum of the squared distances of the data points to the centroids.

For both distributed power iterations and distributed $k$-means, we run the experiments for 30 iterations and consider two different settings: $n = 10, k = 102$ and $n = 50, k = 20$.

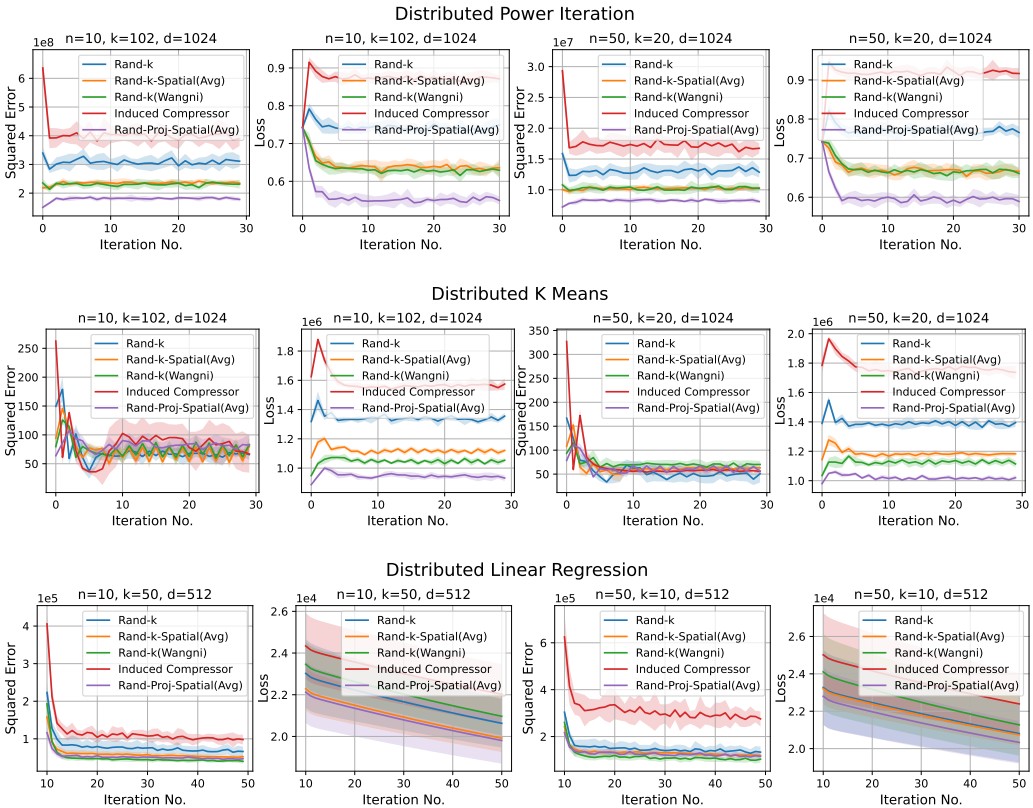

Figure 4: Experiment results on three distributed optimization tasks: distributed power iteration, distributed $k$-means, and distributed linear regression. The first two use the `Fashion-MNIST` dataset with the images resized to $32 \times 32$, hence $d = 1024$. Distributed linear regression uses `UJIndoor` dataset with $d = 512$. All the experiments are repeated for 10 random runs, and we report the mean as the solid lines, and one standard deviation using the shaded region. The violet line in the plots represents our proposed Rand-Proj-Spatial(Avg) estimator.

**3. Distributed linear regression.** We perform linear regression on the `UJIndoor` dataset distributed across $n$ clients using SGD. At each iteration, each client computes a local gradient and sends an encoded version to the server. The server computes a global estimate of the gradient, performs an SGD step, and sends the updated parameter to the clients. We run the experiments for 50 iterations with learning rate 0.001. The task-specific loss is the linear regression loss, i.e. empirical mean squared error. To have a proper scale that better showcases the difference in performance of different estimators, we plot the results starting from the 10th iteration.

**Results.** It is evident from Figure 4 that Rand-Proj-Spatial(Avg), our estimator with the practical configuration $T$ (see Section 4.3) that does not require the knowledge of the actual degree of correlation among clients, consistently outperforms the other estimators in all three tasks. Additional experiments for the three tasks are included in Appendix D.1. Furthermore, we present the wall-clock time to encode and decode client vectors using different sparsification schemes in Figure 5. Though Rand-Proj-Spatial(Avg) has the longest decoding time, the encoding time of Rand-Proj-Spatial(Avg) is less than that of the *adaptive* Rand-$k$(Wangni) sparsifier. In practice, the server has more compu-

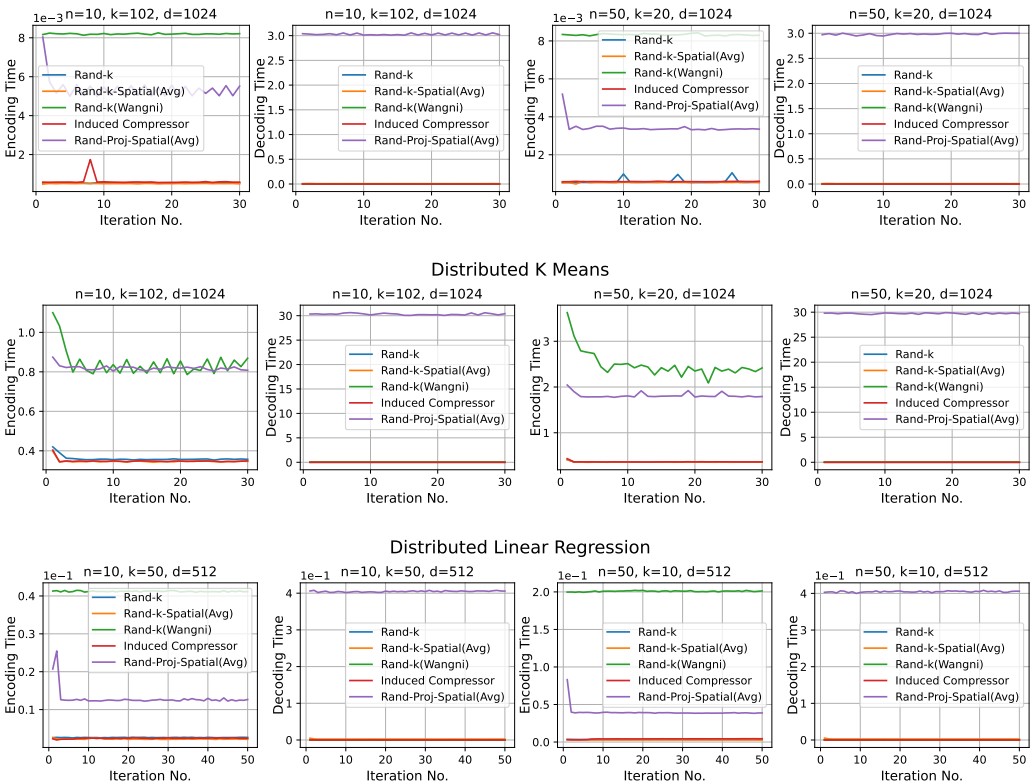

Figure 5: The corresponding wall-clock time to encode and decode client vectors (in seconds) using different sparsification schemes, across the three tasks.

tational power than the clients and hence can afford a longer decoding time. Therefore, it is more important to have efficient encoding procedures.

## 6 Limitations

We note two practical limitations of the proposed Rand-Proj-Spatial.
**1) Computation Time of Rand-Proj-Spatial.** The encoding time of Rand-Proj-Spatial is $O(kd)$, while the decoding time is $O(d^2 \cdot nk)$. The computation bottleneck in decoding is computing the eigendecomposition of the $d \times d$ matrix $\sum_{i=1}^{n} \boldsymbol{G}_i^T \boldsymbol{G}_i$ of rank at most $nk$. Improving the computation time for both the encoding and decoding schemes is an important direction for future work.
**2) Perfect Shared Randomness.** It is common to assume perfect shared randomness between the server and the clients in distributed settings [58]. However, to perfectly simulate randomness using Pseudo Random Number Generator (PRNG), at least $\log_2 d$ bits of the seed need to be exchanged in practice. We acknowledge this gap between theory and practice.

## 7 Conclusion

In this paper, we propose the Rand-Proj-Spatial estimator, a novel encoding-decoding scheme, for communication-efficient distributed mean estimation. The proposed client-side encoding generalizes and improves the commonly used Rand-$k$ sparsification, by utilizing projections onto general $k$-dimensional subspaces. On the server side, cross-client correlation is leveraged to improve the approximation error. Compared to existing methods, the proposed scheme consistently achieves better mean estimation error across a variety of tasks. Potential future directions include improving the computation time of Rand-Proj-Spatial and exploring whether the proposed Rand-Proj-Spatial achieves the optimal estimation error among the class of *non-adaptive* estimators, given correlation information. Furthermore, combining sparsification and quantization techniques and deriving such algorithms with the optimal communication cost-estimation error trade-offs would be interesting.

## Acknowledgments

We would like to thank the anonymous reviewer for providing valuable feedback on the title of this work, interesting open problems, alternative motivating regression problems and practical limitations of shared randomness. This work was supported in part by NSF grants CCF 2045694, CCF 2107085, CNS-2112471, and ONR N00014-23-1- 2149.

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

# Appendices

## A  Additional Details on Motivation in Introduction

### A.1  Preprocssing all client vectors by the same random matrix does not improve performance

Consider $n$ clients. Suppose client $i$ holds a vector $\mathbf{x}_i \in \mathbb{R}^d$. We want to apply Rand-$k$ or Rand-$k$-Spatial, while also making the encoding process more flexible than just randomly choosing $k$ out of $d$ coordinates. One naïve way of doing this is for each client to pre-process its vector by applying an orthogonal matrix $\boldsymbol{G} \in \mathbb{R}^{d \times d}$ that is the *same* across all clients. Such a technique might be helpful in improving the performance of quantization because the MSE due to quantization often depends on how uniform the coordinates of $\mathbf{x}_i$'s are, i.e. whether the coordinates of $\mathbf{x}_i$ have values close to each other. $\boldsymbol{G}$ is designed to be the random matrix (e.g. SRHT) that rotates $\mathbf{x}_i$ and makes its coordinates uniform.

Each client sends the server $\widehat{\mathbf{x}}_i = \boldsymbol{E}_i \boldsymbol{G} \mathbf{x}_i$, where $\boldsymbol{E}_i \in \mathbb{R}^{k \times d}$ is the subsamaping matrix. If we use Rand-$k$, the server can decode each client vector by first applying the decoding procedure of Rand-$k$ and then rotating it back to the original space, i.e., $\widehat{\mathbf{x}}_i^{(\text{Naïve})} = \frac{d}{k} \boldsymbol{G}^T \boldsymbol{E}_i^T \boldsymbol{E}_i \boldsymbol{G} \mathbf{x}_i$. Note that

$$
\mathbb{E}[\widehat{\mathbf{x}}_i^{(\text{Naïve})}] = \frac{d}{k} \mathbb{E}[\boldsymbol{G}^T \boldsymbol{E}_i^T \boldsymbol{E}_i \boldsymbol{G} \mathbf{x}_i]
$$
$$
= \frac{d}{k} \boldsymbol{G}^T \frac{k}{d} \boldsymbol{I}_d \boldsymbol{G} \mathbf{x}_i
$$
$$
= \mathbf{x}_i.
$$

Hence, $\widehat{\mathbf{x}}_i^{(\text{Naïve})}$ is unbiased. The MSE of $\widehat{\mathbf{x}}^{(\text{Naïve})} = \frac{1}{n} \sum_{i=1}^{n} \widehat{\mathbf{x}}_i^{(\text{Naïve})}$ is given as

$$
\mathbb{E} \left\| \bar{\mathbf{x}} - \widehat{\mathbf{x}}^{(\text{Naïve})} \right\|_2^2 = \mathbb{E} \left\| \frac{1}{n} \sum_{i=1}^{n} \mathbf{x}_i - \frac{1}{n} \frac{d}{k} \sum_{i=1}^{n} \boldsymbol{G}^T \boldsymbol{E}_i^T \boldsymbol{E}_i \boldsymbol{G} \mathbf{x}_i \right\|_2^2
$$
$$
= \frac{1}{n^2} \mathbb{E} \left\| \sum_{i=1}^{n} \mathbf{x}_i - \frac{d}{k} \sum_{i=1}^{n} \boldsymbol{G}^T \boldsymbol{E}_i^T \boldsymbol{E}_i \boldsymbol{G} \mathbf{x}_i \right\|_2^2
$$
$$
= \frac{1}{n^2} \left\{ \frac{d^2}{k^2} \mathbb{E} \left\| \sum_{i=1}^{n} \boldsymbol{G}^T \boldsymbol{E}_i^T \boldsymbol{E}_i \boldsymbol{G} \mathbf{x}_i \right\|^2 - \left\| \sum_{i=1}^{n} \mathbf{x}_i \right\|^2 \right\}
$$
$$
= \frac{1}{n^2} \left\{ \frac{d^2}{k^2} \left( \sum_{i=1}^{n} \mathbb{E} \| \boldsymbol{G}^T \boldsymbol{E}_i^T \boldsymbol{E}_i \boldsymbol{G} \mathbf{x}_i \|_2^2 + \sum_{i \neq j} \mathbb{E} \left\langle \boldsymbol{G}^T \boldsymbol{E}_i^T \boldsymbol{E}_i \boldsymbol{G} \mathbf{x}_i, \boldsymbol{G} \boldsymbol{E}_l^T \boldsymbol{E}_l \boldsymbol{G} \mathbf{x}_l \right\rangle \right) - \left\| \sum_{i=1}^{n} \mathbf{x}_i \right\|^2 \right\}.
$$
$$\tag{12}$$

Next, we bound the first term in Eq. 12.

$$
\mathbb{E} \| \boldsymbol{G}^T \boldsymbol{E}_i^T \boldsymbol{E}_i \boldsymbol{G} \mathbf{x}_i \|_2^2 = \mathbb{E}[\mathbf{x}_i^T \boldsymbol{G}^T \boldsymbol{E}_i^T \boldsymbol{E}_i \boldsymbol{G} \boldsymbol{G}^T \boldsymbol{E}_i^T \boldsymbol{E}_i \boldsymbol{G} \mathbf{x}_i] = \mathbb{E}[\mathbf{x}_i^T \boldsymbol{G}^T \boldsymbol{E}_i^T \boldsymbol{E}_i \boldsymbol{E}_i^T \boldsymbol{E}_i \boldsymbol{G} \mathbf{x}_i]
$$
$$
= \mathbf{x}_i^T \boldsymbol{G}^T \mathbb{E}[(\boldsymbol{E}_i^T \boldsymbol{E}_i)^2] \boldsymbol{G} \mathbf{x}_i
$$
$$
= \mathbf{x}_i^T \frac{k}{d} \boldsymbol{I}_d \mathbf{x}_i \qquad\qquad (\because (\boldsymbol{E}_i^T \boldsymbol{E}_i)^2 = \boldsymbol{E}_i^T \boldsymbol{E}_i)
$$
$$
= \frac{k}{d} \| \mathbf{x}_i \|_2^2 \tag{13}
$$

The second term in Eq. 12 can also be simplified as follows.

$$
\mathbb{E}[\langle \boldsymbol{G}^T \boldsymbol{E}_i^T \boldsymbol{E}_i \boldsymbol{G} \mathbf{x}_i, \boldsymbol{G}^T \boldsymbol{E}_l^T \boldsymbol{E}_l \boldsymbol{G} \mathbf{x}_l \rangle]
$$
$$
= \langle \boldsymbol{G}^T \mathbb{E}[\boldsymbol{E}_i^T \boldsymbol{E}_i] \boldsymbol{G} \mathbf{x}_i, \boldsymbol{G}^T \mathbb{E}[\boldsymbol{E}_l^T \boldsymbol{E}_l] \boldsymbol{G} \mathbf{x}_l \rangle
$$
$$
= \langle \boldsymbol{G}^T \frac{k}{d} \boldsymbol{I}_d \boldsymbol{G} \mathbf{x}_i, \boldsymbol{G}^T \frac{k}{d} \boldsymbol{I}_d \boldsymbol{G} \mathbf{x}_l \rangle
$$

$$= \frac{k^2}{d^2}\langle \mathbf{x}_i, \mathbf{x}_l \rangle. \tag{14}$$

Plugging Eq. 13 and Eq. 14 into Eq. 12, we get the MSE is

$$\mathbb{E}\|\bar{\mathbf{x}} - \hat{\mathbf{x}}^{(\text{Naïve})}\|_2^2$$

$$= \frac{1}{n^2}\Big\{ \frac{d^2}{k^2}\Big( \sum_{i=1}^{n}\frac{k}{d}\|\mathbf{x}_i\|_2^2 + 2\sum_{i=1}^{n}\sum_{l=i+1}^{n}\frac{k^2}{d^2}\langle \mathbf{x}_i, \mathbf{x}_l\rangle \Big) - \Big\|\sum_{i=1}^{n}\mathbf{x}_i\Big\|^2 \Big\}$$

$$= \frac{1}{n^2}\Big(\frac{d}{k} - 1\Big)\sum_{i=1}^{n}\|\mathbf{x}_i\|_2^2,$$

which has exactly the same MSE as that of Rand-$k$. The problem is that if each client applies the same rotational matrix $\boldsymbol{G}$, simply rotating the vectors will not change the $\ell_2$ norm of the decoded vector, and hence the MSE. Similarly, if one applies Rand-$k$-Spatial, one ends up having exactly the same MSE as that of Rand-$k$-Spatial as well. Hence, we need to design a new decoding procedure when the encoding procedure at the clients are more flexible.

## A.2   $nk \gg d$ is not interesting

One can rewrite $\sum_{i=1}^{n}\boldsymbol{G}_i^T\boldsymbol{G}_i$ in the Rand-Proj-Spatial estimator (Eq. 5) as $\sum_{i=1}^{n}\boldsymbol{G}_i^T\boldsymbol{G}_i = \sum_{j=1}^{nk}\mathbf{g}_j\mathbf{g}_j^T$, where $\mathbf{g}_j \in \mathbb{R}^d$ and $\mathbf{g}_{ik}, \mathbf{g}_{ik+1}, \ldots, \mathbf{g}_{(i+1)k}$ are the rows of $\boldsymbol{G}_i$. Since when $nk \gg d$, $\sum_{j=1}^{nk}\mathbf{g}_j\mathbf{g}_j^T \to \mathbb{E}[\sum_{j=1}^{n}\mathbf{g}_j\mathbf{g}_j^T]$ due to Law of Large Numbers, one way to see the limiting MSE of Rand-Proj-Spatial when $nk$ is large is to approximate $\sum_{i=1}^{n}\sum_{j=1}^{nk}\mathbf{g}_i\mathbf{g}_i^T$ by its expectation.

By Lemma 4.1, when $\boldsymbol{G}_i = \boldsymbol{E}_i$, Rand-Proj-Spatial recovers Rand-$k$-Spatial. We now discuss the limiting behavior of Rand-$k$-Spatial when $nk \gg d$ by leveraging our proposed Rand-Proj-Spatial. In this case, each $\mathbf{g}_j$ can be viewed as a random based vector $\mathbf{e}_w$ for $w$ randomly chosen in $[d]$. $\sum_{i=1}^{nk}\mathbf{g}_j\mathbf{g}_j^T \to \mathbb{E}[\sum_{i=1}^{nk}\mathbf{g}_j\mathbf{g}_j^T] = \sum_{i=1}^{nk}\frac{1}{d}\boldsymbol{I}_d = \frac{nk}{d}\boldsymbol{I}_d$. And so the scalar $\bar{\beta}$ in Eq. 5 to ensure an unbiased estimator is computed as

$$\bar{\beta}\mathbb{E}[(\frac{nk}{d}\boldsymbol{I}_d)^{\dagger}\boldsymbol{G}_i^T\boldsymbol{G}_i] = \boldsymbol{I}_d$$

$$\bar{\beta}\frac{d}{nk}\boldsymbol{I}_d\mathbb{E}[\boldsymbol{G}_i^T\boldsymbol{G}_i] = \boldsymbol{I}_d$$

$$\bar{\beta}\frac{d}{nk}\frac{k}{d} = \boldsymbol{I}_d$$

$$\bar{\beta} = n$$

And the MSE is now

$$\mathbb{E}\Big[\|\bar{\mathbf{x}} - \hat{\mathbf{x}}\|\Big] = \mathbb{E}\Big[\|\frac{1}{n}\sum_{i=1}^{n}\mathbf{x}_i - \frac{1}{n}\bar{\beta}\frac{d}{nk}\boldsymbol{I}_d\sum_{i=1}^{n}\boldsymbol{E}_i^T\boldsymbol{E}_i\mathbf{x}_i\|_2^2\Big]$$

$$= \frac{1}{n^2}\Big\{ \bar{\beta}^2\frac{d^2}{n^2k^2}\mathbb{E}\Big[\|\sum_{i=1}^{n}\boldsymbol{E}_i^T\boldsymbol{E}_i\mathbf{x}_i\|_2^2\Big] - \|\sum_{i=1}^{n}\mathbf{x}_i\|_2^2 \Big\}$$

$$= \frac{1}{n^2}\Big\{ n^2\frac{d^2}{n^2k^2}\Big( \sum_{i=1}^{n}\mathbb{E}\Big[\|\boldsymbol{E}_i^T\boldsymbol{E}_i\mathbf{x}_i\|_2^2\Big] + 2\sum_{i=1}^{n}\sum_{l=i+1}^{n}\langle \boldsymbol{E}_i^T\boldsymbol{E}_i\mathbf{x}_i, \boldsymbol{E}_l^T\boldsymbol{E}_l\mathbf{x}_l\rangle\Big) - \|\sum_{i=1}^{n}\mathbf{x}_i\|_2^2 \Big\}$$

$$= \frac{1}{n^2}\Big\{ \frac{d^2}{k^2}\Big( \sum_{i=1}^{n}\mathbb{E}\Big[\mathbf{x}_i^T(\boldsymbol{E}_i^T\boldsymbol{E}_i)^2\mathbf{x}_i\Big] + 2\sum_{i=1}^{n}\sum_{l=i+1}^{n}\frac{k^2}{d^2}\langle \mathbf{x}_i, \mathbf{x}_l\rangle\Big) - \|\sum_{i=1}^{n}\mathbf{x}_i\|_2^2 \Big\}$$

$$= \frac{1}{n^2}\Big\{ \frac{d^2}{k^2}\Big( \sum_{i=1}^{n}\frac{k}{d}\|\mathbf{x}_i\|_2^2 + 2\sum_{i=1}^{n}\sum_{l=i+1}^{n}\frac{k^2}{d^2}\langle \mathbf{x}_i, \mathbf{x}_l\rangle\Big) - \sum_{i=1}^{n}\|\mathbf{x}_i\|_2^2 - 2\sum_{i=1}^{n}\sum_{l=i+1}^{n}\langle \mathbf{x}_i, \mathbf{x}_l\rangle \Big\}$$

$$= \frac{1}{n^2}\Big(\frac{d}{k} - 1\Big)\sum_{i=1}^{n}\|\mathbf{x}_i\|_2^2$$

which is exactly the same MSE as Rand-$k$. This implies when $nk$ is large, the MSE of Rand-$k$-Spatial does not get improved compared to Rand-$k$ with correlation information. Intuitively, this implies when $nk \gg d$, the server gets enough amount of information from the client, and does not need correlation to improve its estimator. Hence, we focus on the more interesting case when $nk < d$ — that is, when the server does not have enough information from the clients, and thus wants to use additional information, i.e. cross-client correlation, to improve its estimator.

## B    Additional Details on the Rand-Proj-Spatial Family Estimator

### B.1    $\bar{\beta}$ is a scalar

From Eq. 20 in the proof of Theorem 4.3 and Eq. 25 in the proof of Theorem 4.4, it is evident that the unbiasedness of the mean estimator $\widehat{\mathbf{x}}^{\text{Rand-Proj-Spatial}}$ is ensured collectively by

- The random sampling matrices $\{\boldsymbol{E}_i\}$.
- The orthogonality of scaled Hadamard matrices $\boldsymbol{H}^T\boldsymbol{H} = d\boldsymbol{I}_d = \boldsymbol{H}\boldsymbol{H}^T$.
- The rademacher diagonal matrices, with the property $(\boldsymbol{D}_i)^2 = \boldsymbol{I}_d$.

### B.2    Alternative motivating regression problems

**Alternative motivating regression problem 1.**

Let $\boldsymbol{G}_i \in \mathbb{R}^{k \times d}$ and $\boldsymbol{W}_i \in \mathbb{R}^{d \times k}$ be the encoding and decoding matrix for client $i$. One possible alternative estimator that translates the intuition that the decoded vector should be close to the client's original vector, for all clients, is by solving the following regression problem,

$$\hat{\mathbf{x}} = \arg\min_{\boldsymbol{W}} f(\boldsymbol{W}) = \mathbb{E}[\|\bar{\mathbf{x}} - \frac{1}{n}\sum_{i=1}^{n} \boldsymbol{W}_i\boldsymbol{G}_i\mathbf{x}_i\|_2^2]$$

$$\text{subject to } \bar{\mathbf{x}} = \frac{1}{n}\sum_{i=1}^{n} \mathbb{E}[\boldsymbol{W}_i\boldsymbol{G}_i\mathbf{x}_i] \tag{15}$$

where $\boldsymbol{W} = (\boldsymbol{W}_1, \boldsymbol{W}_2, \ldots, \boldsymbol{W}_n)$ and the constraint enforces unbiasedness of the estimator. The estimator is then the solution of the above problem. However, we note that optimizing a decoding matrix $\boldsymbol{W}_i$ for each client leads to performing individual decoding of each client's compressed vector instead of a joint decoding process that considers all clients' compressed vectors. Only a joint decoding process can achieve the goal of leveraging cross-client information to reduce the estimation error. Indeed, we show as follows that solving the above optimization problem in Eq. 15 recovers the MSE of our baseline Rand-$k$. Note

$$f(\boldsymbol{W}) = \mathbb{E}[\|\frac{1}{n}\sum_{i=1}^{n}(\mathbf{x}_i - \boldsymbol{W}_i\boldsymbol{G}_i\mathbf{x}_i)\|_2^2] = \mathbb{E}[\|\frac{1}{n}\sum_{i=1}^{n}(\boldsymbol{I}_d - \boldsymbol{W}_i\boldsymbol{G}_i)\mathbf{x}_i\|_2^2]$$

$$= \mathbb{E}\Big[\frac{1}{n^2}\Big(\sum_{i=1}^{n}\|(\boldsymbol{I}_d - \boldsymbol{W}_i\boldsymbol{G}_i\mathbf{x}_i)\|_2^2 + \sum_{i \neq j}\Big\langle(\boldsymbol{I}_d - \boldsymbol{W}_i\boldsymbol{G}_i)\mathbf{x}_i, (\boldsymbol{I}_d - \boldsymbol{W}_j\boldsymbol{G}_j)\mathbf{x}_j\Big\rangle\Big)\Big]$$

$$= \frac{1}{n^2}\Big(\sum_{i=1}^{n}\mathbb{E}\Big[\|(\boldsymbol{I}_d - \boldsymbol{W}_i\boldsymbol{G}_i)\mathbf{x}_i\|_2^2\Big] + \sum_{i \neq j}\mathbb{E}\Big[\Big\langle(\boldsymbol{I}_d - \boldsymbol{W}_i\boldsymbol{G}_i)\mathbf{x}_i, (\boldsymbol{I}_d - \boldsymbol{W}_j\boldsymbol{G}_j)\mathbf{x}_j\Big\rangle\Big]\Big). \tag{16}$$

By the constraint of unbiasedness, i.e., $\bar{\mathbf{x}} = \frac{1}{n}\sum_{i=1}^{n}\mathbf{x}_i = \frac{1}{n}\sum_{i=1}^{n}\mathbb{E}[\boldsymbol{W}_i\boldsymbol{G}_i\mathbf{x}_i]$, there is

$$\frac{1}{n}\sum_{i=1}^{n}\mathbf{x}_i - \frac{1}{n}\sum_{i=1}^{n}\mathbb{E}[\boldsymbol{W}_i\boldsymbol{G}_i\mathbf{x}_i] = 0 \Leftrightarrow \frac{1}{n}\sum_{i=1}^{n}\mathbb{E}[(\boldsymbol{I}_d - \boldsymbol{W}_i\boldsymbol{G}_i)\mathbf{x}_i] = 0.$$

We now show that a sufficient and necessary condition to satisfy the above unbiasedness constraint is that for all $i \in [n]$, $\mathbb{E}[\boldsymbol{W}_i\boldsymbol{G}_i] = \boldsymbol{I}_d$.

*Sufficiency.* It is obvious that if for all $i \in [n]$, $\mathbb{E}[\boldsymbol{W}_i\boldsymbol{G}_i] = \boldsymbol{I}_d$, then we have $\frac{1}{n}\mathbb{E}[(\boldsymbol{I}_d - \boldsymbol{W}_i\boldsymbol{G}_i)\mathbf{x}_i] = 0$.

*Necessity.* Consider the special case that for some $i \in [n]$ and $\lambda \in [d]$, $\mathbf{x}_i = n\mathbf{e}_\lambda$, where $\mathbf{e}_\lambda$ is the $\lambda$-th canonical basis vector, and $\mathbf{x}_j = 0$, and for all $j \in [n] \setminus \{i\}$. Then,

$$\mathbf{e}_\lambda = \bar{\mathbf{x}} = \frac{1}{n}\sum_{i=1}^{n}\mathbb{E}[\boldsymbol{W}_i\boldsymbol{G}_i\mathbf{x}_i] = \frac{1}{n}\mathbb{E}[\boldsymbol{W}_i\boldsymbol{G}_i]\mathbf{e}_\lambda = [\mathbb{E}[\boldsymbol{W}_i\boldsymbol{G}_i]]_\lambda,$$

where $[\cdot]_\lambda$ denotes the $\lambda$-th column of matrix $\mathbb{E}[\boldsymbol{W}_i\boldsymbol{G}_i]$.

Since our approach is agnostic to the choice of vectors, we need this choice of decoder matrices, by varying $\lambda$ over $[d]$, we see that we need $\mathbb{E}[\boldsymbol{W}_i\boldsymbol{G}_i] = \boldsymbol{I}_d$. And by varying $i$ over $[n]$, we see that we need $\mathbb{E}[\boldsymbol{W}_j\boldsymbol{G}_j] = \boldsymbol{I}_d$ for all $j \in [n]$.

Therefore, $\bar{\mathbf{x}} = \frac{1}{n}\sum_{i=1}^{n}\mathbb{E}[\boldsymbol{W}_i\boldsymbol{G}_i\mathbf{x}_i] \Leftrightarrow \forall i \in [n], \mathbb{E}[\boldsymbol{W}_i\boldsymbol{G}_i] = \boldsymbol{I}_d$.

This implies the second term of $f(\boldsymbol{W})$ in Eq. 16 is 0, that is,

$$\sum_{i \neq j}\mathbb{E}\Big[\Big\langle(\boldsymbol{I}_d - \boldsymbol{W}_i\boldsymbol{G}_i)\mathbf{x}_i, (\boldsymbol{I}_d - \boldsymbol{W}_j\boldsymbol{G}_j)\mathbf{x}_j\Big\rangle\Big] = 0.$$

Hence, we only need to solve

$$\hat{\mathbf{x}} = \arg\min_{\boldsymbol{W}} f_2(\boldsymbol{W}) = \sum_{i=1}^{n}\mathbb{E}\Big[\|(\boldsymbol{I}_d - \boldsymbol{W}_i\boldsymbol{G}_i)\mathbf{x}_i\|_2^2\Big] \tag{17}$$

Since each $\boldsymbol{W}_i$ appears in $f_2(\boldsymbol{W})$ separately, each $\boldsymbol{W}_i$ can be optimized separately, via solving

$$\min_{\boldsymbol{W}_i}\mathbb{E}\Big[\|(\boldsymbol{I}_d - \boldsymbol{W}_i\boldsymbol{G}_i)\mathbf{x}_i\|_2^2\Big] \quad \text{subject to } \mathbb{E}[\boldsymbol{W}_i\boldsymbol{G}_i] = \boldsymbol{I}_d.$$

One natural solution is to take $\boldsymbol{W}_i = \frac{d}{k}\boldsymbol{G}_i^\dagger, \forall i \in [n]$. For $i \in [n]$, let $\boldsymbol{G}_i = \boldsymbol{V}_i\Lambda_i\boldsymbol{U}_i^T$ be its SVD, where $\boldsymbol{V}_i \in \mathbb{R}^{k \times d}$ and $\boldsymbol{U}_i \in \mathbb{R}^{d \times d}$ are orthogonal matrices. Then,

$$\boldsymbol{W}_i\boldsymbol{G}_i = \frac{d}{k}\boldsymbol{U}_i\Lambda_i^\dagger\boldsymbol{V}_i^T\boldsymbol{V}_i\Lambda\boldsymbol{U}^T = \frac{d}{k}\boldsymbol{U}_i\Lambda_i^\dagger\Lambda\boldsymbol{U}^T = \frac{d}{k}\boldsymbol{U}_i\Sigma\boldsymbol{U}_i^T,$$

where $\Sigma$ is a diagonal matrix with 0s and 1s on the diagonal.

For simplicity, we assume the random matrix $\boldsymbol{U}_i$ follows a continuous distribution. $\boldsymbol{U}_i$ being discrete follows a similar analysis. Let $\mu(\boldsymbol{U}_i)$ be the measure of $\boldsymbol{U}_i$.

$$\begin{aligned}
\mathbb{E}[\boldsymbol{W}_i\boldsymbol{G}_i] &= \frac{d}{k}\mathbb{E}[\boldsymbol{U}_i\Sigma\boldsymbol{U}_i^T] = \frac{d}{k}\int_{\boldsymbol{U}_i}\mathbb{E}[\boldsymbol{U}_i\Sigma_i\boldsymbol{U}_i^T \mid \boldsymbol{U}_i] \cdot d\mu(\boldsymbol{U}_i) \\
&= \frac{d}{k}\int_{\boldsymbol{U}_i}\boldsymbol{U}_i\mathbb{E}[\Sigma_i \mid \boldsymbol{U}_i]\boldsymbol{U}_i^T \cdot \mu(\boldsymbol{U}_i) \\
&= \frac{d}{k}\int_{\boldsymbol{U}_i}\boldsymbol{U}_i\frac{k}{d}\boldsymbol{I}_d\boldsymbol{U}_i^T \cdot d\mu(\boldsymbol{U}_i) \\
&= \frac{d}{k}\frac{k}{d}\boldsymbol{I}_d = \boldsymbol{I}_d,
\end{aligned}$$

which means the estimator $\frac{1}{n}\sum_{i=1}^{n}\frac{k}{d}\boldsymbol{G}_i^\dagger\boldsymbol{G}_i$ satisfies unbiasedness. The MSE is now

$$\begin{aligned}
MSE &= \mathbb{E}\Big[\|\bar{\mathbf{x}} - \frac{1}{n}\sum_{i=1}^{n}\boldsymbol{W}_i\boldsymbol{G}_i\mathbf{x}_i\|_2^2\Big] = \frac{1}{n^2}\sum_{i=1}^{n}\mathbb{E}\Big[\|(\boldsymbol{I}_d - \boldsymbol{W}_i\boldsymbol{G}_i)\mathbf{x}_i\|_2^2\Big] \\
&= \frac{1}{n^2}\sum_{i=1}^{n}\Big(\|\mathbf{x}_i\|_2^2 + \mathbb{E}[\|\boldsymbol{W}_i\boldsymbol{G}_i\mathbf{x}_i\|_2^2] - 2\langle\mathbf{x}_i, \mathbb{E}[\boldsymbol{W}_i\boldsymbol{G}_i]\mathbf{x}_i\rangle\Big) \\
&= \frac{1}{n^2}\sum_{i=1}^{n}\Big(\|\mathbf{x}_i\|_2^2 + \mathbb{E}[\|\boldsymbol{W}_i\boldsymbol{G}_i\mathbf{x}_i\|_2^2] - 2\langle\mathbf{x}_i, \mathbf{x}_i\rangle\Big)
\end{aligned}$$

$$= \frac{1}{n^2} \sum_{i=1}^{n} \Big( \mathbb{E}[\|\boldsymbol{W}_i \boldsymbol{G}_i \mathbf{x}_i\|_2^2 - \|\mathbf{x}_i\|_2^2] \Big)$$

$$= \frac{1}{n^2} \sum_{i=1}^{n} \Big( \mathbf{x}_i \mathbb{E}[(\boldsymbol{W}_i \boldsymbol{G}_i)^T (\boldsymbol{W}_i \boldsymbol{G}_i)] \mathbf{x}_i - \|\mathbf{x}_i\|_2^2 \Big).$$

Again, let $\boldsymbol{G}_i = \boldsymbol{V}_i \Lambda_i \boldsymbol{U}_i^T$ be its SVD and consider $\boldsymbol{W}_i \boldsymbol{G}_i = \frac{d}{k} \boldsymbol{U}_i \Sigma_i \boldsymbol{U}_i^T$, where $\Sigma_i$ is a diagonal matrix with 0s and 1s. Then,

$$MSE = \frac{1}{n^2} \sum_{i=1}^{n} \sum_{i=1}^{n} \Big( \mathbf{x}_i^T \frac{d^2}{k^2} \mathbb{E}[\boldsymbol{U}_i \Sigma_i \boldsymbol{U}_i^T \boldsymbol{U}_i \Sigma_i \boldsymbol{U}_i^T] \mathbf{x}_i - \|\mathbf{x}_i\|_2^2 \Big)$$

$$= \frac{1}{n^2} \sum_{i=1}^{n} \Big( \frac{d^2}{k^2} \mathbf{x}_i^T \mathbb{E}[\boldsymbol{U}_i \Sigma^2 \boldsymbol{U}_i^T] \mathbf{x}_i - \|\mathbf{x}_i\|_2^2 \Big).$$

Since $\boldsymbol{G}_i$ has rank $k$, $\Sigma_i$ is a diagonal matrix with $k$ out of $d$ entries being 1 and the rest being 0. Let $\mu(\boldsymbol{U}_i)$ be the measure of $\boldsymbol{U}_i$. Hence, for $i \in [n]$,

$$\mathbb{E}[\boldsymbol{U}_i \Sigma_i^2 \boldsymbol{U}_i^T] = \int_{\boldsymbol{U}_i} \mathbb{E}[\boldsymbol{U}_i \Sigma_i^2 \boldsymbol{U}_i^T \mid \boldsymbol{U}_i] d\mu(\boldsymbol{U}_i)$$

$$= \int_{\boldsymbol{U}_i} \boldsymbol{U}_i \mathbb{E}[\Sigma_i^2 \mid \boldsymbol{U}_i] \boldsymbol{U}_i^T d\mu(\boldsymbol{U}_i)$$

$$= \int_{\boldsymbol{U}_i} \frac{k}{d} \boldsymbol{U}_i \boldsymbol{I}_d \boldsymbol{U}_i^T d\mu(\boldsymbol{U}_i)$$

$$= \frac{k}{d} \int_{\boldsymbol{U}_i} \boldsymbol{I}_d d\mu(\boldsymbol{U}_i)$$

$$= \frac{k}{d} \boldsymbol{I}_d.$$

Therefore, the MSE of the estimator, which is the solution of the optimization problem in Eq. 15, is

$$MSE = \frac{1}{n^2} \sum_{i=1}^{n} \Big( \frac{d^2}{k^2} \mathbf{x}_i^T \frac{k}{d} \boldsymbol{I}_d \mathbf{x}_i - \|\mathbf{x}_i\|_2^2 \Big) = \frac{1}{n^2} \Big( \frac{d}{k} - 1 \Big) \sum_{i=1}^{n} \|\mathbf{x}_i\|_2^2,$$

which is the same MSE as that of Rand-$k$.

**Alternative motivating regression problem 2.**

Another motivating regression problem based on which we can design our estimator is

$$\widehat{\mathbf{x}} = \arg\min_{\mathbf{x}} \| \frac{1}{n} \sum_{i=1}^{n} \mathbf{G}_i \mathbf{x} - \frac{1}{n} \sum_{i=1}^{n} \mathbf{G}_i \mathbf{x}_i \|_2^2 \tag{18}$$

Note that $\boldsymbol{G}_i \in \mathbb{R}^{k \times d}, \forall i \in [n]$, and so the solution to the above problem is

$$\widehat{\mathbf{x}}^{\text{(solution)}} = \Big( \frac{1}{n} \sum_{i=1}^{n} \boldsymbol{G}_i \Big)^{\dagger} \Big( \frac{1}{n} \sum_{i=1}^{n} \boldsymbol{G}_i \mathbf{x}_i \Big),$$

and to ensure unbiasedness of the estimator, we can set $\bar{\beta} \in \mathbb{R}$ and have the estimator as

$$\widehat{\mathbf{x}}^{\text{(estimator)}} = \bar{\beta} \Big( \frac{1}{n} \sum_{i=1}^{n} \boldsymbol{G}_i \Big)^{\dagger} \Big( \frac{1}{n} \sum_{i=1}^{n} \boldsymbol{G}_i \mathbf{x}_i \Big).$$

It is not hard to see this estimator does not lead to an MSE as low as Rand-Proj-Spatial does. Consider the full correlation case, i.e., $\mathbf{x}_i = \mathbf{x}, \forall i \in [n]$, for example, the estimator is now

$$\widehat{\mathbf{x}}^{\text{(estimator)}} = \bar{\beta} \Big( \frac{1}{n} \sum_{i=1}^{n} \boldsymbol{G}_i \Big)^{\dagger} \Big( \frac{1}{n} \sum_{i=1}^{n} \boldsymbol{G}_i \Big) \mathbf{x}.$$

Note that $\text{rank}(\frac{1}{n}\sum_{i=1}^{n} \boldsymbol{G}_i)$ is at most $k$, since $\boldsymbol{G}_i \in \mathbb{R}^{k \times d}$, $\forall i \in [k]$. This limits the amount of information of $\mathbf{x}$ the server can recover.

While recall that in this case, the Rand-Proj-Spatial estimator is

$$\widehat{\mathbf{x}}^{(\text{Rand-Proj-Spatial})} = \bar{\beta}\Big(\sum_{i=1}^{n} \boldsymbol{G}_i^T \boldsymbol{G}_i\Big)^{\dagger} \sum_{i=1}^{n} \boldsymbol{G}_i^T \boldsymbol{G}_i \mathbf{x} = \bar{\beta}\boldsymbol{S}^{\dagger}\boldsymbol{S}\mathbf{x},$$

where $\boldsymbol{S}$ can have rank at most $nk$.

## B.3  Why deriving the MSE of Rand-Proj-Spatial with SRHT is hard

To analyze Eq. 11, one needs to compute the distribution of eigendecomposition of $\boldsymbol{S} = \sum_{i=1}^{n} \boldsymbol{G}_i^T \boldsymbol{G}_i$, i.e. the sum of the covariance of SRHT. To the best of our knowledge, there is no non-trivial closed form expression of the distribution of eigen-decomposition of even a single $\boldsymbol{G}_i^T \boldsymbol{G}_i$, when $\boldsymbol{G}_i$ is SRHT, or other commonly used random matrices, e.g. Gaussian. When $\boldsymbol{G}_i$ is SRHT, since $\boldsymbol{G}_i^T \boldsymbol{G}_i = \boldsymbol{D}_i \boldsymbol{H} \boldsymbol{E}_i^T \boldsymbol{E}_i \boldsymbol{H} \boldsymbol{D}_i$ and the eigenvalues of $\boldsymbol{E}_i^T \boldsymbol{E}_i$ are just diagonal entries, one might attempt to analyze $\boldsymbol{H}\boldsymbol{D}_i$. While the hardmard matrix $\boldsymbol{H}$'s eigenvalues and eigenvectors are known[3], the result can hardly be applied to analyze the distribution of singular values or singular vectors of $\boldsymbol{H}\boldsymbol{D}_i$.

Even if one knows the eigen-decomposition of a single $\boldsymbol{G}_i^T \boldsymbol{G}_i$, it is still hard to get the eigen-decomposition of $\boldsymbol{S}$. The eigenvalues of a matrix $\boldsymbol{A}$ can be viewed as a non-linear function in the $\boldsymbol{A}$, and hence it is in general hard to derive closed form expressions for the eigenvalues of $\boldsymbol{A} + \boldsymbol{B}$, given the eigenvalues of $\boldsymbol{A}$ and that of $\boldsymbol{B}$. One exception is when $\boldsymbol{A}$ and $\boldsymbol{B}$ have the same eigenvector and the eigenvalues of $\boldsymbol{A} + \boldsymbol{B}$ becomes a sum of the eigenvalues of $\boldsymbol{A}$ and $\boldsymbol{B}$. Recall when $\boldsymbol{G}_i = \boldsymbol{E}_i$, Rand-Proj-Spatial recovers Rand-$k$-Spatial. Since $\boldsymbol{E}_i^T \boldsymbol{E}_i$'s all have the same eigenvectors (i.e. same as $\boldsymbol{I}_d$), the eigenvalues of $\boldsymbol{S} = \sum_{i=1}^{n} \boldsymbol{E}_i^T \boldsymbol{E}_i$ are just the sum of diagonal entries of $\boldsymbol{E}_i^T \boldsymbol{E}_i$'s. Hence, deriving the MSE for Rand-$k$-Spatial is not hard compared to the more general case when $\boldsymbol{G}_i^T \boldsymbol{G}_i$'s can have different eigenvectors.

Since one can also view $\boldsymbol{S} = \sum_{i=1}^{nk} \mathbf{g}_i \mathbf{g}_i^T$, i.e. the sum of $nk$ rank-one matrices, one might attempt to recursively analyze the eigen-decomposition of $\sum_{i=1}^{n'} \mathbf{g}_i \mathbf{g}_i^T + \mathbf{g}_{n'+1} \mathbf{g}_{n'+1}^T$ for $n' \leq n$. One related problem is eigen-decomposition of a low-rank updated matrix in perturbation analysis: Given the eigen-decomposition of a matrix $\boldsymbol{A}$, what is the eigen-decomposition of $\boldsymbol{A} + \boldsymbol{V}\boldsymbol{V}^T$, where $\boldsymbol{V}$ is low-rank matrix (or more commonly rank-one)? To compute the eigenvalues of $\boldsymbol{A} + \boldsymbol{V}\boldsymbol{V}^T$ directly from that of $\boldsymbol{A}$, the most effective and widely applied solution is to solve the so-called secular equation, e.g. [59, 60, 61]. While this can be done computationally efficiently, it is hard to get a closed form expression for the eigenvalues of $\boldsymbol{A} + \boldsymbol{V}\boldsymbol{V}^T$ from the secular equation.

The previous analysis of SRHT in e.g. [37, 38, 39, 45, 55] is based on asymptotic properties of SRHT, such as the limiting eigen-spectrum, or concentration bounds that bounds the singular values. To analyze the MSE of Rand-Proj-Spatial, however, we need an exact, non-asymptotic analysis of the distribution of SRHT. Concentration bounds does not apply, since computing the pseudo-inverse in Eq. 5 naturally bounds the eigenvalues, and applying concentration bounds will only lead to a loose upper bound on MSE.

---

[3]See this note `https://core.ac.uk/download/pdf/81967428.pdf`

### B.4 More simulation results on incorporating various degrees of correlation

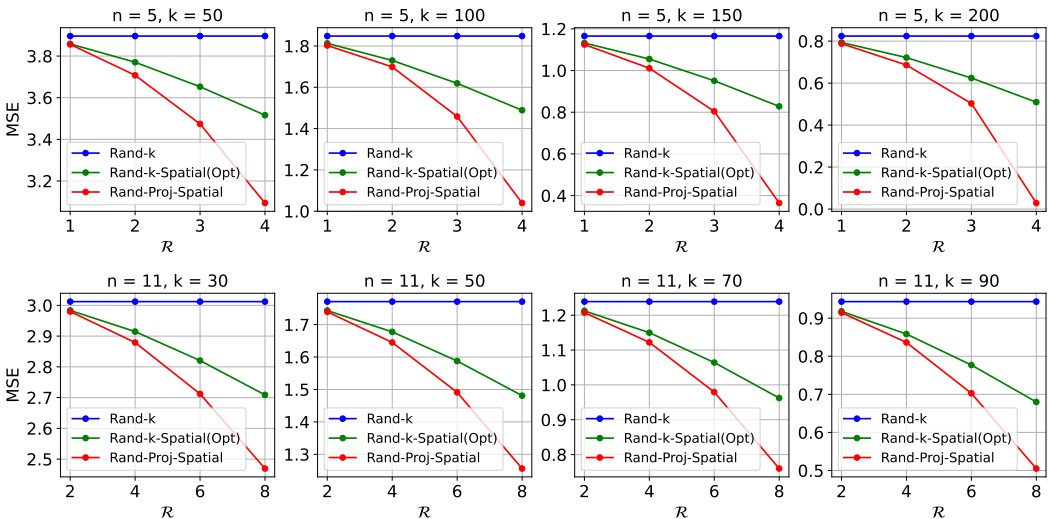

Figure 6: MSE comparison of estimators Rand-$k$, Rand-$k$-Spatial(Opt), Rand-Proj-Spatial, given the degree of correlation $\mathcal{R}$. Rand-$k$-Spatial(Opt) denotes the estimator that gives the lowest possible MSE from the Rand-$k$-Spatial family. We consider $d = 1024$, a smaller number of clients $n \in \{5, 11\}$, and $k$ values such that $nk < d$. In each plot, we fix $n, k, d$ and vary the degree of positive correlation $\mathcal{R}$. Note the range of $\mathcal{R}$ is $\mathcal{R} \in [0, n-1]$. We choose $\mathcal{R}$ with equal space in this range.

## C All Proof Details

### C.1 Proof of Theorem 4.3

**Theorem 4.3** (MSE under Full Correlation). *Consider $n$ clients, each holding the same vector $\mathbf{x} \in \mathbb{R}^d$. Suppose we set $T(\lambda) = \lambda$, $\bar{\beta} = \frac{d}{k}$ in Eq. 5, and the random linear map $\boldsymbol{G}_i$ at each client to be an SRHT matrix. Let $\delta$ be the probability that $\boldsymbol{S} = \sum_{i=1}^n \boldsymbol{G}_i^T \boldsymbol{G}_i$ does not have full rank. Then, for $nk \leq d$,*

$$\mathbb{E}\left[\|\widehat{\mathbf{x}}^{(\textit{Rand-Proj-Spatial(Max)})} - \bar{\mathbf{x}}\|_2^2\right] \leq \left[\frac{d}{(1-\delta)nk + \delta k} - 1\right]\|\mathbf{x}\|_2^2 \tag{19}$$

*Proof.* All clients have the same vector $\mathbf{x}_1 = \mathbf{x}_2 = \cdots = \mathbf{x}_n = \mathbf{x} \in \mathbb{R}^d$. Hence, $\bar{\mathbf{x}} = \frac{1}{n}\sum_{i=1}^n \mathbf{x}_i = \mathbf{x}$, and the decoding scheme is

$$\widehat{\mathbf{x}}^{(\textit{Rand-Proj-Spatial(Max)})} = \bar{\beta}\Big(\sum_{i=1}^n \boldsymbol{G}_i^T \boldsymbol{G}_i\Big)^\dagger \sum_{i=1}^n \boldsymbol{G}_i^T \boldsymbol{G}_i \mathbf{x} = \bar{\beta}\boldsymbol{S}^\dagger \boldsymbol{S}\mathbf{x},$$

where $\boldsymbol{S} = \sum_{i=1}^n \boldsymbol{G}_i^T \boldsymbol{G}_i$. Let $\boldsymbol{S} = \boldsymbol{U}\Lambda\boldsymbol{U}^T$ be its eigendecomposition. Since $\boldsymbol{S}$ is a real symmetric matrix, $\boldsymbol{U}$ is orthogonal, i.e., $\boldsymbol{U}^T\boldsymbol{U} = \boldsymbol{I}_d = \boldsymbol{U}\boldsymbol{U}^T$. Also, $\boldsymbol{S}^\dagger = \boldsymbol{U}\Lambda^\dagger\boldsymbol{U}^T$, where $\Lambda^\dagger$ is a diagonal matrix, such that

$$[\Lambda^\dagger]_{ii} = \begin{cases} 1/[\Lambda]_{ii} & \text{if } \Lambda_{ii} \neq 0, \\ 0 & \text{else.} \end{cases}$$

Let $\delta_c$ be the probability that $\boldsymbol{S}$ has rank $c$, for $c \in \{k, k+1, \ldots, nk-1\}$. Note that $\delta = \sum_{c=k}^{nk-1} \delta_c$. For vector $\mathbf{m} \in \mathbb{R}^d$, we use $\text{diag}(\mathbf{m}) \in \mathbb{R}^{d \times d}$ to denote the matrix whose diagonal entries correspond to the coordinates of $\mathbf{m}$ and the rest of the entries are zeros.

**Computing $\bar{\beta}$.** First, we compute $\bar{\beta}$. To ensure that our estimator $\widehat{\mathbf{x}}^{(\textit{Rand-Proj-Spatial(Max)})}$ is unbiased, we need $\bar{\beta}\mathbb{E}[\boldsymbol{S}^\dagger \boldsymbol{S}\mathbf{x}] = \mathbf{x}$. Consequently,

$$\mathbf{x} = \bar{\beta}\mathbb{E}[\boldsymbol{U}\Lambda^\dagger\boldsymbol{U}^T\boldsymbol{U}\Lambda\boldsymbol{U}^T]\mathbf{x}$$

$$= \bar{\beta} \left[ \sum_{U=\Phi} \Pr[U = \Phi] \mathbb{E}[U \Lambda^{\dagger} \Lambda U^T \mid U = \Phi] \right] \mathbf{x}$$

$$= \bar{\beta} \left[ \sum_{U=\Phi} \Pr[U = \Phi] U \mathbb{E}[\Lambda^{\dagger} \Lambda \mid U = \Phi] U^T \right] \mathbf{x}$$

$$\overset{(a)}{=} \bar{\beta} \left[ \sum_{U=\Phi} \Pr[U = \Phi] U \mathbb{E}[\mathrm{diag}(\mathbf{m}) \mid U = \Phi] U^T \right] \mathbf{x}$$

$$\overset{(b)}{=} \bar{\beta} \sum_{U=\Phi} \Pr[U = \Phi] \left[ U \Big( (1-\delta) \frac{nk}{d} I_d + \sum_{c=k}^{nk-1} \delta_c \frac{c}{d} I_d \Big) U^T \right] \mathbf{x}$$

$$= \bar{\beta} \Big[ (1-\delta) \frac{nk}{d} + \sum_{c=k}^{nk-1} \delta_c \frac{c}{d} \Big] \mathbf{x}$$

$$\Rightarrow \bar{\beta} = \frac{d}{(1-\delta)nk + \sum_{c=k}^{nk-1} \delta_c c} \tag{20}$$

where in $(a)$, $\mathbf{m} \in \mathbb{R}^d$ such that

$$\mathbf{m}_i = \begin{cases} 1 & \text{if } \Lambda_{jj} > 0 \\ 0 & \text{else.} \end{cases}$$

Also, by construction of $S$, $\mathrm{rank}(\mathrm{diag}(\mathbf{m})) \leq nk$. Further, $(b)$ follows by symmetry across the $d$ dimensions.

Since $\delta k \leq \sum_{c=k}^{nk-1} \delta_c c \leq \delta(nk-1)$, there is

$$\frac{d}{(1-\delta)nk + \delta(nk-1)} \leq \bar{\beta} \leq \frac{d}{(1-\delta)nk + \delta k} \tag{21}$$

**Computing the MSE.** Next, we use the value of $\bar{\beta}$ in Eq. 20 to compute MSE.

$$MSE(\text{Rand-Proj-Spatial(Max)}) = \mathbb{E}[\|\widehat{\mathbf{x}}^{(\text{Rand-Proj-Spatial(Max)})} - \bar{\mathbf{x}}\|_2^2] = \mathbb{E}[\|\bar{\beta} S^{\dagger} S \mathbf{x} - \mathbf{x}\|_2^2]$$

$$= \bar{\beta}^2 \mathbb{E}[\|S^{\dagger} S \mathbf{x}\|_2^2] + \|\mathbf{x}\|_2^2 - 2 \Big\langle \bar{\beta} \mathbb{E}[S^{\dagger} S \mathbf{x}], \mathbf{x} \Big\rangle$$

$$= \bar{\beta}^2 \mathbb{E}[\|S^{\dagger} S \mathbf{x}\|_2^2] - \|\mathbf{x}\|_2^2 \qquad \text{(Using unbiasedness of } \widehat{\mathbf{x}}^{(\text{Rand-Proj-Spatial(Max)})})$$

$$= \bar{\beta}^2 \mathbf{x}^T \mathbb{E}[S^T (S^{\dagger})^T S^{\dagger} S] \mathbf{x} - \|\mathbf{x}\|_2^2. \tag{22}$$

Using $S^{\dagger} = U \Lambda^{\dagger} U^T$,

$$\mathbb{E}[S^T (S^{\dagger})^T S^{\dagger} S] = \mathbb{E}[U \Lambda U^T U \Lambda^{\dagger} U^T U \Lambda^{\dagger} U^T U \Lambda U^T]$$

$$= \mathbb{E}[U \Lambda (\Lambda^{\dagger})^2 \Lambda U^T]$$

$$= \sum_{U=\Phi} U \mathbb{E}[\Lambda (\Lambda^{\dagger})^2 \Lambda] U^T \cdot \Pr[U = \Phi]$$

$$= \sum_{U=\Phi} U \Big[ (1-\delta) \frac{nk}{d} I_d + \sum_{c=k}^{nk-1} \delta_c \frac{c}{d} I_d \Big] U^T \cdot \Pr[U = \Phi]$$

$$= \Big[ (1-\delta) \frac{nk}{d} + \sum_{c=k}^{nk-1} \delta_c \frac{c}{d} \Big] \cdot \sum_{U=\Phi} U U^T \cdot \Pr[U = \Phi]$$

$$= \Big[ (1-\delta) \frac{nk}{d} + \sum_{c=k}^{nk-1} \delta_c \frac{c}{d} \Big] I_d$$

$$= \frac{1}{\bar{\beta}} I_d \tag{23}$$

Substituting Eq. 23 in Eq. 22, we get

$$MSE(\text{Rand-Proj-Spatial(Max)}) = \bar{\beta}^2 \mathbf{x}^T \frac{1}{\bar{\beta}} \boldsymbol{I}_d \mathbf{x} - \|\mathbf{x}\|_2^2 = (\bar{\beta} - 1)\|\mathbf{x}\|_2^2$$

$$\leq \left[ \frac{d}{(1-\delta)nk + \delta k} - 1 \right] \|\mathbf{x}\|_2^2,$$

where the inequality is by Eq 21. $\qquad\square$

## C.2 Comparing against Rand-$k$

Next, we compare the MSE of Rand-Proj-Spatial(Max) with the MSE of the baseline Rand-$k$ analytically in the full-correlation case. Recall that in this case,

$$MSE(\text{Rand-}k) = \frac{1}{n}(\frac{d}{k} - 1)\|\mathbf{x}\|_2^2.$$

We have

$$MSE(\text{Rand-Proj-Spatial(Max)}) \leq MSE(\text{Rand-}k)$$

$$\Leftrightarrow \frac{d}{(1-\delta)nk + \delta k} - 1 \leq \frac{1}{n}(\frac{d}{k} - 1)$$

$$\Leftrightarrow \frac{d}{k} \frac{n - (1-\delta)n - \delta}{n((1-\delta)n + \delta)} \leq 1 - \frac{1}{n}$$

$$\Leftrightarrow \frac{d}{k} \cdot \frac{\delta - \delta/n}{(1-\delta)n + \delta} \leq \frac{n-1}{n}$$

$$\Leftrightarrow d\delta(1 - \frac{1}{n})n \leq k(n-1) \cdot ((1-\delta)n + \delta)$$

$$\Leftrightarrow d\delta \leq k \cdot ((1-\delta)n + \delta)$$

$$\Leftrightarrow d\delta + kn\delta - k\delta \leq kn$$

$$\Leftrightarrow \delta \leq \frac{kn}{d + kn - k}$$

$$\Leftrightarrow \delta \leq \frac{1}{\frac{d}{kn} + 1 - \frac{1}{n}}$$

Since $nk \leq d$, for $n \geq 2$, the above implies when

$$\delta \leq \frac{1}{1 + \frac{1}{2}} = \frac{2}{3},$$

the MSE of Rand-Proj-Spatial(Max) is always less than that of Rand-$k$.

## C.3 $\boldsymbol{S}$ has full rank with high probability

We empirically verify that $\delta \approx 0$. With $d \in \{32, 64, 128, \ldots, 1024\}$ and 4 different $nk$ value such that $nk \leq d$ for each $d$, we compute rank($\boldsymbol{S}$) for $10^5$ trials for each pair of $(nk, d)$ values, and plot the results for all trials. All results are presented in Figure 7. As one can observe from the plots, rank($\boldsymbol{S}$) $= nk$ with high probability, suggesting $\delta \approx 0$.

This implies the MSE of Rand-Proj-Spatial(Max) is

$$MSE(\text{Rand-Proj-Spatial(Max)}) \approx (\frac{d}{nk} - 1)\|\mathbf{x}\|_2^2,$$

in the full correlation case.

## C.4 Proof of Theorem 4.4

**Theorem 4.4** (MSE under No Correlation). *Consider $n$ clients, each holding a vector $\mathbf{x}_i \in \mathbb{R}^d$, $\forall i \in [n]$. Suppose we set $T \equiv 1$, $\bar{\beta} = \frac{d^2}{k}$ in Eq. 5, and the random linear map $\boldsymbol{G}_i$ at each client to be*

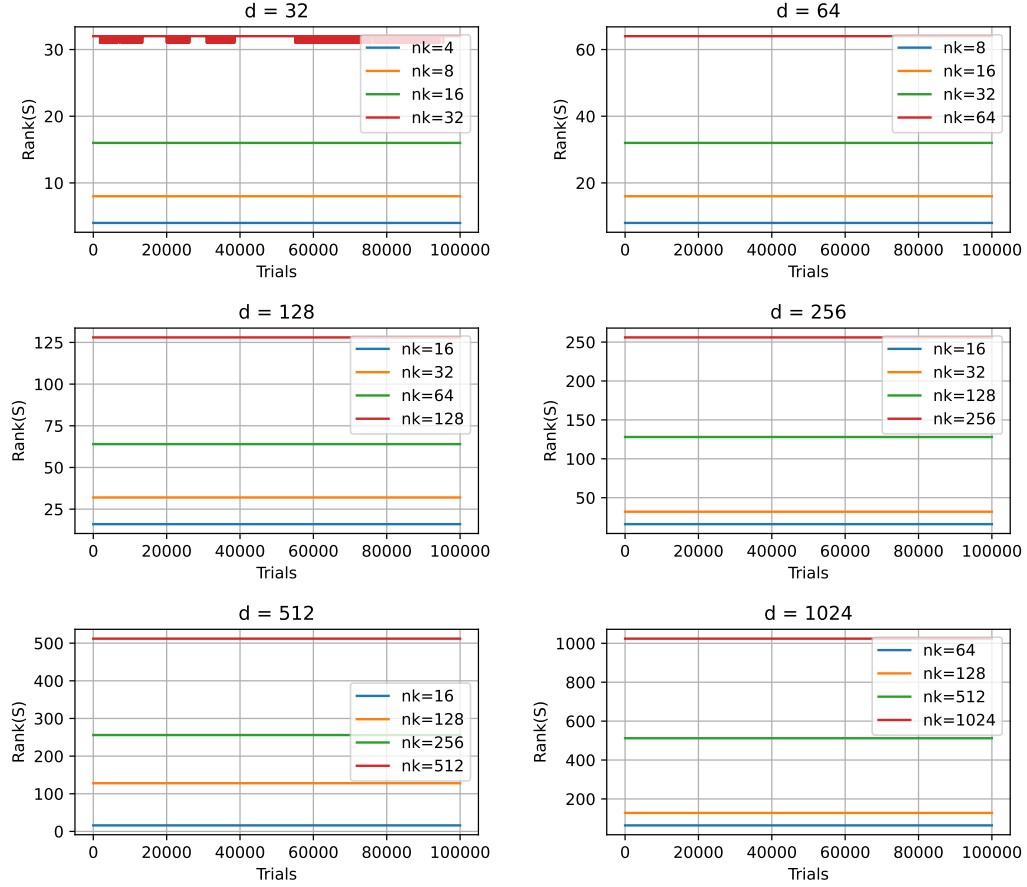

Figure 7: Simulation results of rank($\boldsymbol{S}$), where $\boldsymbol{S} = \sum_{i=1}^{n} \boldsymbol{G}_i^T \boldsymbol{G}_i$, with $\boldsymbol{G}_i$ being SRHT. With $d \in \{32, 64, 128, \ldots, 1024\}$ and 4 different $nk$ values such that $nk \leq d$ for each $d$, we compute rank($\boldsymbol{S}$) for $10^5$ trials for each pairs of $(nk, d)$ values and plot the results for all trials. When $d = 32$ and $nk = 32$ in the first plot, rank($\boldsymbol{S}$) = 31 in 2100 trials, and rank($\boldsymbol{S}$) = $nk$ = 32 in all the rest of the trials. For all other $(nk, d)$ pairs, $\boldsymbol{S}$ always has rank $nk$ in the $10^5$ trials. This verifies that $\delta = \Pr[\text{rank}(\boldsymbol{S}) < nk] \approx 0$.

*an SRHT matrix. Then, for $nk \leq d$,*

$$\mathbb{E}\left[\|\widehat{\mathbf{x}}^{(\textit{Rand-Proj-Spatial})} - \bar{\mathbf{x}}\|_2^2\right] = \frac{1}{n^2}\left(\frac{d}{k} - 1\right)\sum_{i=1}^{n}\|\mathbf{x}_i\|_2^2.$$

*Proof.* When the client vectors are all orthogonal to each other, we define the transformation function on the eigenvalue to be $T(\lambda) = 1, \forall \lambda \geq 0$. We show that by considering the above constant $T$, SRHT becomes the same as rand $k$. Recall $\boldsymbol{S} = \sum_{i=1}^{n} \boldsymbol{G}_i^T \boldsymbol{G}_i$ and let $\boldsymbol{G}^T \boldsymbol{G} = \boldsymbol{U}\Lambda\boldsymbol{U}^T$ be its eigendecompostion. Then,

$$T(\boldsymbol{S}) = \boldsymbol{U}T(\Lambda)\boldsymbol{U}^T = \boldsymbol{U}\boldsymbol{I}_d\boldsymbol{U}^T = \boldsymbol{I}_d.$$

Hence, $(T(\boldsymbol{S}))^\dagger = \boldsymbol{I}_d$. And the decoded vector for client $i$ becomes

$$\widehat{\mathbf{x}}_i = \bar{\beta}\Big(T(\boldsymbol{G}^T\boldsymbol{G})\Big)^\dagger \boldsymbol{G}_i^T \boldsymbol{G}_i\mathbf{x}_i = \bar{\beta}\boldsymbol{G}_i^T\boldsymbol{G}_i\mathbf{x}_i = \bar{\beta}\frac{1}{d}\boldsymbol{D}_i\boldsymbol{H}^T\boldsymbol{E}_i^T\boldsymbol{E}_i\boldsymbol{H}\boldsymbol{D}_i\mathbf{x}_i,$$

$$\widehat{\mathbf{x}} = \frac{1}{n}\sum_{i=1}^{n}\widehat{\mathbf{x}}_i = \frac{1}{n}\bar{\beta}\sum_{i=1}^{n}\frac{1}{d}\boldsymbol{D}_i\boldsymbol{H}^T\boldsymbol{E}_i^T\boldsymbol{E}_i\boldsymbol{H}\boldsymbol{D}_i\mathbf{x}_i \tag{24}$$

$D_i$ is a diagonal matrix. Also, $E_i^T E_i \in \mathbb{R}^{d \times d}$ is a diagonal matrix, where the $i$-th entry is 0 or 1.

**Computing $\bar{\beta}$.** To ensure that $\widehat{\mathbf{x}}$ is an unbiased estimator, from Eq. 24

$$
\begin{aligned}
\mathbf{x}_i &= \bar{\beta} \mathbb{E}[\boldsymbol{G}_i^T \boldsymbol{G}_i] \mathbf{x}_i \\
&= \frac{\bar{\beta}}{d} \mathbb{E}[\boldsymbol{D}_i \boldsymbol{H}^T \boldsymbol{E}_i^T \boldsymbol{E}_i \boldsymbol{H} \boldsymbol{D}_i] \mathbf{x}_i \\
&= \frac{\bar{\beta}}{d} \mathbb{E}_{\boldsymbol{D}_i} \Big[ \boldsymbol{D}_i \boldsymbol{H}^T \underbrace{\mathbb{E}[\boldsymbol{E}_i^T \boldsymbol{E}_i]}_{=(k/d)\boldsymbol{I}_d} \boldsymbol{H} \boldsymbol{D}_i \Big] \mathbf{x}_i && (\because \boldsymbol{E}_i \text{ is independent of } \boldsymbol{D}_i) \\
&= \frac{\bar{\beta}}{d} k \mathbb{E}_{\boldsymbol{D}_i} \big[ \boldsymbol{D}_i^2 \big] \mathbf{x}_i && (\because \boldsymbol{H}^T \boldsymbol{H} = d\boldsymbol{I}_d) \\
&= \frac{\bar{\beta} k}{d} \mathbf{x}_i && (\because \boldsymbol{D}_i^2 = \boldsymbol{I} \text{ is now deterministic.}) \\
\Rightarrow \bar{\beta} &= \frac{d}{k}. && (25)
\end{aligned}
$$

**Computing the MSE.**

$$
\begin{aligned}
MSE &= \mathbb{E}\Big\| \widehat{\mathbf{x}} - \bar{\mathbf{x}} \Big\|_2^2 \\
&= \mathbb{E}\Big\| \frac{1}{n} \bar{\beta} \sum_{i=1}^n \frac{1}{d} \boldsymbol{D}_i \boldsymbol{H}^T \boldsymbol{E}_i^T \boldsymbol{E}_i \boldsymbol{H} \boldsymbol{D}_i \mathbf{x}_i - \frac{1}{n} \sum_{i=1}^n \mathbf{x}_i \Big\|_2^2 \\
&= \frac{1}{n^2} \Big\{ \mathbb{E}\Big\| \bar{\beta} \sum_{i=1}^n \frac{1}{d} \boldsymbol{D}_i \boldsymbol{H}^T \boldsymbol{E}_i^T \boldsymbol{E}_i \boldsymbol{H} \boldsymbol{D}_i \mathbf{x}_i \Big\|_2^2 + \Big\| \sum_{i=1}^n \mathbf{x}_i \Big\|_2^2 \\
&\qquad -2\Big\langle \bar{\beta} \mathbb{E}[\sum_{i=1}^n \frac{1}{d} \boldsymbol{D}_i \boldsymbol{H}^T \boldsymbol{E}_i^T \boldsymbol{E}_i \boldsymbol{H} \boldsymbol{D}_i \mathbf{x}_i], \sum_{i=1}^n \mathbf{x}_i \Big\rangle \Big\} \\
&= \frac{1}{n^2} \Big\{ \bar{\beta}^2 \mathbb{E}\Big\| \sum_{i=1}^n \frac{1}{d} \boldsymbol{D}_i \boldsymbol{H}^T \boldsymbol{E}_i^T \boldsymbol{E}_i \boldsymbol{H} \boldsymbol{D}_i \mathbf{x}_i \Big\|_2^2 - \Big\| \sum_{i=1}^n \mathbf{x}_i \Big\|_2^2 \Big\} && (\because \mathbb{E}[\widehat{\mathbf{x}}] = \bar{\mathbf{x}}) \\
&= \frac{1}{n^2} \Big\{ \sum_{i=1}^n \frac{\bar{\beta}^2}{d^2} \mathbb{E}\Big\| \boldsymbol{D}_i \boldsymbol{H}^T \boldsymbol{E}_i^T \boldsymbol{E}_i \boldsymbol{H} \boldsymbol{D}_i \mathbf{x}_i \Big\|_2^2 - \sum_{i=1}^n \big\| \mathbf{x}_i \big\|_2^2 && (26) \\
&\qquad +2 \sum_{i=1}^n \sum_{l=i+1}^n \frac{\bar{\beta}^2}{d^2} \Big\langle \mathbb{E}[\boldsymbol{D}_i \boldsymbol{H}^T \boldsymbol{E}_i^T \boldsymbol{E}_i \boldsymbol{H} \boldsymbol{D}_i \mathbf{x}_i], \mathbb{E}[\boldsymbol{D}_l \boldsymbol{H}^T \boldsymbol{E}_l^T \boldsymbol{E}_l \boldsymbol{H} \boldsymbol{D}_l \mathbf{x}_l] \Big\rangle - 2 \sum_{i=1}^n \sum_{l=i+1}^n \Big\langle \mathbf{x}_i, \mathbf{x}_l \Big\rangle \Big\}.
\end{aligned}
$$

Note that in Eq. 26

$$
\begin{aligned}
\mathbb{E}\Big\| \boldsymbol{D}_i \boldsymbol{H}^T \boldsymbol{E}_i^T \boldsymbol{E}_i \boldsymbol{H} \boldsymbol{D}_i \mathbf{x}_i \Big\|_2^2 &= \mathbb{E}[\mathbf{x}_i^T \boldsymbol{D}_i \boldsymbol{H}^T \boldsymbol{E}_i^T \boldsymbol{E}_i \boldsymbol{H} \boldsymbol{D}_i \boldsymbol{D}_i \boldsymbol{H}^T \boldsymbol{E}_i^T \boldsymbol{E}_i \boldsymbol{H} \boldsymbol{D}_i \mathbf{x}_i] \\
&= d\mathbb{E}[\mathbf{x}_i^T \boldsymbol{D}_i \boldsymbol{H}^T (\boldsymbol{E}_i^T \boldsymbol{E}_i)^2 \boldsymbol{H} \boldsymbol{D}_i \mathbf{x}_i] && (\because \boldsymbol{D}_i^2 = \boldsymbol{I}_d; \boldsymbol{H}^T \boldsymbol{H} = \boldsymbol{H} \boldsymbol{H}^T = d\boldsymbol{I}_d) \\
&= d\mathbf{x}_i^T \mathbb{E}_{\boldsymbol{D}_i} \big[ \boldsymbol{D}_i \boldsymbol{H}^T \mathbb{E}[\boldsymbol{E}_i^T \boldsymbol{E}_i] \boldsymbol{H} \boldsymbol{D}_i \big] \mathbf{x}_i && (\boldsymbol{E}_i, \boldsymbol{D}_i \text{ are independent}; (\boldsymbol{E}_i^T \boldsymbol{E}_i)^2 = \boldsymbol{E}_i^T \boldsymbol{E}_i) \\
&= kd\|\mathbf{x}_i\|_2^2, && (27)
\end{aligned}
$$

since $\mathbb{E}[\boldsymbol{E}_i^T \boldsymbol{E}_i] = (k/d)\boldsymbol{I}_d$, $\boldsymbol{H}^T \boldsymbol{H} = d\boldsymbol{I}_d$ and for $i \neq l$

$$
\Big\langle \mathbb{E}[\boldsymbol{D}_i \boldsymbol{H}^T \boldsymbol{E}_i^T \boldsymbol{E}_i \boldsymbol{H} \boldsymbol{D}_i \mathbf{x}_i], \mathbb{E}[\boldsymbol{D}_l \boldsymbol{H}^T \boldsymbol{E}_l^T \boldsymbol{E}_l \boldsymbol{H} \boldsymbol{D}_l \mathbf{x}_l] \Big\rangle = \Big\langle k\mathbf{x}_i, k\mathbf{x}_l \Big\rangle = k^2 \Big\langle \mathbf{x}_i, \mathbf{x}_l \Big\rangle. \qquad (28)
$$

Substituting Eq. 27, 28 in Eq. 26, we get

$$
\begin{aligned}
MSE &= \frac{1}{n^2} \Big\{ \Big( \frac{\bar{\beta}^2}{d^2} \sum_{i=1}^n kd\|\mathbf{x}_i\|_2^2 + 2 \sum_{i=1}^n \sum_{l=i+1}^n \frac{\bar{\beta}^2 k^2}{d^2} \Big\langle \mathbf{x}_i, \mathbf{x}_l \Big\rangle \Big) - \sum_{i=1}^n \big\| \mathbf{x}_i \big\|_2^2 - 2 \sum_{i=1}^n \sum_{l=i+1}^n \Big\langle \mathbf{x}_i, \mathbf{x}_l \Big\rangle \Big\} \\
&= \frac{1}{n^2} \Big( \frac{d}{k} - 1 \Big) \sum_{i=1}^n \|\mathbf{x}_i\|_2^2,
\end{aligned}
$$

which is exactly the same as the MSE of rand $k$. $\qquad \square$

## C.5   Rand-Proj-Spatial recovers Rand-$k$-Spatial (Proof of Lemma 4.1)

**Lemma 4.1** (Recovering Rand-$k$-Spatial)**.** *Suppose client $i$ generates a subsampling matrix $\boldsymbol{E}_i = [\mathbf{e}_{i_1}, \quad \ldots, \quad \mathbf{e}_{i_k}]^\top$, where $\{\mathbf{e}_j\}_{j=1}^d$ are the canonical basis vectors, and $\{i_1, \ldots, i_k\}$ are sampled from $\{1, \ldots, d\}$ without replacement. The encoded vectors are given as $\widehat{\mathbf{x}}_i = \boldsymbol{E}_i \mathbf{x}_i$. Given a function $T$, $\widehat{\mathbf{x}}$ computed as in Eq. 5 recovers the Rand-$k$-Spatial estimator.*

*Proof.* If client $i$ applies $\boldsymbol{E}_i \in \mathbb{R}^{k \times d}$ as the random matrix to encode $\mathbf{x}_i$ in Rand-Proj-Spatial, by Eq. 5, client $i$'s encoded vector is now

$$\hat{\mathbf{x}}_i^{\text{(Rand-Proj-Spatial)}} = \bar{\beta}\Big(T(\sum_{i=1}^n \boldsymbol{E}_i^T \boldsymbol{E}_i)\Big)^\dagger \boldsymbol{E}_i^T \boldsymbol{E}_i \mathbf{x}_i \tag{29}$$

Notice $\boldsymbol{E}_i^T \boldsymbol{E}_i$ is a diagonal matrix, where the $j$-th diagonal entry is 1 if coordinate $j$ of $\mathbf{x}_i$ is chosen. Hence, $\boldsymbol{E}_i^T \boldsymbol{E}_i \mathbf{x}_i$ can be viewed as choosing $k$ coordinates of $\mathbf{x}_i$ without replacement, which is exactly the same as Rand-$k$-Spatial's (and Rand-$k$'s) encoding procedure.

Notice $\sum_{i=1}^n \boldsymbol{E}_i^T \boldsymbol{E}_i$ is also a diagonal matrix, where the $j$-th diagonal entry is exactly $M_j$, i.e. the number of clients who selects the $j$-th coordinate as in Rand-$k$-Spatial [12]. Furthermore, notice $\Big(T(\sum_{i=1}^n \boldsymbol{E}_i^T \boldsymbol{E}_i)\Big)^\dagger$ is also a diagonal matrix, where the $j$-th diagonal entry is $\frac{1}{T(M_j)}$, which recovers the scaling factor used in Rand-$k$-Spatial's decoding procedure.

Rand-Proj-Spatial computes $\bar{\beta}$ as $\bar{\beta}\mathbb{E}\Big[\Big(T(\sum_{i=1}^n \boldsymbol{E}_i^T \boldsymbol{E}_i)\Big)^\dagger \boldsymbol{E}_i^T \boldsymbol{E}_i \mathbf{x}_i\Big] = \mathbf{x}_i$. Since $\Big(T(\sum_{i=1}^n \boldsymbol{E}_i^T \boldsymbol{E}_i)\Big)^\dagger$ and $\boldsymbol{E}_i^T \boldsymbol{E}_i \mathbf{x}_i$ recover the scaling factor and the encoding procedure of Rand-$k$-Spatial, and $\bar{\beta}$ is computed in exactly the same way as Rand-$k$-Spatial does, $\bar{\beta}$ will be exactly the same as in Rand-$k$-Spatial.

Therefore, $\hat{\mathbf{x}}_i^{\text{(Rand-Proj-Spatial)}}$ in Eq. 29 with $\boldsymbol{E}_i$ as the random matrix at client $i$ recovers $\hat{\mathbf{x}}_i^{\text{(Rand-}k\text{-Spatial)}}$. This implies Rand-Proj-Spatial recovers Rand-$k$-Spatial in this case. $\square$

# D  Additional Experiment Details and Results

**Implementation.** All experiments are conducted in a cluster of 20 machines, each of which has 40 cores. The implementation is in `Python`, mainly based on `numpy` and `scipy`. All code used for the experiments can be found at `https://github.com/11hifish/Rand-Proj-Spatial`.

**Data Split.** For the non-IID dataset split across the clients, we follow [62] to split `Fashion-MNIST`, which is used in distributed power iteration and distributed $k$-means. Specifically, the data is first sorted by labels and then divided into $2n$ shards with each shard corresponding to the data of a particular label. Each client is then assigned 2 shards (i.e., data from 2 classes). However, this approach only works for datasets with discrete labels (i.e. datasets used in classification tasks). For the other dataset `UJIndoor`, which is used in distributed linear regression, we first sort the dataset by the ground truth prediction and then divides the sorted dataset across the clients.

## D.1  Additional experimental results

For each one of the three tasks, distributed power iteration, distributed $k$-means, and distributed linear regression, we provide additional results when the data split is IID across the clients for smaller $n, k$ values in Section D.1.1, and when the data split is Non-IID across the clients in Section D.1.2. For the Non-IID case, we use the same settings (i.e. $n, k, d$ values) as in the IID case.

**Discussion.** For smaller $n, k$ values compared to the data dimension $d$, there is less information or less correlation from the client vectors. Hence, both Rand-$k$-Spatial and Rand-Proj-Spatial perform better as $nk$ increases. When $n, k$ is small, one might notice Rand-Proj-Spatial performs worse than Rand-$k$-Wangni in some settings. However, Rand-$k$-Wangni is an *adaptive* estimator, which optimizes the sampling weights for choosing the client vector coordinates through an iterative process. That means Rand-$k$-Wangni requires more computation from the clients, while in practice, the clients often have limited computational power. In contrast, our Rand-Proj-Spatial estimator is *non-adaptive* and the server does more computation instead of the clients. This is more practical since the central server usually has more computational power than the clients in applications like FL. See the introduction for more discussion.

In most settings, we observe the proposed Rand-Proj-Spatial has a better performance compared to Rand-$k$-Spatial. Furthermore, as one would expect, both Rand-$k$-Spatial and Rand-Proj-Spatial perform better when the data split is IID across the clients since there is more correlation among the client vectors in the IID case than in the Non-IID case.

### D.1.1  More results in the IID case

**Distributed Power Iteration and Distributed $K$-Means.** We use the `Fashion-MNIST` dataset for both distributed power iteration and distributed $k$-means, which has a dimension of $d = 1024$. We consider more settings for distributed power iteration and distributed $k$-means here: $n = 10, k \in \{5, 25, 51\}$, and $n = 50, k \in \{5, 10\}$.

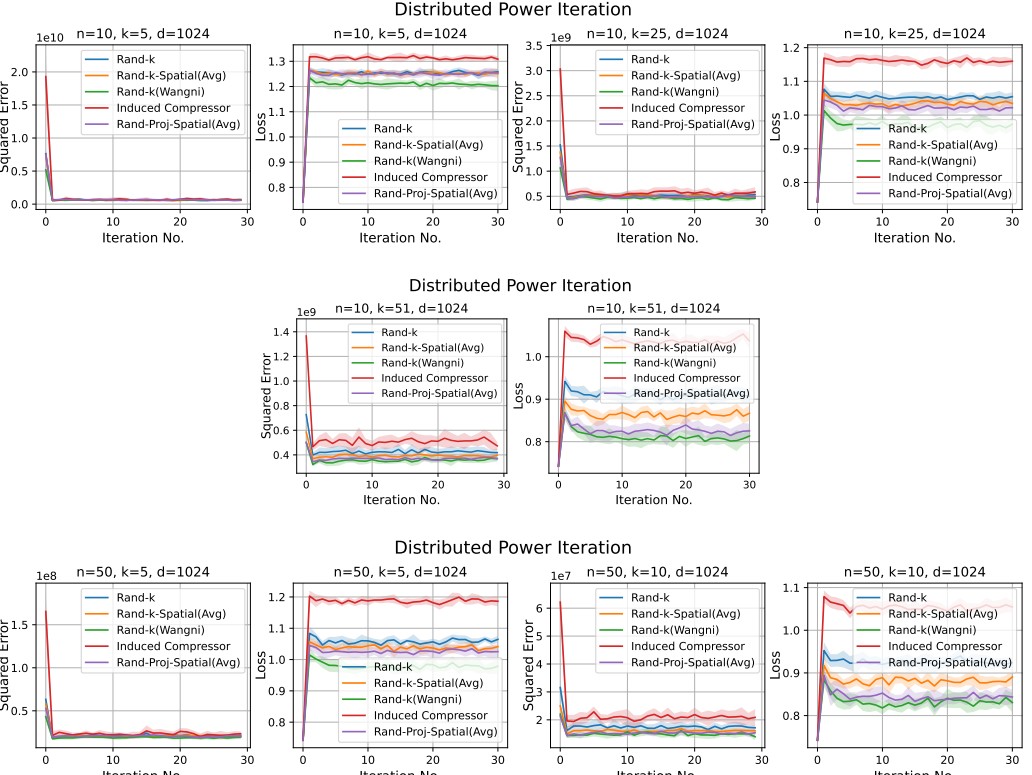

Figure 8: More results of distributed power iteration on `Fashion-MNIST` (IID data split) with $d = 1024$ when $n = 10$, $k \in \{5, 25, 51\}$ and when $n = 50$, $k \in \{5, 10\}$.

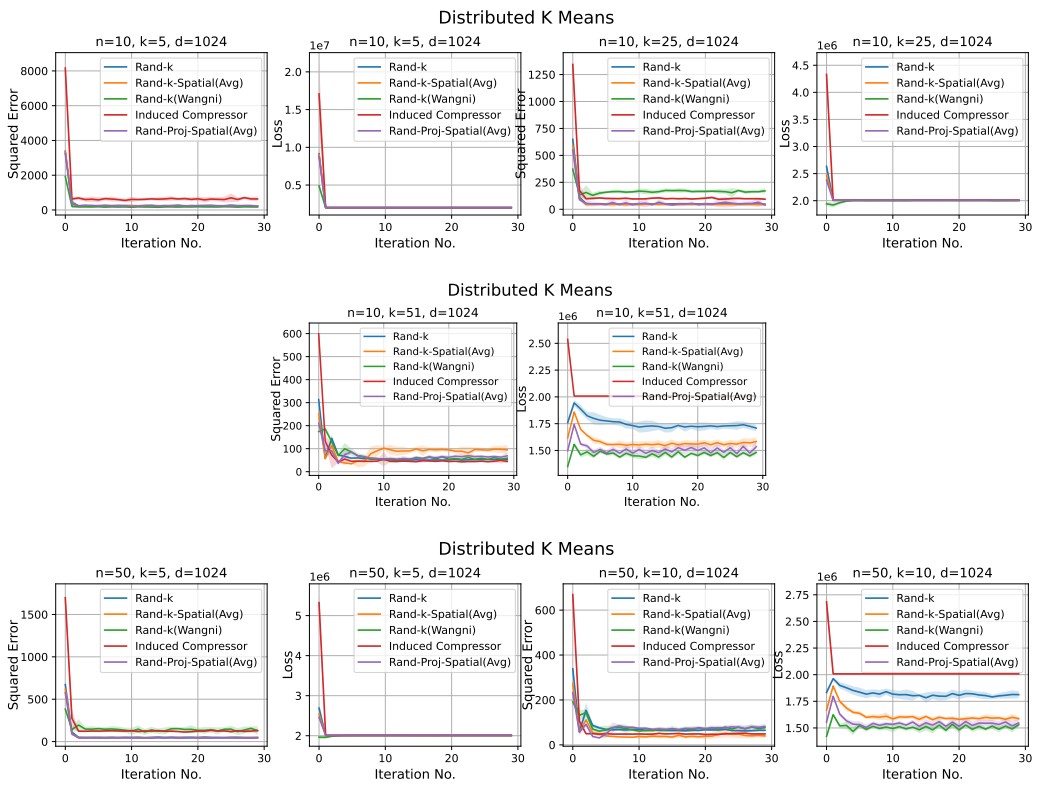

Figure 9: More results on distributed $k$-means on `Fashion-MNIST` (IID data split) with $d = 1024$ when $n = 10$, $k \in \{5, 25, 51\}$ and when $n = 50$, $k \in \{10, 51\}$.

**Distributed Linear Regression.** We use the `UJIndoor` dataset distributed linear regression, which has a dimension of $d = 512$. We consider more settings here: $n = 10, k \in \{5, 25\}$ and $n = 50, k \in \{1, 5\}$.

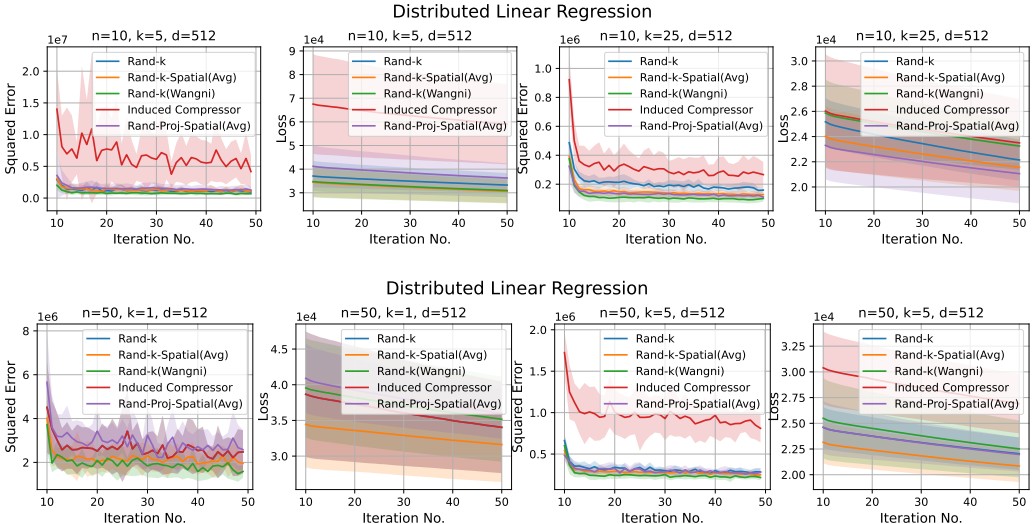

Figure 10: More results of distributed linear regression on `UJIndoor` (IID data split) with $d = 512$, when $n = 10, k \in \{5, 25\}$ and when $n = 50, k \in \{1, 5\}$. Note when $k = 1$, the Induced estimator is the same as Rand-$k$.

### D.1.2  Additional results in the Non-IID case

In this section, we report results when the dataset split across the clients are Non-IID, using the same datasets as in the IID case. We choose exactly the same set of $n, k$ values as in the IID case.

**Distributed Power Iteration and Distributed $K$-Means.**  Again, both distributed power iteration and distributed $k$-means use the `Fashion-MNIST` dataset, with a dimension $d = 1024$. We consider the following settings for both tasks: $n = 10, k \in \{5, 25, 51, 102\}$ and $n = 50, k \in \{5, 10, 20\}$.

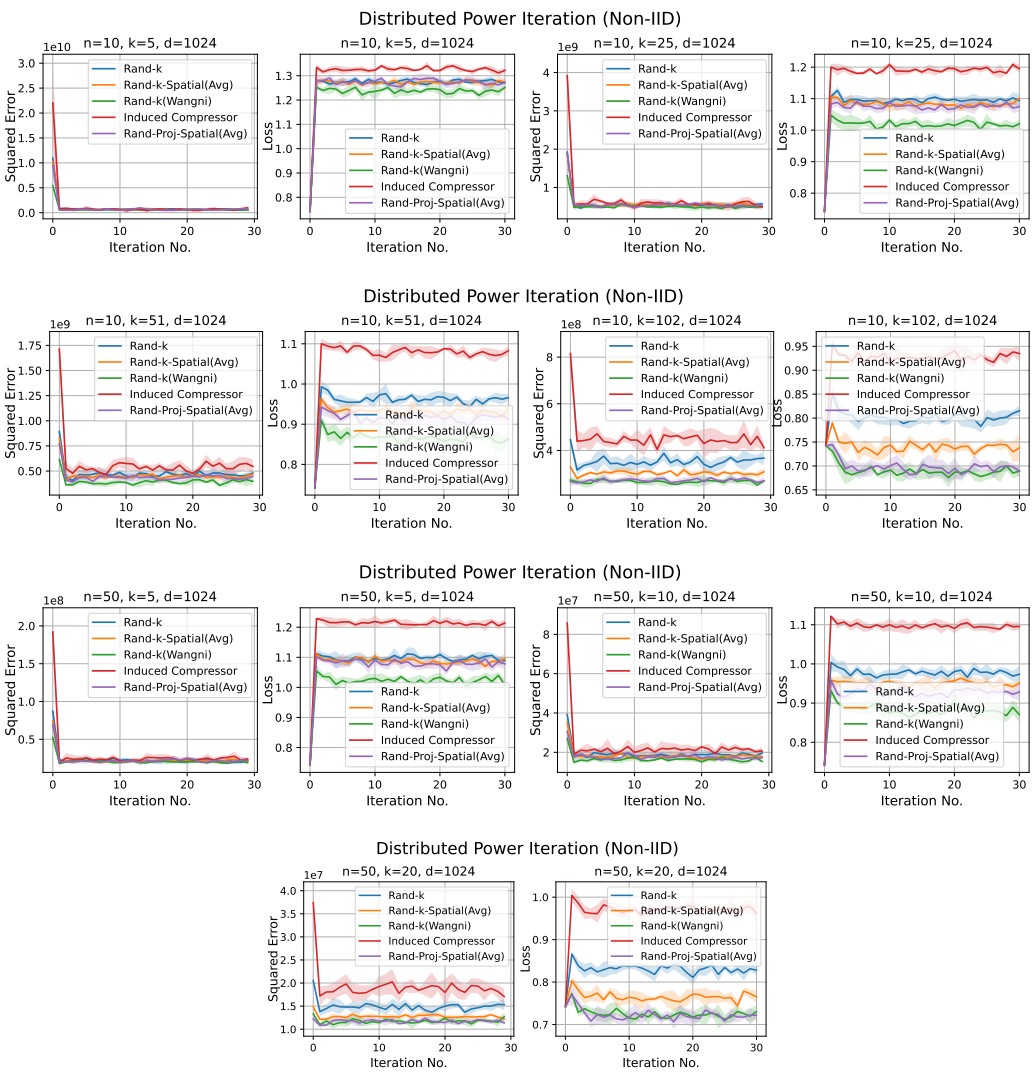

Figure 11: Results of distributed power iteration when the data split is Non-IID. $n = 10, k \in \{5, 25, 51, 102\}$ and $n = 50, k \in \{5, 10, 20\}$.

**Distributed Linear Regression.**  Again, we use the `UJIndoor` dataset for distributed linear regression, which has a dimension $d = 512$. We consider the following settings: $n = 10, k \in \{5, 25, 50\}$ and $n = 50, k \in \{1, 5, 50\}$.

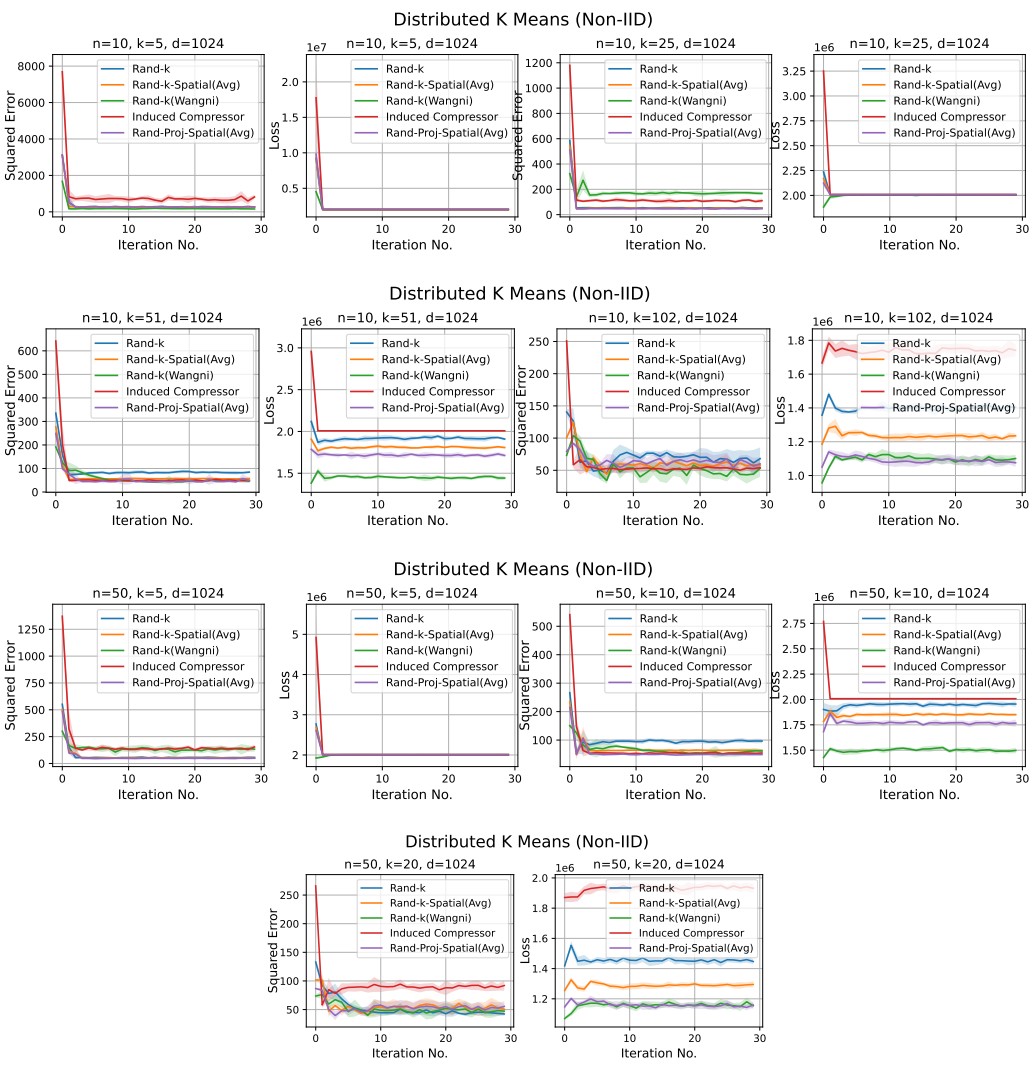

Figure 12: Results of distributed $k$-means when the data split is Non-IID. $n = 10, k \in \{5, 25, 51, 102\}$ and $n = 50, k \in \{5, 10, 20\}$.

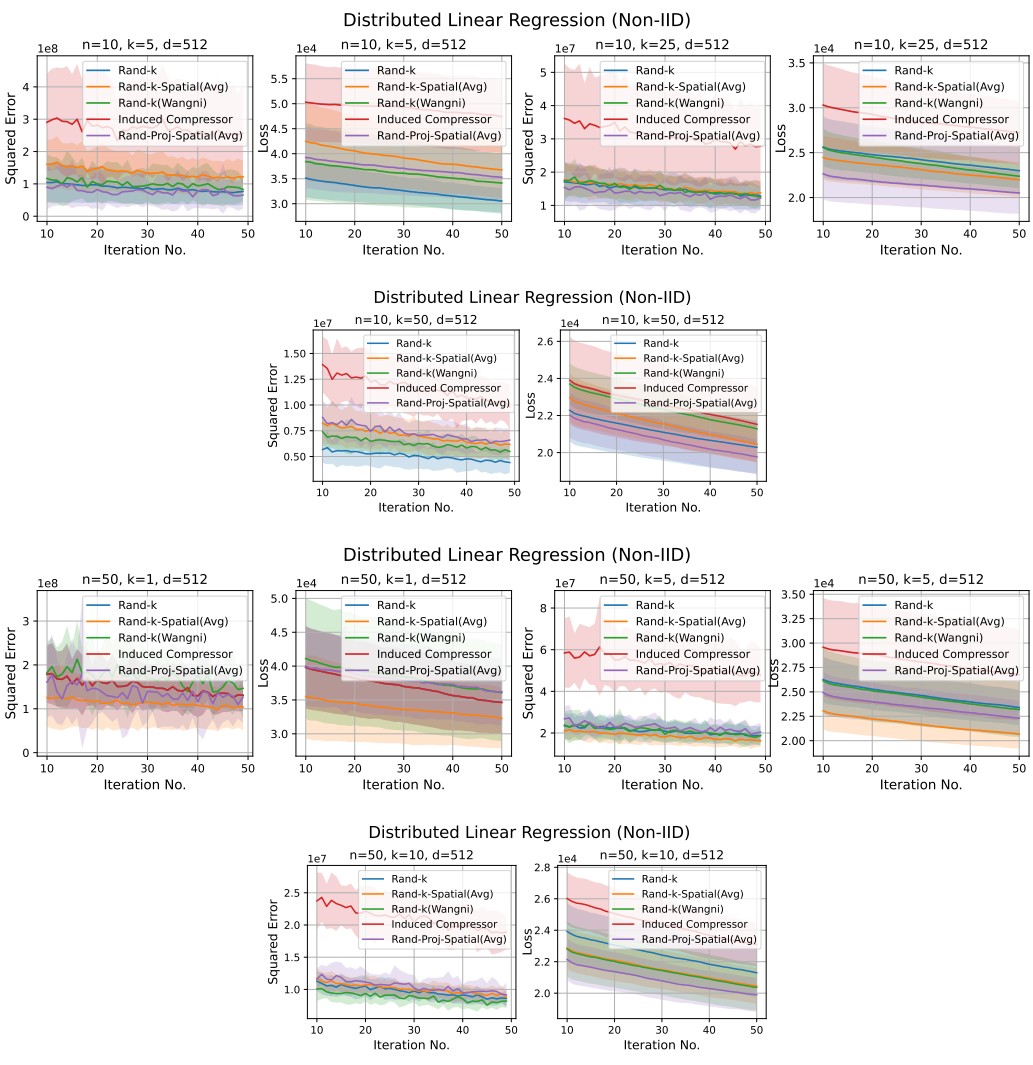

Figure 13: Results of distributed linear regression when the data split is Non-IID. $n = 10, k \in \{5, 25, 50\}$ and $n = 50, k \in \{1, 5, 50\}$.

