_{\boldsymbol{U}=\Phi} \Pr[\boldsymbol{U}=\Phi]\mathbb{E}[\boldsymbol{U}\Lambda^{\dagger}\Lambda\boldsymbol{U}^T \mid \boldsymbol{U}=\Phi]\right]\mathbf{x}$$

$$= \bar{\beta}\left[\sum_{\boldsymbol{U}=\Phi} \Pr[\boldsymbol{U}=\Phi]\boldsymbol{U}\mathbb{E}[\Lambda^{\dagger}\Lambda \mid \boldsymbol{U}=\Phi]\boldsymbol{U}^T\right]\mathbf{x}$$

$$\overset{(a)}{=} \bar{\beta}\left[\sum_{\boldsymbol{U}=\Phi} \Pr[\boldsymbol{U}=\Phi]\boldsymbol{U}\mathbb{E}[\text{diag}(\mathbf{m}) \mid \boldsymbol{U}=\Phi]\boldsymbol{U}^T\right]\mathbf{x}$$

$$\overset{(b)}{=} \bar{\beta}\sum_{\boldsymbol{U}=\Phi} \Pr[\boldsymbol{U}=\Phi]\left[\boldsymbol{U}\Big((1-\delta)\frac{nk}{d}\boldsymbol{I}_d + \sum_{c=k}^{nk-1}\delta_c\frac{c}{d}\boldsymbol{I}_d\Big)\boldsymbol{U}^T\right]\mathbf{x}$$

$$= \bar{\beta}\Big[(1-\delta)\frac{nk}{d} + \sum_{c=k}^{nk-1}\delta_c\frac{c}{d}\Big]\mathbf{x}$$

$$\Rightarrow \bar{\beta} = \frac{d}{(1-\delta)nk + \sum_{c=k}^{nk-1}\delta_c c} \tag{20}$$

where in $(a)$, $\mathbf{m} \in \mathbb{R}^d$ such that

$$\mathbf{m}_i = \begin{cases} 1 & \text{if } \Lambda_{jj} > 0 \\ 0 & \text{else.} \end{cases}$$

Also, by construction of $\boldsymbol{S}$, $\text{rank}(\text{diag}(\mathbf{m})) \leq nk$. Further, $(b)$ follows by symmetry across the $d$ dimensions.

Since $\delta k \leq \sum_{c=k}^{nk-1}\delta_c c \leq \delta(nk-1)$, there is

$$\frac{d}{(1-\delta)nk + \delta(nk-1)} \leq \bar{\beta} \leq \frac{d}{(1-\delta)nk + \delta k} \tag{21}$$

**Computing the MSE.** Next, we use the value of $\bar{\beta}$ in Eq. 20 to compute MSE.

$$MSE(\text{Rand-Proj-Spatial(Max)}) = \mathbb{E}[\|\widehat{\mathbf{x}}^{(\text{Rand-Proj-Spatial(Max)})} - \bar{\mathbf{x}}\|_2^2] = \mathbb{E}[\|\bar{\beta}\boldsymbol{S}^{\dagger}\boldsymbol{S}\mathbf{x} - \mathbf{x}\|_2^2]$$

$$= \bar{\beta}^2\mathbb{E}[\|\boldsymbol{S}^{\dagger}\boldsymbol{S}\mathbf{x}\|_2^2] + \|\mathbf{x}\|_2^2 - 2\Big\langle \bar{\beta}\mathbb{E}[\boldsymbol{S}^{\dagger}\boldsymbol{S}\mathbf{x}], \mathbf{x}\Big\rangle$$

$$= \bar{\beta}^2\mathbb{E}[\|\boldsymbol{S}^{\dagger}\boldsymbol{S}\mathbf{x}\|_2^2] - \|\mathbf{x}\|_2^2 \qquad \text{(Using unbiasedness of } \widehat{\mathbf{x}}^{(\text{Rand-Proj-Spatial(Max)})})$$

$$= \bar{\beta}^2\mathbf{x}^T\mathbb{E}[\boldsymbol{S}^T(\boldsymbol{S}^{\dagger})^T\boldsymbol{S}^{\dagger}\boldsymbol{S}]\mathbf{x} - \|\mathbf{x}\|_2^2. \tag{22}$$

Using $\boldsymbol{S}^{\dagger} = \boldsymbol{U}\Lambda^{\dagger}\boldsymbol{U}^T$,

$$\mathbb{E}[\boldsymbol{S}^T(\boldsymbol{S}^{\dagger})^T\boldsymbol{S}^{\dagger}\boldsymbol{S}] = \mathbb{E}[\boldsymbol{U}\Lambda\boldsymbol{U}^T\boldsymbol{U}\Lambda^{\dagger}\boldsymbol{U}^T\boldsymbol{U}\Lambda^{\dagger}\boldsymbol{U}^T\boldsymbol{U}\Lambda\boldsymbol{U}^T]$$

$$= \mathbb{E}[\boldsymbol{U}\Lambda(\Lambda^{\dagger})^2\Lambda\boldsymbol{U}^T]$$

$$= \sum_{\boldsymbol{U}=\Phi} \boldsymbol{U}\mathbb{E}[\Lambda(\Lambda^{\dagger})^2\Lambda]\boldsymbol{U}^T \cdot \Pr[\boldsymbol{U}=\Phi]$$

$$= \sum_{\boldsymbol{U}=\Phi} \boldsymbol{U}\Big[(1-\delta)\frac{nk}{d}\boldsymbol{