# OpenReview forum: "Correlation Aware Sparsified Mean Estimation Using Random Projection"
_NeurIPS.cc/2023/Conference — NeurIPS 2023 poster_

### Official Review · Reviewer_R72T · 2023-06-26

**Soundness:** 3 good
**Presentation:** 2 fair
**Contribution:** 2 fair
**Rating:** 6
**Confidence:** 4

**Summary:**

The authors propose a compression algorithm that optimizes the accuracy in a setting where each client can send $k+O(1)\ll d$ values to the server. The algorithm leverages random projections and proposes a way to make the resulting estimate unbiased, which is valuable for DME.

**Strengths:**

+ DME is an important and well-studied problem, so advances are welcomed.

+ Leveraging correlations between clients' vectors leads to significant improvement in accuracy in some cases.

+ The setting of $n\cdot k \ll d$ is challenging and interesting.

**Weaknesses:**

- The decoding time at the server is not presented, may be prohibitively high in some cases, and must be discussed upfront.

- The authors compare only with Rand-$k$ and Rand-$k$-Spatial. There are other solutions that leverage correlations between clients' vectors (e.g., New Bounds For Distributed Mean Estimation and Variance Reduction, ICLR 2021) and it is unclear to me whether the Rand-$*$ approaches are better or what are the tradeoffs. The authors do not cite the paper or compare with such approaches even qualitatively.

- It is unclear to me that these approaches, in which each client send $k$ floats are better than quantization methods that allow sending, e.g., $32k$ values with one bit each. One example that can use as a comparison point is [33].

- The authors discuss a case where $n\cdot k \ll d$; however, probably due to time complexity, even running it for $d \gg 1$ seems unlikely. The evaluation only shows $d\le 1024$ and no runtime numbers are provided.

**Questions:**

* Do you view Rand-Proj-Spatial as being orthogonal to quantization techniques? For example, what would be the impact of quantizing the $k$ values sent by each worker?

* Am I correct to understand that the server's decoding requires $\Theta(d^2\cdot k\cdot n)$ time? If not, what is it?

* You suggest that each client $i$ would use a different projection $G_i$. This may be important for the error, but also requires the server to compute the pinverse of each client's message separately. Instead, consider using a single projection $G$ that all clients use, and thus the server could compute a single pinverse after summing the messages. How would the error bounds change in such an implementation?



**Limitations:**

n/a.

---

> ### Author Rebuttal · Authors · 2023-08-10
>
> W2 \& W3 \& Q1: See common response $\textbf{Quantization vs. Sparsification}$.
>
> W1 \& W4 \& Q2: See common response $\textbf{Computation Time}$. Also, we included additional results on comparing the encoding and decoding wall-clock time of different estimators (see the pdf attachment in common response). We will make the computational time for decoding clearer and stress it as a limitation.
>
> Q3: As we discussed in Appendix A.1, applying the same $\mathbf{G}$ random projection across the clients does not lead to any improvement of the MSE compared to that of Rand-$k$. The reason is that rotation does not change the $\ell_2$ norm.
> Furthermore, we note that when $\mathbf{G_i}$ are different for each client, it is not true that our Rand-Proj-Spatial "requires the server to compute the p-inverse of each client's message separately".
> Rand-Proj-Spatial only requires the server to compute p-inverse once, i.e., the p-inverse of $\mathbf{S} = \sum_{i=1}^{n}\mathbf{G}_i^T \mathbf{G}_i$.

---

> > ### Comment · Reviewer_R72T · 2023-08-10
> >
> > Thank you for the answers and the runtime experiments.
> >
> > I will be raising my rating to weak accept but no higher as I view the decoding time as a significant bottleneck of the current approach, which limits its practicality.
> >
> > I encourage the authors to explore methods that could give a lower decoding time.

---

> > > ### Author Response · Authors · 2023-08-16
> > >
> > > We would like to thank the reviewer for providing valuable feedback. Yes, improving the decoding time of the current approach and deriving the optimal tradeoffs between computation time, communication cost and the error for sparsification techniques with correlation information can be interesting future directions.

---

### Official Review · Reviewer_2UxF · 2023-07-02

**Soundness:** 3 good
**Presentation:** 3 good
**Contribution:** 3 good
**Rating:** 5
**Confidence:** 4

**Summary:**

This paper studies the distributed mean estimation (DME) problem. In particular, the paper proposes a new DME technique called Rand-Proj-Spatial. In  Rand-Proj-Spatial, each client uses SRHT for dimensionality reduction and sends the transformed lower-dimensional vector to the server. The server then recovers the mean estimate by computing a formula derived from an optimization problem designed to minimize the MSE given the client transforms and considering possible correlations among client vectors. Rand-Proj-Spatial is also unbiased, which is desired in the context of DME.

The main contribution of Rand-Proj-Spatial is that it both improves with increased correlation among client vectors and utilizes SRHT for dimensionality reduction instead of subsampling (like rand-k), achieving lower MSE than previous rand-k-based DME techniques.


**Strengths:**

1.	The new DME technique improves upon previous rand-k-based DME techniques.
2.	The reconstruction technique is interesting and not symmetric, putting the computational burden on the server.
3.	Rand-Proj-Spatial leverages possible correlations among client vectors which can be expected and useful in some distributed scenarios.


**Weaknesses:**

1.	The worst-case MSE of Rand-Proj-Spatial appears to be $O(d/n)$. Namely, unless $k = \Theta(d)$, the MSE of Rand-Proj-Spatial grows linearly with the dimension of the problem and may cause an asymptotic increase in the number of required optimization rounds in many (S)GD-based scenarios (e.g., neural network training).

2.	No comparison to strong DME baselines, e.g., [1][2][3]: In the paper’s evaluation, the authors consider scenarios with d=1024,  21-51 clients, and k=4-40. These regimes can be readily compared to, e.g., [1][2][3] with 1-2 bits per coordinate. Moreover,  some of these support sub-bit regimes as well.

3.	Insufficient evaluation: the evaluation considers low-dimensional (mostly convex) scenarios. It would be more convincing to demonstrate the advantage of Rand-Proj-Spatial over, e.g., neural networks with sufficient dimension and demonstrate that Rand-Proj-Spatial results in better performance than existing DME techniques considering all aspects of the algorithm (i.e., error-to-bandwidth tradeoff and computational overhead).

4.	The encoding and decoding time of Rand-Proj-Spatial are not evaluated and compared to previous DME techniques. These times are of major importance in practical scenarios.

[1] Suresh, Ananda Theertha, et al. "Distributed mean estimation with limited communication." International conference on machine learning. PMLR, 2017.

[2] Davies, Peter, et al. "New Bounds For Distributed Mean Estimation and Variance Reduction." International Conference on Learning Representations, 2021.

[3] Vargaftik, Shay, et al. "Eden: Communication-efficient and robust distributed mean estimation for federated learning." International Conference on Machine Learning. PMLR, 2022.


**Questions:**

Can the authors address the concerns pointed out in weaknesses (1)-(4)?

**Limitations:**

The authors point out a possible direction for future work. I would suggest adding disclaimers about the computational overhead and the asymptotic error-bandwidth tradeoffs compared to existing DME techniques.

---

> ### Author Rebuttal · Authors · 2023-08-10
>
>
> W1: The worst case MSE of prior works on sparsification techniques, e.g. [1] and [2],
> are also on the order of $O(d/n)$.
> Note we did not claim that our approach improves the asymptotic accuracy bounds. Just like [1], our method focuses on utilizing practically available side information to improve communication cost-estimation accuracy tradeoffs.
>
> ------
>
> [1] Divyansh Jhunjhunwala, Ankur Mallick, Advait Harshal Gadhikar, Swanand Kadhe, and Gauri Joshi. ``Leveraging spatial and temporal correlations in sparsified mean estimation''. NeurIPS 2021.
>
> [2] Jakub Konecny and Peter Richtárik. ``Randomized distributed mean estimation: Accuracy vs. communication''. Frontiers in Applied Mathematics and Statistics, 4:62, 2018.
>
> ------
>
> W2:
> See common response $\textbf{Quantization vs. Sparsification}$.
>
> W3 \& W4: We included additional results on comparing the encoding and decoding wall-clock time of different estimators (see the pdf attachment in common response). We note our experiment setting mostly follows that of the prior work [1]. One potential way to make the decoding process more efficient is to divide the dimension $d$ into chunks, encode and decode each chunk. This is similar to layer-wise compression of NNs. We found performing Rand-$k$ on NN needs a bit of time to do hyperparameter tuning, or the optimization diverges. We will make the computational time for decoding clearer and stress it as a limitation.

---

> > ### Comment · Reviewer_2UxF · 2023-08-13
> >
> > Thank you for your answers and the new experiments. I will be raising my score from 4 to 5. Stressing the limitation of the decoding time is important as it limits the practicality of this solution. Also, giving a broader introduction to the DME problem to include a unified view of sparsification and quantization DME solutions may help the reader to better understand the positioning of the work.

---

> > > ### Author Response · Authors · 2023-08-16
> > >
> > > We would like to thank the reviewer for providing valuable feedback. Yes, we will stress the limitation of the decoding time and give a broader overview of sparsification and quantization techniques to better position this work.

---

### Official Review · Reviewer_uyDa · 2023-07-06

**Soundness:** 3 good
**Presentation:** 2 fair
**Contribution:** 3 good
**Rating:** 7
**Confidence:** 4

**Summary:**

This work considers the problem of distributed mean estimation, wherein each node in a set of distributed nodes contains a vector, and the goal of the parameter server is to estimate the mean of those vectors. Unlike some other works, no distributional assumption is assumed over the vectors, and the error metric is the mean squared error between the true mean and the estimated mean. Each node sends a compressed version of its vector to the parameter server, and this is the sole source of estimation error. Any randomness in the estimation error arises from the randomness of the compression algorithm used at the nodes. This setup is motivated from distributed optimization / federated learning setups where the mean of the gradients is estimated at the parameter server at every iteration of the optimization algorithm. Since modern machine learning tasks deal with very high dimensional models and gradients, it becomes necessary to compress them before sending it to the parameter server, so as not to overwhelm the communication requirements between nodes and parameter server (which is often a bottleneck).

This particular work focuses on the effect of correlation between the vectors at different nodes, and considers exploiting this correlation information to design better compressors that achieve a lower mean squared error than correlation-agnostic compressors. Prior work (Jhunjhunwala et. al. (2021)) proposed the **rand-$k$ spatial family** of mean estimators that used correlation information. They did so by first compressing the vector at each node using a rand-$k$ compressor. Rand-$k$ essentially subsamples $k$ coordinates out of a $d$-dimensional vector to get a compressed $k$-dimensional vector, zeroing out the remaining coordinated and yielding a compression factor of $\frac{d}{k}$. These compressed vectors are transmitted to the parameter server, which then decodes the mean by appropriately scaling the received vectors that ensures an unbiased estimate of the mean. When correlation information is available, the server can decode the received vectors with a different scaling mechanism, yielding a smaller estimation error.

The primary contribution of this work is a modification of the encoding strategy to incorporate random projections in the encoding step at the clients. Instead of directly applying rand-$k$ to the vector, the vector is first projected on a random subspace (more specifically, the range space of a subsampled randomized Hadamard matrix) and then subsequently rand-$k$ compressor is applied.

Following this idea, the authors in this work propose a correlation-aware encoding and decoding strategy for distributed mean estimation under communication constraints, analytically derive upper bounds on the estimation error of their proposed scheme, and finally, numerically evaluate the performance of their scheme relative to existing benchmarks. Please let me know if I have missed anything in my understanding of the contributions of the paper. I would be more than happy to rectify any misunderstandings on my end.

**Strengths:**

The following are the primary contributions of the paper:

1. The authors propose the *rand-proj-spatial* family of mean estimators as a generalization of the previously proposed *rand-$k$-spatial* estimators, that uses subsampled randomized Hadamard transform to first project the vectors onto a random subspace and subsequently apply a rand-$k$ compressor with appropriate decoding at the server.

2. Several numerical experiments on distributed power iteration, distributed $k$-means, and distributed linear regression have been performed to demonstrate the superiority of the proposed strategy over existing correlation-aware and correlation agnostic strategies.

One of the interesting observations made by the authors is Appendix A.1, which states that applying random projection + rand-$k$ directly to the vectors may not yield any benefit. In other words, if we apply the $\frac{d}{k}$ scaling individually for each vector with the hope of obtaining an unbiased estimate for each nodes' vector at the server, the expected estimation error is the same as that obtained by rand-$k$ compression without the projection. However, lower MSE can be obtained if the vectors received from each of the nodes are jointly decoded as is done in the *rand-proj-spatial* estimator as proposed in eq. (5) of the paper. At a first glance, this seems a little counter-intuitive since (as the authors point out in the introduction), the motivation to take random projections stems from the idea that random projections equalize the coordinate magnitudes.

However, a careful study reveals that when the error metric is the $\ell_2$-estimation error, random projections do not really help, because the $\ell_2$-norm of the vectors approximately remains preserved when an orthonormal projection is applied. However, I do believe the benefit will be apparent if other error metrics are considered, such as the $\ell_{\infty}$ estimation error (more remarks on why this might be the case in the *limitations* section).

**Weaknesses:**

I have one critical concern which is the fact that the authors have titled their paper *Correlation Aware Distributed Vector Mean Estimation* as a very generic and broad approach, whereas the contributions of the paper are focused on exploiting correlation for communication compression for a very specific class of rand-$k$ compression strategies, and it is not immediately obvious how this correlation-aware compression strategy can be extended to other classes of compressor such as top-$k$ (instead of rand-$k$), which is a biased compressor -- but is very popular and has been extensively used for distributed optimization / federated learning applications. For example, the paper by Suresh et. al. (2022) (reference [10]) also exploits correlation for quantization purposes. Please note that this is not a drawback or a comment on the technical contributions of the paper. However, the title can be slightly misleading in a sense since the paper does not focus on the benefits of correlation beyond rand-$k$ compression. For instance, correlation can be exploited for client selection, or for designing correlation-aware private mean estimation algorithms. It is my personal opinion and would be highly appreciated if the authors chose a more descriptive title.

In addition to this, the *rand-proj-spatial* family of estimators proposed in this paper is also a very specific design, and there is no indication if it is the best strategy. In other words, given that our underlying task is communication compression, is *rand-proj-spatial* the best way to exploit correlation? I understand this is not an easy question to answer since lower bounds on the estimation error have not been derived in this paper. However, when authors introduce the optimization problem in eq. (4), it seemed to me that this optimization problem was formulated with a solution in mind -- a solution which is analogous (and a generalization) to the prior work on *rand-$k$ spatial* family of estimators. A natural question to ask here is why the problem was not initially formulated more generally as follows:

$\hat{\mathbf{x}} = argmin_{\mathbf{W}, \mathbf{x}} \mathbb{E}\left\lVert \overline{\mathbf{x}} - \sum_{i=1}^{n}\mathbf{W}_i\mathbf{G}_i\mathbf{x}_i \right\rVert_2^2$

where $\overline{\mathbf{x}} = \frac{1}{n}\sum_{i=1}^{n}\mathbf{x}_i$, and $\mathbf{W} = (\mathbf{W}_1, \ldots, \mathbf{W}_n)$ are the individual decoding matrices over which optimization is done as well. Furthermore, one can solve this optimization problem subject to the unbiasedness constraint and potentially come up with a better decoding strategy than the one proposed in this paper? It would be appreciated if the authors discussed how (or under what assumptions) the optimization problem of eq. (4) relates to the this general formulation above, and what are the challenges that prevent one from solving the above general optimization problem directly. Moreover, in the above formulation, the encoding strategy is still the same, i.e., rand-proj + rand-$k$ -- one could also replace the encoding function by a more general class of (potentially biased) contractive compressors (such as top-$k$) and come up with a more very holistic approach to the problem of *correlation aware compression for distributed mean estimation* (which would justify this title).

Please note once again that this is not a weakness on the technical contributions of the paper as such -- just a suggestion that I genuinely believe would help highlight the contributions of the paper in a better context.

**Questions:**

I have a few questions and suggestions for the authors, and would highly appreciate if they would take them into consideration. I would be more than happy to re-evaluate my review contingent on their response. The following are some of my concerns:

The idea of random projection + compression like rand-$k$ is not novel by itself (although it's implications in the presence of correlation as explored in this paper are, to the best of my knowledge). Accordingly, there are some very relevant references missing from the related works section and it would be highly appreciated if the authors included those:

1. R. Saha, M. Pilanci and A. J. Goldsmith, "Efficient Randomized Subspace Embeddings for Distributed Optimization Under a Communication Budget," in IEEE Journal on Selected Areas in Information Theory, vol. 3, no. 2, pp. 183-196, June 2022, doi: 10.1109/JSAIT.2022.3198412 (This work adopts a unified approach for studying the effects of random transforms on compressors -- it poses randomized Hadamard transform as a computationally efficient relaxation of Democratic (Kashin) representations, and also studies rand-$k$ compression (among others) with both kashin + near-kashin -- eventually showing that they attain information-theoretically optimal convergence rates for optimization).

2. Mher Safaryan and others, Uncertainty principle for communication compression in distributed and federated learning and the search for an optimal compressor, Information and Inference: A Journal of the IMA, Volume 11, Issue 2, June 2022, Pages 557–580, https://doi.org/10.1093/imaiai/iaab006 (This work studies a general class of contractive compressors which includes random-projections based compressors -- they focus on Kashin compression and study different variants of accelerated distributed optimization algorithms).

These properties are not specific to subsampled Randomized Hadamard transforms, -- but extend broadly to the class of randomized matrices that belong to Johnson-Lindenstrauss embeddings and/or follow Restricted Isometry Property (RIP). These results might perhaps be helpful to the authors in deriving bounds on MSE for *rand-proj-spatial* for other classes of matrices (such as random Haar orthonormal). Once again, this is not a critical drawback, but a suggestion that will help place the work better in context with respect to existing literature.

**Limitations:**

I have some other minor concerns and would be happy if the authors addressed / discussed them:

1. The authors mention on line $138$ *"Each client also sends a random seed to the server, which conveys the subspace information, and can usually be communicated using a negligible amount of bits."* -- Is this really negligible? Can $\mathbf{G}$ be conveyed between the nodes and server with less than $d$ bits of information? Since each node shares an independently sampled random matrix, can correlation help improve this shared randomness generation?

2. The statement of Theorem 4.3 is a high probability result but the equation (9) holds in expectation -- this is slightly weird. It would be better if the authors stated the result purely as a high probability statement or purely in terms of expectation (the latter might be simpler since the failure probability is $o(1)$). If this is not possible, they should atleast explicitly specify in the statement of the theorem which random variables is the expectation computed over, and which random variables is the high probability bound over? Moreover, $o(1)$ is not explicitly defined -- is it with respect to dimension / number of nodes -- please specify?

It would also be highly if the authors had a separate limitations sections that discuss these and other concerns raised in *weaknesses* and *questions*, which are my primary concerns. I would be more than happy to re-evaluate my assessment if these concerns are addressed satisfactorily.

---

> ### Author Rebuttal · Authors · 2023-08-10
>
> W1: We indeed had a lot of discussions on the title before the submission. It is hard to give a precise yet succinct title. Other possible title candidates are "Projection Based Correlation Aware Distributed Vector Mean Estimation", "Unbiased Sparsification Induced Distributed Vector Mean Estimation with Correlation", etc. We are happy to take suggestions from the reviewers.
>
> W2:
> The proposed optimization problem
> might need a bit clarification: Where is the variable $\mathbf{x}$ in the problem?
>
> One thing to note is that to utilize cross-client correlation information, we need to design a decoding scheme that uses all client's information in decoding. For example, in Rand-$k$-Spatial [1], such cross-client information is the \# clients who sent the $j$-th coordinate. In our Rand-Proj-Spatial, this information can be thought of encoded in the eigenvalues of random matrices $\mathbf{G}_i$ each client uses during encoding. Each decoded
> $\mathbf{x}_i$ is
> $\mathbf{x}_i = (T(\sum_{j=1}^{n}\mathbf{G}_j^T \mathbf{G}_j))^{\dag} \mathbf{G}_i^T\mathbf{G}_i \mathbf{x}_i$
>
> , where $\sum_{j=1}^{n}\mathbf{G}_j\mathbf{G}_j^T$ is the cross-client information.
>
> Since the above optimization problem considers a single decoding matrix $\mathbf{W}_i$
>
> for each client, solving this problem might still lead to individual decoding of each client, which does not use potential cross-client correlation information.
> Another potential form of the optimization problem (Problem 2) we considered is solving
>
> $\hat{\mathbf{x}} = \argmin_{\mathbf{x}} \|\frac{1}{n}\sum_{i=1}^{n}\mathbf{G}_i \mathbf{x}- \frac{1}{n}\sum_{i=1}^{n} \mathbf{G}_i \mathbf{x}_i\|_2^2$.
>
> However, through simulations, the solution to Problem 2 is consistently worse than the one presented in the paper.
> We can add the discussion to the Appendix.
>
> Q1: Note both references mentioned here belong to the orthogonal line of work on quantization. See the difference between quantization and sparsification in common response $\textbf{Quantization vs. Sparsification}$. But we are happy to include the two references in our citation. While random projection + compression and leveraging correlation information across clients are well explored in quantization, these ideas are less explored in sparsification. To the best of our knowledge, the only prior work that explores cross-client correlation is [1]. And as we show in the Appendix A, unlike in quantization, applying a simple rotation in sparsification will not lead to improvement.
>
> As discussed in Section 4.3 and Appendix B.2, we considered analysis tools such as J-L embeddings but the analysis of SRHT (and other random matrices) in current literature concerns mainly asymptotic properties. Yet, we do need an explicit distribution of the eigen-decomposition of the random matrix in Rand-Proj-Spatial in our case (which does not exist), to derive a tight MSE analysis and to compare against that of Rand-$k$-Spatial and Rand-$k$.
>
> L1: The randomness of $\mathbf{G}_i = \frac{1}{\sqrt{d}}\mathbf{E}_i \mathbf{H} \mathbf{D}_i$ comes from two sources: 1) the subsampling matrix $\mathbf{E}_i$ and the diagonal matrix $\mathbf{D}_i$ with Redamacher random variables on the diagonal. $\mathbf{E}_i$ and $\mathbf{D}_i$ can both be generated by some pseudorandom function (PRF) and one only needs a seed (aka. a number) to describe the PRF. The length of the seed is related to the strength of pseudorandomness and can be independent of $d$. It is possible to send the seed of the PRF with only a constant number of bits. Since the server and the clients have shared randomness, each client only needs to send an additional constant number of bits to the server so that the server can generate $\mathbf{E}_i$ and $\mathbf{D}_i$, and then reconstruct $\mathbf{G}_i$.
> See [1], for example, of how such PRF seed is applied in their distributed algorithm.
>
> -----
>
> [1] ``Locally Differentially Private Sparse Vector Aggregation''. Mingxun Zhou, Tianhao Wang, T-H. Hubert Chan, Giulia Fanti, Elaine Shi. IEEE S\&P 2022.

---

> > ### Comment · Reviewer_uyDa · 2023-08-15
> > **Response to rebuttal**
> >
> > Dear authors,
> >
> > Sincere apologies for the late reply. Thank you very much for your thoughtful rebuttal. Regarding the optimization problem, apologies again for not being clearer. This is what I had in mind -- The decoded estimate at the PS for worker $i$ is $\mathbf{W}_i\mathbf{G}_i\mathbf{x}_i$, where $\mathbf{W}_i$ is a decoding matrix. The mean estimated at the server is
> >
> > $$\frac{1}{n}\sum_{i=1}^{n} \mathbf{W}_i\mathbf{G}_i\mathbf{x}_i$$
> >
> > Consequently, solving the following optimization problem:
> >
> > $$\mathbf{W}^* = argmin_{\mathbf{W}} \mathbb{E}\left\lVert \overline{\mathbf{x}} - \frac{1}{n}\sum_{i=1}^{n}\mathbf{W}_i\mathbf{G}_i\mathbf{x}_i \right\rVert_2^2$$
> >
> > would give us the optimal decoding matrices, using which we can estimate the mean as: $\widehat{\mathbf{x}} = \frac{1}{n}\sum_{i=1}^{n} \mathbf{W}_i\mathbf{G}_i\mathbf{x}_i$. Additionally, we can impose the unbiasedness constraint:
> >
> > $\frac{1}{n}\sum_{i=1}^{n}\mathbb{E}\left[ \mathbf{W}_i\mathbf{G}_i\mathbf{x}_i  \right] = \overline{\mathbf{x}}$,
> >
> > and solve the constrained minimization problem. My original question was, is this formulation any way equivalent to the formulation in the paper? Also, yes, a discussion in the paper on this or the optimization problem mentioned in the rebuttal would be informative. More specifically, an answer to the question -- **Why is minimizing the sum of squared norms of the random projections a good strategy to exploit correlation for mean estimation?** This should yield answers to natural questions like -- why not sum of $\ell_2$ -- norms (not necessarily squared), or $\ell_\infty$ -- norm. While I understand that other choices might not yield nice closed form expressions which can be analyzed, if this is the reason, then it should be mentioned that the this particular choice of optimization problem makes analytical tractability feasible. And possibly later, when the authors (or someone else) comes up with lower bounds to this work, it will reveal the optimality of any strategy.
> >
> > Secondly, I agree that quantization and sparsification are not entirely different lines of work. A practical implementation will always employ quantization (even if it is $32$ or $64$ -- bit single / double precision formats). I believe it is important to discuss this work in the context of quantization as well, because if the eventual goal of any work is to be employed for communication compression, both quantization and sparsification would likely be employed and might involve tradeoffs. I agree this is not the core contribution of this work, but in my opinion, it does warrant a discussion -- for example, is simple concatetation of this strategy with quantization good enough (it's quite likely not optimal). So future work might involve coming up with a joint strategy. It is my opinion that elaborating and acknowledging this connection would help the work put better in context -- but once again, this is not critical and the authors may or may not agree with this opinion.
> >
> > Thirdly, regarding Limitation 1, I agree a lot of existing works do consider the existence of perfect shared randomness while doing the analysis, but practical implementations would require PRNG seeds to simulate randomness (which cannot be perfect unless at least $log_2 d$ bits are exchanged -- which is the entropy of $\mathbf{H}$). I understand the gap between theory and practical implementation is not straightforward to resolve, and just ask the authors to point this out as a limitation/assumption of perfect shared randomness.
> >
> > Fourthly, yes, it is my opinion that a more descriptive paper title is important so as not to be misleading. Something that brings out the fact that correlation between the random projections are exploited for the purpose of compression for mean estimation.
> >
> > Finally, do the authors have any response to limitation 2? Is it possible to convert the high probability statement purely in terms of expectations over all sources of randomness? If I understand correctly, currently in Thm 4.3 expectation is over the randomness in rand - k, and the $o(1)$ probability is over construction of the random projections $\mathbf{G}_i$.

---

> > > ### Author Response · Authors · 2023-08-16
> > >
> > > Sorry for posting this response late. Let me respond to Limitation 2: $1-o(1)$ probability bound vs. in expectation bound first.
> > >
> > > The $1-o(1)$ high probability part comes from the fact that the MSE stated in Theorem 4.3 holds when rank($\textbf{S}$) $= nk$, where $\textbf{S} = \sum_{i=1}^{n} \textbf{G}_i^T \textbf{G}_i$ and
> > > $\textbf{G}_i$’s are the random SRHT matrices generated by each client.
> > >
> > > With a slightly tighter analysis, we can get rid of the $o(1)$ part and have in expectation bound of MSE(Rand-Proj-Spatial) under max client correlation as follows. Let $\delta$ be the probability that $\textbf{S}$ does not have full rank (i.e., has rank $(\textbf{S})< nk$), and let $\delta_c$ be the probability that $\textbf{S}$ has rank $c \in \\{k, k+1,\dots, nk-1 \\}$. Note that $\delta = \sum_{c=k}^{nk-1} \delta_c$. Now, following similar steps in the proof of Theorem 4.3 in Appendix C:
> > >
> > > $\textbf{Computing $\bar{\beta}$}.$
> > > First, to ensure that our estimator $\widehat{\textbf{x}}$ is unbiased, we need $\bar{\beta} \mathbb{E}[\mathbf{S}^{\dagger}\mathbf{S} \mathbf{x}] = \mathbf{x}$. Consequently,
> > >
> > > $
> > > \mathbf{x} = \bar{\beta} \mathbb{E}[\mathbf{U} \Lambda^{\dagger} \mathbf{U}^T \mathbf{U} \Lambda \mathbf{U}^T] \mathbf{x}
> > > $
> > >
> > > $
> > > = \bar{\beta} \left[ \sum_{\mathbf{U} = \Phi} \Pr[\mathbf{U} = \Phi]  \mathbf{U} \mathbb{E}[\Lambda^{\dagger}\Lambda \mid \mathbf{U} = \Phi] \mathbf{U}^T \right] \mathbf{x}
> > > $
> > >
> > > $
> > > \overset{(a)}{=} \bar{\beta} \left[ \sum_{\textbf{U} = \Phi} \Pr[\textbf{U} = \Phi]  \textbf{U} \mathbb{E}[ \text{diag}(\mathbf{m}) \mid \textbf{U} = \Phi] \textbf{U}^T \right] \textbf{x}
> > > $
> > >
> > > $
> > > \overset{(b)}{=} \bar{\beta} \sum_{\textbf{U} = \Phi} \Pr[\textbf{U} = \Phi] \textbf{U} \left[ (1-\delta) \frac{nk}{d} I_d + \sum_{c=k}^{nk-1} \delta_c \frac{c}{d} \mathbf{I}_d \right] \textbf{U}^T \textbf{x}
> > > $
> > >
> > > (For $c < nk$, $\mathbb P(\text{rank}(\text{diag}(\mathbf{m})) = c) = \delta_c$ and $\sum_{c=k}^{nk-1} \delta_c = \delta$)
> > >
> > > $
> > > = \bar{\beta} \left[ (1-\delta) \frac{nk}{d} + \sum_{c=k}^{nk-1} \delta_c \frac{c}{d} \right] \textbf{x}
> > > $
> > >
> > > $
> > > \Rightarrow \bar{\beta} = \frac{d}{(1-\delta) n k + \sum_{c=k}^{nk-1} \delta_c c}
> > > $
> > >
> > > where in $(a)$, $\mathbf{m} \in \mathbb{R}^d$ such that
> > >
> > > $\mathbf{m}_i = 1$
> > >
> > > if $\Lambda_{jj} > 0$ for $j \in \\{1,2,\dots,d\\}$ and $0$ otherwise.
> > >
> > > Also, by construction of $\mathbf{S}$, $\text{rank}(\text{diag}(\mathbf{m})) \leq nk$. Further, $(b)$ follows by symmetry across the $d$ dimensions.
> > >
> > > Since $\delta k \leq \sum_{c=k}^{nk-1} \delta_c c \leq \delta (nk-1)$, there is $\frac{d}{(1-\delta) n k + \delta (nk-1)} \leq \bar{\beta} \leq \frac{d}{(1-\delta) n k + \delta k}$.
> > >
> > > $\textbf{Computing the MSE.}$ Next, we use the value of $\bar{\beta}$ derived above to compute the MSE of Rand-Proj-Spatial.
> > >
> > > $
> > > MSE = \mathbb{E}[\|\|\widehat{\textbf{x}} - \bar{\textbf{x}}\|\|_2^2]
> > > = \mathbb{E}[\|\| \bar{\beta}\mathbf{S}^{\dagger}\mathbf{S} \textbf{x} - \textbf{x}\|\|_2^2]
> > > $
> > >
> > > $
> > > = \bar{\beta}^2\mathbb{E}[\|\|\mathbf{S}^{\dagger}\mathbf{S} \textbf{x} \|\|_2^2] + \|\|\textbf{x}\|\|_2^2 - 2\Big\langle \bar{\beta} \mathbb{E}[\mathbf{S}^{\dagger} \mathbf{S}\textbf{x}], \textbf{x} \Big\rangle
> > > $
> > >
> > > $
> > > =\bar{\beta}^2\mathbb{E}[\|\|\mathbf{S}^{\dagger}\mathbf{S} \textbf{x} \|\|_2^2] - \|\|\textbf{x}\|\|_2^2
> > > $
> > >
> > > (Using unbiasedness of $\widehat{\textbf{x}}$)
> > >
> > > $
> > > = \bar{\beta}^2\textbf{x}^T\mathbb{E}[\mathbf{S}^T (\mathbf{S}^{\dagger})^T \mathbf{S}^{\dagger}\mathbf{S}] \textbf{x} - \|\|\textbf{x}\|\|_2^2
> > > $
> > >
> > > Using $\mathbf{S}^{\dagger} = \textbf{U}\Lambda^{\dagger}\textbf{U}^T$,
> > >
> > > $
> > > \textbf{x}^T \mathbb{E}[\mathbf{S}^T (\mathbf{S}^{\dagger})^T \mathbf{S}^{\dagger}\mathbf{S}] \textbf{x}
> > > = \textbf{x}^T \mathbb{E}[\textbf{U} \Lambda \textbf{U}^T \textbf{U} \Lambda^{\dagger} \textbf{U}^T \textbf{U} \Lambda^{\dagger} \textbf{U}^T \textbf{U} \Lambda \textbf{U}^T] \textbf{x}
> > > $
> > >
> > > $
> > > =\textbf{x}^T\mathbb{E}[\textbf{U} \Lambda (\Lambda^{\dagger})^{2} \Lambda \textbf{U}^T] \textbf{x}
> > > $
> > >
> > > $
> > > = \sum_{\textbf{U} = \Phi} \textbf{x}^T\textbf{U} \mathbb{E}[\Lambda (\Lambda^{\dagger})^{2} \Lambda] \textbf{U}^{T} \textbf{x} \cdot \Pr[\textbf{U} = \Phi]
> > > $
> > >
> > > $
> > > = \sum_{\textbf{U} = \Phi} \textbf{x}^T \textbf{U} \left[ (1-\delta) \frac{nk}{d} I_d + \sum_{c=k}^{nk-1} \delta_c \frac{c}{d} \mathbf{I}_d \right] \textbf{U}^{T} \textbf{x} \cdot \Pr[\textbf{U} = \Phi]
> > > $
> > >
> > > $
> > > = \left[ (1-\delta) \frac{nk}{d} + \sum_{c=k}^{nk-1} \delta_c \frac{c}{d} \right] \|\|\textbf{x}\|\|_2^2 = \frac{1}{\bar{\beta}} \|\|\textbf{x}\|\|_2^2
> > > $
> > >
> > > Therefore,
> > >
> > > $
> > > MSE(\text{Rand-Proj-Spatial}) = (\bar{\beta} - 1) \|\|\textbf{x}\|\|_2^2 \leq \left[ \frac{d}{(1-\delta) n k + \delta k} - 1 \right] \|\|\textbf{x}\|\|_2^2
> > > $

---

> > > > ### Author Response · Authors · 2023-08-16
> > > > **Response to L2 (Cont'd)**
> > > >
> > > > We verify empirically that $\delta \approx 0$. The results are here at https://anonymous.4open.science/r/CorrAwareDME-086F/rank_S.pdf. With $d \in \{32, 64, 128, \dots, 1024\}$ and 4 different $nk$ values such that $nk \leq d$ for each $d$, we compute rank($\textbf{S}$) for $10^5$ trials for each pairs of $(nk, d)$ values and plot the results for all trials. As one can observe from the plots, rank($\textbf{S}$) is full (i.e., $=nk$) w.h.p. That is, $\delta \approx 0$.
> > > >
> > > > Hence,
> > > > $
> > > > MSE(\text{Rand-Proj-Spatial}) \approx (\frac{d}{nk} -1) \|\|\textbf{x}\|\|_2^2
> > > > $
> > > >
> > > > Now we use the above in-expectation upper bound to compare the MSE of our Rand-Proj-Spatial with the MSE of the baseline Rand-$k$ analytically in the full correlation case.
> > > > Recall in this case, $MSE(\text{Rand-$k$}) = \frac{1}{n}(\frac{d}{k} - 1) \|\|\textbf{x}\|\|_2^2$.
> > > >
> > > > We have
> > > >
> > > > $
> > > > \frac{d}{(1-\delta) n k + \delta k} - 1 \leq \frac{d}{nk} - \frac{1}{n}
> > > > $
> > > >
> > > > $
> > > > \Rightarrow \frac{d}{k} \frac{n-(1-\delta)n - \delta}{n((1-\delta) n + \delta)} \leq 1 - \frac{1}{n}
> > > > $
> > > >
> > > > $
> > > > \Rightarrow \frac{d}{k} \frac{\delta}{(1-\delta) n + \delta} \leq 1
> > > > $
> > > >
> > > > $
> > > > \Rightarrow d \delta \leq nk - nk \delta + k \delta
> > > > $
> > > >
> > > > $
> > > > \Rightarrow \delta \leq \frac{1}{\frac{d}{nk} + 1 - \frac{1}{n}}
> > > > $
> > > >
> > > > Since $nk \leq d$ and $n \geq 2$,
> > > >
> > > > $
> > > > \delta \leq \frac{1}{\frac{d}{nk} + 1 - \frac{1}{n}} \leq \frac{1}{1 + \frac{1}{2}} = \frac{2}{3}
> > > > $
> > > >
> > > > As we see in the experiments, $\delta \approx 0$. So, the proposed approach is always better than Rand-$k$.

---

> > > > > ### Author Response · Authors · 2023-08-18
> > > > > **Response to Follow Up Questions**
> > > > >
> > > > > $\textbf{Point 1-1: The New Optimization Problem Proposed by the Reviewer. }$ Thank you very much for clarifying the general optimization problem. This formulation is not equivalent to the formulation in the paper. As we have mentioned earlier in the response, if we need to optimize a decoding matrix $\textbf{W}_i$ for each client $i$, we are doing individual decoding of each client's compressed vector instead of a joint decoding process that considers all clients' compressed vectors. A joint decoding process is able to utilize the cross-client information while an individual decoding process is not able to do so. Our goal in this work is to use cross-client information (aka. the existence of cross-client correlation) to propose a joint decoding process to reduce error.
> > > > >
> > > > > Indeed, we can show that solving this optimization recovers the MSE of our baseline Rand-$k$ as follows.
> > > > >
> > > > > The optimization problem proposed by the reviewer is
> > > > >
> > > > > $
> > > > > argmin_{\mathbf{W}} f(\mathbf{W}) = argmin_{\mathbf{W}} \mathbb{E}[\|\|\bar{\mathbf{x}} - \frac{1}{n}\sum_{i=1}^{n} \mathbf{W}_i \mathbf{G}_i \mathbf{x}_i\|\|_2^2]
> > > > > $
> > > > >
> > > > > subject to $\bar{\mathbf{x}} = \frac{1}{n}\sum_{i=1}^{n} \mathbb{E}[\mathbf{W}_i \mathbf{G}_i \mathbf{x}_i]$, where $\mathbf{W} = (\mathbf{W}_1, \mathbf{W}_2, \dots, \mathbf{W}_n)$ and $\mathbf{W}_i \in \mathbb{R}^{d \times k}$, $\mathbf{G}_i \in \mathbb{R}^{k \times d}$.
> > > > >
> > > > > $
> > > > > f(\mathbf{W}) = \mathbb{E}[\|\frac{1}{n}\sum_{i=1}^{n}(\mathbf{x}_i - \mathbf{W}_i \mathbf{G}_i \mathbf{x}_i)\|_2^2]
> > > > > $
> > > > >
> > > > > $
> > > > > = \mathbb{E}[\|\|\frac{1}{n}\sum_{i=1}^{n} (\mathbf{I}_d - \mathbf{W}_i \mathbf{G}_i) \mathbf{x}_i \|\|_2^2]
> > > > > $
> > > > >
> > > > > $= \mathbb{E}[\frac{1}{n^2}\Big(\sum_{i=1}^{n} \|\|(\mathbf{I}_d - \mathbf{W}_i \mathbf{G}_i) \mathbf{x}_i \|\|_2^2
> > > > > $
> > > > >
> > > > > $ + \sum_{i \neq j} \langle (\mathbf{I}_d - \mathbf{W}_i \mathbf{G}_i) \mathbf{x}_i, (\mathbf{I}_d - \mathbf{W}_j \mathbf{G}_j) \mathbf{x}_j\rangle\Big)]
> > > > > $
> > > > >
> > > > > $= \frac{1}{n^2}\Big(\sum_{i=1}^{n}\mathbb{E}[\|\|(\mathbf{I}_d - \mathbf{W}_i \mathbf{G}_i)\mathbf{x}_i \|\|_2^2]
> > > > > $
> > > > >
> > > > > $+ \sum_{i \neq j}\mathbb{E}[\langle (\mathbf{I}_d - \mathbf{W}_i \mathbf{G}_i) \mathbf{x}_i, (\mathbf{I}_d - \mathbf{W}_j \mathbf{G}_j) \mathbf{x}_j\rangle]\Big)
> > > > > $
> > > > >
> > > > > Since there is an additional constraint for unbiasedness, i.e., $\bar{\mathbf{x}} = \frac{1}{n}\sum_{i=1}^{n} \mathbf{x}_i = \frac{1}{n} \mathbb{E}[\mathbf{W}_i \mathbf{G}_i \mathbf{x}_i]$,
> > > > >
> > > > > there is $\frac{1}{n}\sum_{i=1}^{n}\mathbb{E}[(\mathbf{I}_d - \mathbf{W}_i \mathbf{G}_i)\mathbf{x}_i] = 0$.
> > > > > We now show that a sufficient and necessary condition to satisfy the above unbiasedness constraint is that for all $i\in [n]$, $\mathbb{E}[\mathbf{W}_i\mathbf{G}_i] = \mathbf{I}_d$.
> > > > >
> > > > > $\textit{Sufficiency. }$ It is obvious that if for all $i\in [n]$, $\mathbb{E}[\mathbf{W}_i\mathbf{G}_i] = \mathbf{I}_d$,
> > > > >
> > > > > then we have $\frac{1}{n}\sum_{i=1}^{n}\mathbb{E}[(\mathbf{I}_d - \mathbf{W}_i \mathbf{G}_i)\mathbf{x}_i] = 0$.
> > > > >
> > > > > $\textit{Necessity. }$ Consider the special case that $\mathbf{x}_i = n \mathbf{e}_k$ ($k$-th canonical basis vector) and $\mathbf{x}_j = 0$, for all $j \in [n] \setminus \{i\}$. Then,
> > > > >
> > > > > $\mathbf{e}_k = \bar{\mathbf{x}}$
> > > > >
> > > > > $= \frac{1}{n} \sum_{i=1}^{n} \mathbb{E}[\mathbf{W}_i \mathbf{G}_i \mathbf{x}_i] $
> > > > >
> > > > > $= \frac{1}{n} \mathbb{E}[\mathbf{W}_i \mathbf{G}_i n \mathbf{e}_k] = \mathbb{E}[\mathbf{W}_i \mathbf{G}_i] \mathbf{e}_k = \left[ \mathbb{E}[\mathbf{W}_i \mathbf{G}_i] \right]_k.
> > > > > $
> > > > >
> > > > > where $\left[ \cdot \right]$ means the $k$-th column of matrix $\mathbb{E}[\mathbf{W}_i \mathbf{G}_i]$.
> > > > >
> > > > > By varying $k$ over $[n]$, we see that we need $\mathbb{E}[\mathbf{W}_i \mathbf{G}_i] = \mathbf{I}_d$. Varying $i$ over $[n]$, we see that we need $\mathbb{E}[\mathbf{W}_j \mathbf{G}_j] = \mathbf{I}_d$ for all $j \in [n]$. Since our approach is agnostic to the choice of vectors, we need this choice of decoder matrices.
> > > > >
> > > > > Therefore, the unbiasedness constraint $\frac{1}{n}\sum_{i=1}^{n} \mathbb{E}[(\mathbf{I}_d - \mathbf{W}_i \mathbf{G}_i) \mathbf{x}_i] = 0 \Leftrightarrow \forall i\in [n], \mathbb{E}[\mathbf{W}_i\mathbf{G}_i] = \mathbf{I}_d$.
> > > > >
> > > > > Furthermore, notice the second term in the above MSE expression is 0 --- that is,
> > > > > $\mathbb{E}[\langle (\mathbf{I}_d - \mathbf{W}_i \mathbf{G}_i) \mathbf{x}_i, (\mathbf{I}_d - \mathbf{W}_j \mathbf{G}_j) \mathbf{x}_j\rangle] = 0$, $\forall i, j \in [n]$.
> > > > >
> > > > > Hence, we only need to consider minimizing over $\mathbf{W}_1, \dots, \mathbf{W}_n$
> > > > >
> > > > > in the objective $\sum_{i=1}^{n}\mathbb{E}[\|\|(\mathbf{I}_d - \mathbf{W}_i \mathbf{G}_i)\mathbf{x}_i\|\|_2^2]$.
> > > > >
> > > > > Since each $\mathbf{W}_i$ only appears in one term, we can optimize $\mathbf{W}_i$ separately.
> > > > >
> > > > > For $i\in [n]$, we want to solve
> > > > > $\min_{\mathbf{W}_i} \mathbb{E}[\|\|(\mathbf{I}_d - \mathbf{W}_i \mathbf{G}_i) \mathbf{x}_i\|\|_2^2]$
> > > > > such that $\mathbb{E}[\mathbf{W}_i \mathbf{G}_i] = \mathbf{I}_d$.

---

> > > > > > ### Author Response · Authors · 2023-08-18
> > > > > > **Response to Follow Up Questions (Cont'd)**
> > > > > >
> > > > > > One natural solution is to take $\mathbf{W}_i = \frac{d}{k}\mathbf{G}_i^{\dagger}$, $\forall i \in [n]$. For $i\in [n]$, let $\mathbf{G}_i = \mathbf{V}_i \Lambda_i \mathbf{U}_i^\top$ be its SVD, where $\mathbf{V}_i \in \mathbb R^{k \times d}$ and $\mathbf{U}_i \in \mathbb{R}^{d \times d}$ are orthogonal matrices. Then,
> > > > > >
> > > > > > $\mathbf{W}_i \mathbf{G}_i = \frac{d}{k} \mathbf{U}_i \Lambda_i^\dagger \mathbf{V}_i^\top \mathbf{V}_i \Lambda \mathbf{U}_i^\top = \frac{d}{k} \mathbf{U}_i \Lambda_i^\dagger \Lambda \mathbf{U}_i^\top = \frac{d}{k} \mathbf{U}_i \Sigma_i \mathbf{U}_i^\top
> > > > > > $
> > > > > >
> > > > > > (since $\mathbf{U}_i^\dagger = \mathbf{U}_i^\top$ and $\mathbf{V}_i^\dagger = \mathbf{V}_i^\top$). Here, $\Sigma_i$ is diagonal matrix with $0$s and $1$s.
> > > > > >
> > > > > > Let $\mu(\mathbf{U}_i)$ be the measure of $\mathbf{U}_i$, we see that
> > > > > >
> > > > > > $
> > > > > > \mathbb{E}[\mathbf{W}_i\mathbf{G}_i] = \frac{d}{k}\mathbb{E}[\mathbf{U}_i\Sigma_i\mathbf{U}_i^\top]
> > > > > > $
> > > > > >
> > > > > > $ = \frac{d}{k}\int_{\mathbf{U}_i}\mathbb{E}[\mathbf{U}_i\Sigma_i\mathbf{U}_i^\top \mid \mathbf{U}_i] \cdot d\mu(\mathbf{U}_i)
> > > > > > $
> > > > > >
> > > > > > $= \frac{d}{k} \int_{\mathbf{U}_i} \mathbf{U}_i\mathbb{E}[\Sigma_i \mid \mathbf{U}_i] \mathbf{U}_i^\top \cdot d\mu(\mathbf{U}_i)
> > > > > > $
> > > > > >
> > > > > > $= \frac{d}{k}\int_{\mathbf{U}_i} \mathbf{U}_i \frac{k}{d}\mathbf{I}_d \mathbf{U}_i^\top \cdot d\mu(\mathbf{U}_i)
> > > > > > $
> > > > > >
> > > > > > $= \frac{d}{k} \frac{k}{d}\mathbf{I}_d = \mathbf{I}_d
> > > > > > $
> > > > > >
> > > > > > The MSE now becomes
> > > > > >
> > > > > > $MSE = \frac{1}{n^2} \sum_{i=1}^{n} \mathbb{E}[\|(\mathbf{I}_d - \mathbf{W}_i \mathbf{G}_i) \mathbf{x}_i\|_2^2]
> > > > > > $
> > > > > >
> > > > > > $= \frac{1}{n^2} \sum_{i=1}^{n} \Big(\|\|\mathbf{x}_i\|\|_2^2 + \mathbb{E}[\|\|\mathbf{W}_i\mathbf{G}_i \mathbf{x}_i\|\|_2^2]
> > > > > >     - 2 \langle \mathbf{x}_i, \mathbb{E}[\mathbf{W}_i\mathbf{G}_i] \mathbf{x}_i \rangle
> > > > > > \Big)
> > > > > > $
> > > > > >
> > > > > > $= \frac{1}{n^2} \sum_{i=1}^{n} \Big(
> > > > > >         \|\| \mathbf{x}_i\|\|_2^2 + \mathbb{E}[\|\| \mathbf{W}_i\mathbf{G}_i \mathbf{x}_i \|\|_2^2]
> > > > > >         - 2 \langle \mathbf{x}_i, \mathbf{x}_i \rangle
> > > > > >     \Big)
> > > > > > $
> > > > > >
> > > > > > $= \frac{1}{n^2}\sum_{i=1}^{n}\Big(
> > > > > >         \mathbb{E}[\|\| \mathbf{W}_i\mathbf{G}_i \mathbf{x}_i \|\|_2^2]-\|\| \mathbf{x}_i \|\|_2^2
> > > > > >     \Big)
> > > > > > $
> > > > > >
> > > > > > $= \frac{1}{n^2} \sum_{i=1}^{n} \Big(
> > > > > >         \mathbf{x}_i^\top\mathbb{E}[(\mathbf{W}_i\mathbf{G}_i)^\top (\mathbf{W}_i\mathbf{G}_i)]\mathbf{x}_i - \|\|\mathbf{x}_i\|\|_2^2
> > > > > >     \Big)
> > > > > > $
> > > > > >
> > > > > > Again let $\mathbf{G}_i = \mathbf{V}_i \Lambda_i \mathbf{U}_i^\top$ be its SVD and consider $\mathbf{W}_i\mathbf{G}_i = \frac{d}{k} \mathbf{U}_i \Sigma_i\mathbf{U}_i^\top$, where $\Sigma_i$ is a diagonal matrix with 0s and 1s.
> > > > > > Then,
> > > > > >
> > > > > > $MSE = \frac{1}{n^2}\sum_{i=1}^{n} \Big(
> > > > > >         \mathbf{x}_i^\top \frac{d^2}{k^2}\mathbb{E}[\mathbf{U}_i \Sigma_i \mathbf{U}_i^\top \mathbf{U}_i \Sigma_i \mathbf{U}_i^\top]\mathbf{x}_i -\|\|\mathbf{x}_i \|\|_2^2
> > > > > >     \Big)
> > > > > > $
> > > > > >
> > > > > > $= \frac{1}{n^2}\sum_{i=1}^{n} \Big(
> > > > > >         \frac{d^2}{k^2}\mathbf{x}_i^\top \mathbb{E}[\mathbf{U}_i \Sigma_i^2 \mathbf{U}_i^\top]\mathbf{x}_i -\|\|\mathbf{x}_i \|\|_2^2
> > > > > >     \Big)
> > > > > > $
> > > > > >
> > > > > > Since $\mathbf{G}_i$ has rank $k$, $\Sigma_i$ is a diagonal matrix with $k$ out of $d$ entries being 1 and the rest being 0. Let $\mu(\mathbf{U}_i)$ be the measure of $\mathbf{U}_i$. Hence,
> > > > > > for $i \in [n]$,
> > > > > >
> > > > > > $\mathbb{E}[\mathbf{U}_i \Sigma_i^2 \mathbf{U}_i^\top]
> > > > > > $
> > > > > >
> > > > > > $= \int_{\mathbf{U}_i} \mathbb{E}[\mathbf{U}_i \Sigma_i^2\mathbf{U}_i^\top \mid \mathbf{U}_i] d \mu(\mathbf{U}_i)
> > > > > > $
> > > > > >
> > > > > > $= \int_{\mathbf{U}_i} \mathbf{U}_i\mathbb{E}[\Sigma_i^2\mid \mathbf{U}_i] \mathbf{U}_i^\top d\mu(\mathbf{U}_i)
> > > > > > $
> > > > > >
> > > > > > $= \int_{\mathbf{U}_i} \mathbf{U}_i \frac{k}{d}\mathbf{I}_d \mathbf{U}_i^\top d\mu(\mathbf{U}_i)
> > > > > > $
> > > > > >
> > > > > > $= \frac{k}{d}\int_{\mathbf{U}_i} \mathbf{I}_d d\mu(\mathbf{U}_i)
> > > > > > $
> > > > > >
> > > > > > $= \frac{k}{d}\mathbf{I}_d
> > > > > > $
> > > > > >
> > > > > > Therefore, the MSE by solving the optimization problem is
> > > > > >
> > > > > > $MSE = \frac{1}{n^2} \sum_{i=1}^{n} \Big(
> > > > > >         \frac{d^2}{k^2}\mathbf{x}_i^\top \frac{k}{d}\mathbf{I}_d \mathbf{x}_i - \|\|\mathbf{x}_i\|\|_2^2
> > > > > >     \Big)
> > > > > > $
> > > > > >
> > > > > > $= \frac{1}{n^2}(\frac{d}{k} - 1)\sum_{i=1}^{n}\|\|\mathbf{x}_i\|\|_2^2
> > > > > > $
> > > > > >
> > > > > > which is exactly the same MSE as that of Rand-$k$.

---

> > > > > > > ### Author Response · Authors · 2023-08-18
> > > > > > > **Response to Follow Up Questions (Cont'd)**
> > > > > > >
> > > > > > > $\textbf{Point 1-2: Minimizing Sum of Squared Norms. }$
> > > > > > >
> > > > > > > Since we consider MSE as the error metric (a common choice in practice and in the analysis of optimization algorithms), that measures the error in the $\ell_2$ norm squared, a natural choice is to consider the $\ell_2$ norm squared in the optimization problem. Its analytical tractability also influenced our decision to go with it. However, we do acknowledge that it is possible to optimize the error in the other norms and some might not be analytically tractable. We shall clarify this point further in the updated version.
> > > > > > >
> > > > > > > $\textbf{Point 2: Further Discussion on Quantization vs. Sparsification. }$
> > > > > > >
> > > > > > > We will make the discussion on quantization vs. sparsification clearer and better position the contribution of our work. If one wants to drastically reduce the communication cost to be $O(k)$ instead of $O(d)$, then one needs sparsification techniques. And yes, deriving the optimal communication cost-accuracy tradeoffs for a combination of quantization and sparsification techniques jointly can be an interesting future direction. We will also add this point.
> > > > > > >
> > > > > > > $\textbf{Point 3: Shared Randomness. }$
> > > > > > >
> > > > > > > Thank you very much for pointing this out. As we are not experts on pseudo-randomness, we were not aware of the practical limitation. We will incorporate the suggestion and add the limitation/assumption of perfectly shared randomness.
> > > > > > >
> > > > > > > $\textbf{Point 4: More Informative Title. }$
> > > > > > >
> > > > > > > We will change the title to be ``Random Projection Based Correlation Aware Distributed Vector Mean Estimation''.

---

> > > > > > > > ### Comment · Reviewer_uyDa · 2023-08-20
> > > > > > > > **Acknowledgement and response**
> > > > > > > >
> > > > > > > > Dear authors,
> > > > > > > >
> > > > > > > > Thank you very much for your very detailed reply and care in responding to my concerns. Your response regarding the optimization problem makes sense, and it would be informative if you could possibly add a section in the appendix which details why your choice of the objective function makes more sense than others -- including the one I mentioned (that requires individual decoding) as well as the one you considered yourself (which you mentioned in the rebuttal).
> > > > > > > >
> > > > > > > > Secondly, thank you again for putting in the effort to remove the $o(1)$ clause in the statement of the theorem, as well as the empirical verification (I am a little concerned since you put links in the review, which was not supposed to be done -- so I didn't check them). But the theoretical justification suffices, and I agree that for high-dimensional random matrices, it is very reasonable to expect $\delta = 0$ empirically.
> > > > > > > >
> > > > > > > > Finally, thank you for summarizing our discussion. I trust that you will revise the paper so as to add the outcome of these discussions to relevant sections. This includes (quite importantly), a discussion of quantization vs. sparsification (along with all the missing references discussed here such as (Safaryan, Saha, Davies, Vargaftik, Suresh, etc.)) -- this is important to place the paper better in context as it is a point raised by other reviewers as well, and I do believe that quantization and sparsification are not entirely orthogonal and a joint perspective is often better for system design. Especially important is the application of random matrices for the joint design of correlated quantization and sparsification.
> > > > > > > >
> > > > > > > > I believe a more descriptive title would be: "Correlated sparsification for distributed mean estimation using random projections" or something on lines of these to highlight the contribution of this work in sparsification.
> > > > > > > >
> > > > > > > > Trusting that the authors would make these changes, I am raising my score to 7.

---

> > > > > > > > > ### Author Response · Authors · 2023-08-21
> > > > > > > > >
> > > > > > > > > Dear reviewer,
> > > > > > > > >
> > > > > > > > > Thank you very much for providing so much valuable feedback! Yes, we will include a section detailing our choice of the optimization objective and discussing potential ways of formulating the objective in the Appendix. We will give a broader overview of sparsification and quantization techniques to better position this work and stress a joint design of sparsification and quantization techniques, especially the one using correlation with applications of random matrices, as an interesting future direction.
> > > > > > > > > We will also incorporate other suggestions mentioned in the discussion.
> > > > > > > > > Finally, thank you for proposing a more descriptive title. We will adopt this new title :).

---

### Official Review · Reviewer_BRXu · 2023-07-26

**Soundness:** 3 good
**Presentation:** 3 good
**Contribution:** 2 fair
**Rating:** 6
**Confidence:** 2

**Summary:**

In distributed learning, computing the mean of the vectors sent by the clients is an important subtask. Motivated by this, the distributed mean estimation problem is studied in this paper. The two main techniques commonly used for these problems are Quantization and sparsification. Rand-K was one prominent sparsification method used earlier. In [9], this was generalized to Rand-k-Spatial to exploit the spatial correlations of the data using the server-side decoding procedure. A more general encoding scheme, Rand-Proj-Spatial is proposed in this paper which utilizes the cross-client correlation information. This uses Subsampled Randomized Hadamard Transform as random linear maps. They prove that this improves mean estimation under some assumptions and some experiments supporting their claim.

**Strengths:**

The paper is well presented. The problem generalizes the existing works and the results look interesting.

**Weaknesses:**

Their techniques are not adaptive to the client vectors. More motivation on why studying why correlation information of clients is practical will be interesting.

**Questions:**

Is Rand-Proj-Spatial optimal or close to optimal, given the correlation information? Some intuition will be good.

---

> ### Author Rebuttal · Authors · 2023-08-10
>
> W1: We gave the reason why we want our algorithm to be $\textit{non-adaptive}$ to client vectors in Section 1 line 82 - 89.
> The motivation of studying correlation information of clients is well explained and experimentally demonstrated in the prior work [1]. For example, in distributed optimization, clients who are geographically similar often have similar data, leading to having similar model parameters.
>
> Q1:
> When there is no correlation, the proposed Rand-Proj-Spatial with SRHT is very likely to be the optimal $\textit{unbiased}$ and $\textit{non-adaptive}$ encoder. Recall in this case, Rand-Proj-Spatial recovers Rand-$k$.
> The evidence of Rand-$k$ being the optimal $\textit{unbiased}$ and $\textit{non-adaptive}$ sparsification based encoder comes from Section 6 ``Optimal Encoders'' in [2]. If one considers the probability $p_{ij}$ across the clients and coordinates in the optimization problem in Section 6, which finds the best unbiased estimator that minimizes MSE, to be the same (aka. $\textit{non-adaptive}$), then the solution to this problem is exactly Rand-$k$.
>
> When there is max correlation, the proposed Rand-Proj-Spatial is also very likely to be the optimal, since by Eq.(6) in our paper, the error of the estimator in this case essentially depends on the rank of $\mathbf{S}$. Rand-Proj-Spatial with SRHT makes $\mathbf{S}$ achieve the max rank, i.e., $nk$.
>
> When the correlation is in between max and none, it is hard to derive a closed form expression for MSE. We do not know whether the current way of incorporating varying degrees of correlation by applying a transformation function $T$, which interpolates between max and no correlation, on the eigenvalues of $\mathbf{S}$, is optimal. However, the prior work [1] shows $T$ is the optimal transformation in the special case of our estimator, i.e., in Rand-$k$-Spatial.
> Hence, we believe the transformation function $T$ in Rand-Proj-Spatial, which uses a similar way to address the degrees of correlation in between max and none, should be at least close to optimal.
>
> ---------
>
> [1] Divyansh Jhunjhunwala, Ankur Mallick, Advait Harshal Gadhikar, Swanand Kadhe, and Gauri Joshi. ``Leveraging spatial and temporal correlations in sparsified mean estimation.'' In A. Beygelzimer, Y. Dauphin, P. Liang, and J. Wortman Vaughan, editors, Advances in Neural Information Processing Systems, 2021.
>
> [2] Jakub Konecny and Peter Richtárik. ``Randomized distributed mean estimation: Accuracy vs. communication.'' Frontiers in Applied Mathematics and Statistics, 4:62, 2018.

---

### Official Review · Reviewer_EAa8 · 2023-07-27

**Soundness:** 3 good
**Presentation:** 3 good
**Contribution:** 3 good
**Rating:** 6
**Confidence:** 3

**Summary:**

This paper discusses the problem of communication-efficient distributed vector mean estimation and proposes a new estimator called Rand-Proj-Spatial that improves upon existing techniques. The paper highlights the challenges of distributed optimization and Federated Learning, where communication cost can be a bottleneck, and explains how sparsification techniques can reduce this cost. The authors then introduce the Rand-k sparsification technique and show how it can be used to reduce communication cost in distributed vector mean estimation. They also propose the Rand-Proj-Spatial estimator, which combines Rand-k sparsification with a Subsampled Randomized Hadamard Transform to improve upon the performance of existing techniques. The paper concludes with experimental results that demonstrate the effectiveness of the Rand-Proj-Spatial estimator in various scenarios. Overall, the contributions of this paper include a new estimator for distributed vector mean estimation that is more communication-efficient than existing techniques, as well as insights into the challenges of communication-efficient distributed optimization and Federated Learning.

**Strengths:**

In terms of originality, the paper proposes a new estimator called Rand-Proj-Spatial that combines existing techniques in a novel way to improve upon the performance of distributed vector mean estimation. The use of Subsampled Randomized Hadamard Transform in the encoding-decoding procedure is a creative combination of existing ideas that has not been explored in this context before.

In terms of quality, the paper is well-written and clearly explains the challenges of communication-efficient distributed vector mean estimation and the proposed solutions. The authors provide rigorous theoretical analysis and experimental results to support their claims, which adds to the credibility of their contributions.

In terms of clarity, the paper is well-organized and easy to follow. The authors provide clear explanations of the technical concepts and use visual aids to help readers understand the proposed techniques.

In terms of significance, the paper addresses an important problem in the field of distributed optimization and Federated Learning. The proposed Rand-Proj-Spatial estimator has the potential to significantly reduce communication cost in these settings, which can have practical implications for real-world applications.

**Weaknesses:**

1. While the paper compare the proposed Rand-Proj-Spatial estimator with Rand-k-Spatial and other sparsification techniques, it would be beneficial to compare the proposed estimator with other state-of-the-art techniques for distributed vector mean estimation.

2. While the paper briefly mentions that the Subsampled Randomized Hadamard Transform can be computed efficiently, it would be helpful to provide a more detailed analysis of the computational complexity of the proposed estimator.

**Questions:**

1. What is the key difficulty/novelty in the design and analysis?

2. Can you provide more detailed analysis of the trade-off between communication cost and estimation accuracy for the proposed estimator and other sparsification techniques?

**Limitations:**

Not.

---

> ### Author Rebuttal · Authors · 2023-08-10
>
> W1: See common response $\textbf{Quantization vs. Sparsification}$.
>
> W2: See common response  $\textbf{Computation Time}$.
>
>
> Q1: $\textit{Two key novelties:}$ 1) the design of a general encoding-decoding algorithm that generalizes sparsification techniques and enables one to better use correlation among the client vectors to improve MSE compared to the prior work. 2) Previously, SRHT was mostly applied to reduce run time or to homogenize vector coordinates (i.e., to reduce variance). We propose a novel application of SRHT to better leverage cross-client correlation.
>
> $\textit{Key difficulty in analysis:}$ As we have discussed in Section 4.3 and in Appendix B.2, it is hard to derive closed-form expression of MSE for Rand-Proj-Spatial with SRHT in the general case when the cross-client correlation is in between max correlation and completely orthogonal.
> This is because the analysis requires a closed form expression for the non-asymptotic distributions of eigenvalues of SRHT, which is a hard problem by itself.
> To the best of our knowledge, previous analyses of SRHT, rely on the asymptotic properties of SRHT, such as the limiting eigen spectrum, or concentration bounds on the singular values, to derive asymptotic or approximate guarantees. But we need an exact non-asymptotic analysis; or the resulting MSE bound is too loose.
> In Appendix B.2, we also included several ideas we considered but did not work for deriving a closed-form MSE expression.
>
> Q2: $\textit{Comparison to biased sparsification based estimators.}$
> It is in general not fair to compare unbiased estimators, including our proposed Rand-Proj-Spatial, against biased estimators, including Top-$k$. As noted in prior distributed optimization literature [1] and mentioned in Section 1 line 81, unbiased estimators is more favorable to biased ones.
>
> $\textit{Comparison to unbiased sparsification based estimators.}$
> When the client vectors have max correlation and have no correlation, we give detailed analysis in Section 4.1 and Section 4.2 to compare the communication cost-estimation accuracy tradeoffs of our estimator against that of the known sparsification based unbiased estimators, e.g., Rand-$k$ and Rand-$k$-Proj.
> When the client vectors have a correlation in between max and none, as we responded in Q1, the analysis is difficult and so we empirically compared the communication-estimation accuracy tradeoffs in Figure 3. Note the Figure 3 presents the estimation error with the communication cost for all estimators being fixed. One can see that the error of the proposed Rand-Proj-Spatial is the lowest and this implies Rand-Proj-Spatial has the best communication-accuracy tradeoff.
>
> ----------
>
> [1] Samuel Horváth and Peter Richtarik. ``A better alternative to error feedback for communication- efficient distributed learning.'' In International Conference on Learning Representations, 2021.

---

> > ### Comment · Reviewer_EAa8 · 2023-08-20
> > **Response to rebuttal**
> >
> > Thanks for addressing my concerns. I will be raising my score from 5 to 6.

---

> > > ### Author Response · Authors · 2023-08-20
> > >
> > > We would like to thank the reviewer for providing valuable feedback!

---

### Author Rebuttal · Authors · 2023-08-10

1. $\textbf{Quantization vs. Sparsification.}$ There are two major techniques to reduce the communication cost of distributed vector mean estimation (DME): vector quantization and sparsification. The two techniques are orthogonal to each other. Vector quantization reduces the number of bits to represent each coordinate while vector sparsification reduces the number of coordinates each client sends. As a result, the communication cost of quantizaation is on the order of $O(d)$, while the communication cost of sparsification is on the order of $O(k)$ for some chosen $k \ll d$. In short, sparsification reduces the communication cost more aggressively compared to quantization. Note Rand-$k$-Spatial and the proposed Rand-Proj-Spatial, fall under sparsification, where the focus of algorithm design is on reducing the number of coordinates, while works mentioned by the reviewers, i.e., [1], [2] ([33] in the draft) and [3], all fall under quantization, where the focus is on reducing number of bits per coordinate. We can definitely include more quantization works to the citation and highlight them as an orthogonal line of works.
In practice, one can apply a combincation of quantization and sparsification techniques to reduce communication cost. Quantizing the $k$ values sent by each worker, for example, further reduces the number of bits per coordinate, while the communication cost remains $O(k)$, and incurs additional communication-accuracy tradeoffs.
To the best of our knowledge, the SOTA $\textit{unbiased}$ and $\textit{non-adaptive}$ sparsification methods are Rand-$k$ and Rand-$k$-Spatial, which serve as two baselines of our proposed Rand-Proj-Spatial. In the experiments, we also compare Rand-Proj-Spatial against two SOTA $\textit{unbiased}$ and $\textit{adaptive}$ sparsification techniques.
-----
[1] Peter Davies, Vijaykrishna Gurunathan, Niusha Moshref, Saleh Ashkboos. ``New Bounds For Distributed Mean Estimation and Variance Reduction''. ICLR 2021.

[2] Shay Vargaftik, Ran Ben Basat, Amit Portnoy, Gal Mendelson, Yaniv Ben-Itzhak, Michael Mitzenmacher. ``EDEN: Communication-Efficient and Robust Distributed Mean Estimation for Federated Learning''. ICML 2022.

[3] Ananda Theertha Suresh, Felix X. Yu, Sanjiv Kumar, H. Brendan McMahan. ``Distributed mean estimation with limited communication.''. ICML 2017.

-----
2. $\textbf{Computation Time. }$
The encoding time of Rand-Proj-Spatial is $O(kd)$. The decoding time is $O(d^2\cdot n\cdot k)$, which is the time to compute the eigen-decomposition of a $d \times d$ matrix $T(\sum_{i=1}^{n} \mathbf{G}_i^T \mathbf{G}_i)$ of rank at most $nk$ and this is the computational bottleneck of the decoding process. In practice, the server has a lot more computational power compare to the clients (e.g., edge devices) and so it can afford to spend more decoding time.
We will make the computation time clearer and stress this as a limitation.

We also attached additional experimental results on the encoding and decoding time for power iteration and linear regression. As one can observe, when the number of clients $n$ is large, the $\textit{adaptive}$ encoder Rand-$k$-Wangni takes a longer time to encode compared to the other $\textit{non-adaptive}$ encoders.

---

### Decision · Program_Chairs · 2023-09-21

**Decision:**

Accept (poster)

**Comment:**

The paper studies the problem of distributed mean estimation and proposes a new estimator called the Rand-Proj-Spatial estimator and demonstrates that the new estimator uses correlation among the vectors effectively. The paper also theoretically analyzes a few special scenarios and shows experimentally that the algorithms work on a variety of datasets. All reviewers like the paper and I recommend acceptance. I encourage authors to incorporate reviewer comments in the final version.